# Genome-wide modelling of plant transcription factor binding captures regulatory variants associated with phenotypic traits

Fritz Forbang Peleke [1,2], Simon Maria Zumkeller[3,4] ✉, Dominic Schirmer [4], Gernot Schmitz [3], Thomas Hartwig [4,5], Julia Engelhorn[4,5,6], Sergius Weizel[4,5], Armin Otto Schmitt [7,8], Tobias Jores [4] & Jędrzej Szymański [1,3] ✉

The sequence-specific recognition of *cis*-regulatory elements (CRE) by transcription factors (TF) propagates genotype information to phenotypes. Understanding how genetic variation affects gene regulation remains limited by the diversity and complexity of CRE interactions. Here, we address this challenge using an explainable multi-label deep learning model trained on *A. thaliana* DNA-binding data to capture how CRE sequence, their broader sequence context, and syntax influence TF occupancy. Once trained, the model annotates cistrome-wide TF-binding sites and uncovers condition-specific regulatory syntax. By integrating genomic and GWAS data from *A. thaliana*, our approach predicts differential TF-binding and identifies regulatory gene variants within quantitative trait loci. Experimental validation highlights the link between *cis*-regulatory variation, gene expression, and phenotypic outcomes. Finally, applying our model to untargeted DNA binding assays in *Z. mays* under heat-stress conditions demonstrates its potential to characterize condition-responsive TF binding in phylogenetically distant crops.

*C is*-regulatory elements (CREs) control transcript abundance, splicing and RNA stability, driving phenotypic diversity and evolution[1]. Deciphering plant gene regulatory networks is complicated by the extreme diversification of transcription factor (TF) families[2,3]. In crops like *Zea mays* and *Brassica napus*, most SNPs linked to defensive traits via genome-wide association studies (GWAS) reside in intergenic regions, suggesting they function as CRE-associated quantitative trait loci[4–6]. Systematic studies of protein–DNA interactions confirm that genetic variation within these regions correlates with phenotypic variation, yet setting up functional links between non-coding variants and plant traits remains a significant challenge[7].

*Cis*-regulation is largely controlled by transcription factors, which recognise specific DNA sequences and affect chromatin structure. CREs form modules in promoters, enhancers, silencers and insulators, which each play distinct roles in gene regulation[8,9]. Enhancer regions can contain clusters of different TF binding sites[10], and the combinatorial binding of TFs has been linked to differential spatiotemporal transcriptional activity[11]. Deep learning models, particularly

[1]Leibniz Institute of Plant Genetics and Crop Plant Research (IPK), Seeland, Germany. [2]NPZ Innovation GmbH, Holtsee, Germany. [3]Institute of Bio- and Geosciences (IBG-4: Bioinformatics), CEPLAS, BIOSC, Forschungszentrum Jülich, Jülich, Germany. [4]Institute for Molecular Physiology, Faculty of Mathematics and Natural Sciences, Heinrich-Heine University Düsseldorf, CEPLAS, Düsseldorf, Germany. [5]Max Planck Institute for Plant Breeding Research, Cologne, Germany. [6]DIADE, Univ Montpellier, CIRAD, IRD, Montpellier, France. [7]Breeding Informatics Group, University of Göttingen, Göttingen, Germany. [8]Center of Integrated Breeding Research (CiBreed), Göttingen, Germany. ✉e-mail: s.zumkeller@fz-juelich.de; j.szymanski@fz-juelich.de

convolutional neural networks (CNNs), offer a powerful framework for decoding regulatory grammar, yet they perform poorly on individual variant effect prediction unless trained with much larger, finer-resolution functional genomics datasets[12]. Several deep learning methods based on CNN exist to study, for example, transcription factor binding sites (TFBS) directly from DNA sequences[13,14]. Deep-STARR is developed to understand motif syntax with respect to enhancer activity in *Drosophila melanogaster* S2 cells, thereby facilitating the development of synthetic enhancers[15]. Basset is developed to model DNA accessibility over hundreds of human cell types[16]. While most approaches modelled DNA binding as binary classification problems, BPNet provides an approach to base-resolution protein–DNA modelling[17]. Current classification approaches to model TF-DNA binding have developed independent binary classification models per TF[13,18]. In such approaches, the number of models increases with each transcription factor, increasing the computational cost and scaling poorly for genome-wide annotations.

In this study, we utilise public DNA affinity purification sequencing (DAP-seq) datasets[19], spanning diverse TF families, to train deep learning models capable of simultaneously identifying and classifying TF-DNA binding events at the family level. We demonstrate that these models effectively annotate untargeted DNA-binding assays, such as MNase-defined cistrome-occupancy analysis sequencing (MOA-seq) in *A. thaliana*. They also enable cross-species transfer learning in *Z. mays* under heat stress and predict the functional impact of genetic variants

on phenotypic traits using plant self-transcribing active regulatory region sequencing (STARR-seq).

## Results

### Modelling protein–DNA interactions in *A. thaliana*

To model DNA–TF interactions and the effects of variation in plant gene regulatory elements, we first trained convolutional neural network classifiers on a collection of *A. thaliana* DAP-seq experiments (Fig. 1a). A total of 219 ampDAP-seq and 349 DAP-seq datasets with a fraction of reads in peaks (FRiP ≥ 5%) spanning 46 TF families were used (Supplementary Fig. 1)[19]. Importantly, grouping the DAP-seq datasets by their respective TF families increased the number of binding events per class, improving data balance and training efficiency.

To generate the training data, the genome of *A. thaliana* was initially split into non-overlapping 250 bp windows, which were labelled based on overlap with TF families. These labels represented the experimentally observed binding events. A multi-label CNN was trained to map 250 bp sequence windows to TF family binding profiles (Fig. 1a).

Models were trained with chromosome-level cross-validation, with four chromosomes used for training and one for validation. The performance was evaluated genome-wide and on promoter regions using the areas under precision-recall (auPR) and receiver operating characteristic (auROC) curves. These models achieved

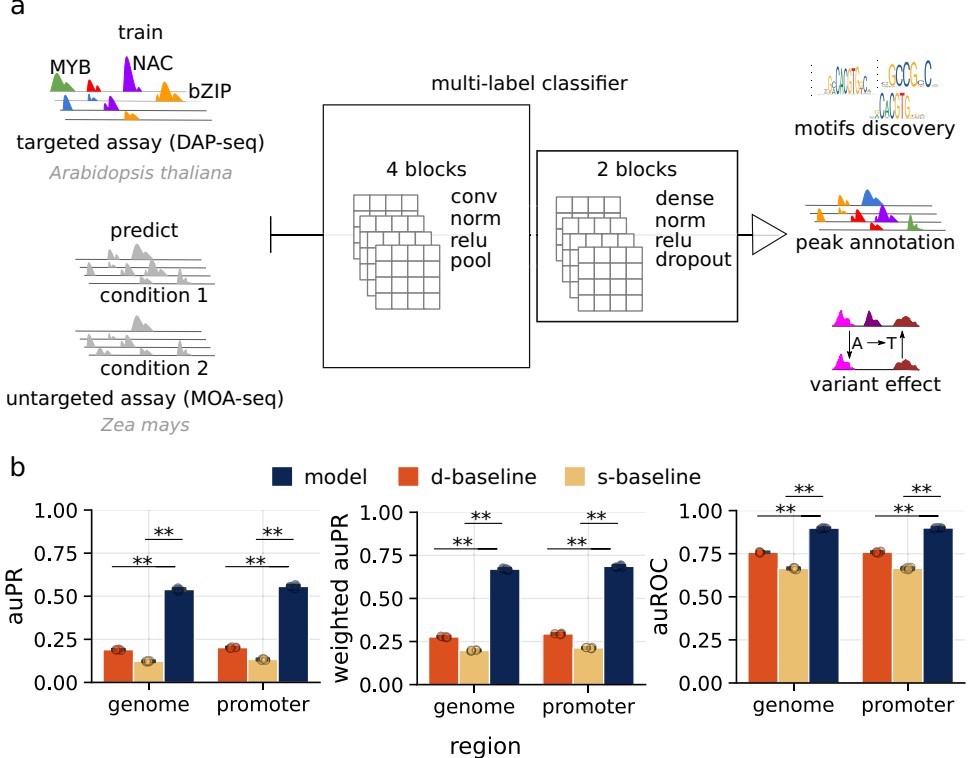

**Fig. 1 | Neural network architecture and performance trained for multi-label prediction of transcription factor family binding. a** The multi-label classification strategy is presented for modelling DNA-TF binding utilising deep convolutional neural networks. The predictive model for DNA-TF binding was trained on DAP-seq data derived from *A. thaliana* Col-0 and is applicable for the prediction of TF-binding sites within genomic DNA, such as from untargeted assays like MOA-seq across various species. The main CNN architecture consisted of four convolutional blocks, followed by two blocks of fully connected layers and a final output layer for multi-label classification. Each convolutional block was made up of a convolutional layer, followed by batch normalisation, ReLU activation and pooling (MaxPooling) layers. The final convolutional block was followed by two fully connected blocks.

Each fully connected block contained a dense layer, followed by batch normalisation, ReLU activation and dropout layers. Finally, the output layer used a sigmoid activation function with 46 output units. **b** The predictive performance of ($n = 5$) models, each trained on four *A. thaliana* chromosomes and evaluated on genomic bins and promoter-region bins from the left-out chromosome. This evaluation included a comparative analysis against single- and di-nucleotide shuffle controls (s-baseline, d-baseline). Statistical significance was assessed using a one-sided Mann–Whitney $U$ rank test (p-value *, **, *** <0.05, 0.01 and 0.005). Bar plots represent the mean performance across models, and error bars indicate standard deviations. Source data are provided as Source data files.

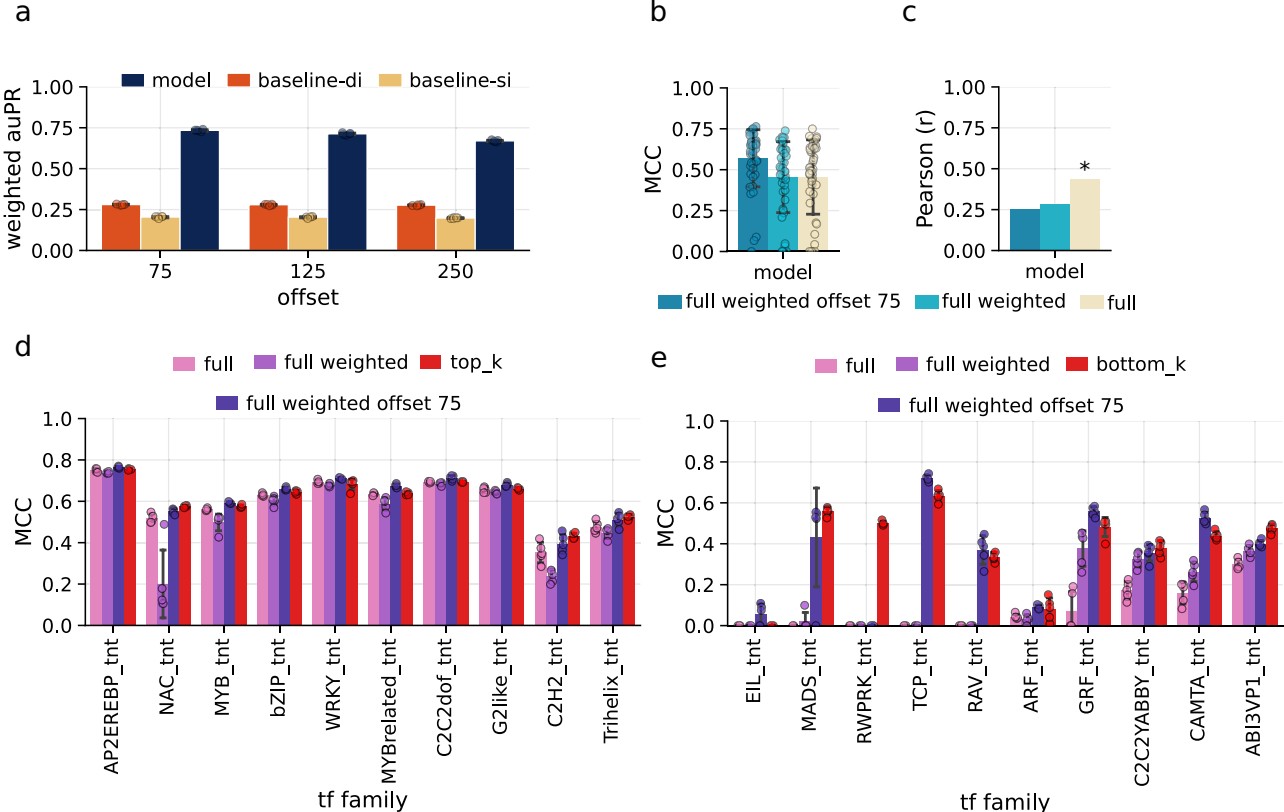

**Fig. 2 | Explanation of performance for the transcription factor binding sites (TFBS) predictive standard and optimised models of *A. thaliana*. a** The predictive performance measured as weighted auPR of models each trained on four *A. thaliana* chromosomes and evaluated on the left-out chromosome (*n* = 5). The effect of overlapping training windows on model performance was assessed using 250 bp windows with offsets of 75, 125 and 250 bp. This analysis also included a comparison against single- and di-nucleotide shuffle controls (s-baseline, d-baseline). Bar plots represent the mean performance across models, and error bars indicate standard deviations. **b** Comparison of model performance, measured by Matthews correlation coefficient (MCC), for models trained on the full dataset, the full dataset with weighted training, and the full dataset with weighted training using a 75 bp offset. All models were evaluated on the same test dataset generated with a 250 bp offset. Barplots show the mean Matthews correlation coefficients (MCCs) across (*n* = 46) TF families, and the error bars indicate standard deviations from the means. **c** Comparison of the Pearson correlation coefficients (*r*) between class abundance and class-specific model performance, measured by Matthews correlation coefficient (MCC), for models trained on the full dataset, the full dataset with

weighted training, and the full dataset with weighted training using a 75 bp offset (*p*-value *, **, *** <0.05, 0.01, 0.005). *P*-values are computed using a two-sided t-test. **d** Class-specific comparison of models trained on all 46 transcription factor (TF) families and on the subset of the 10 most abundant (top-k) TF families ranked by training data abundance. Barplots show the mean Matthews correlation coefficients (MCCs) across (*n* = 5) models for training on the full dataset, weighted training on the full dataset, weighted training on the full dataset with a 75 bp offset, and unweighted training on the ten most abundant TF families. Error bars indicate standard deviations from the means. **e** Class-specific comparison of models trained on all 46 transcription factor (TF) families and on the subset of the 10 least abundant (bottom-k) TF families ranked by training data abundance. Barplots show the mean Matthews correlation coefficients (MCCs) across (*n* = 5) models for training on the full dataset, weighted training on the full dataset, weighted training on the full dataset with a 75 bp offset, and unweighted training on the ten least abundant TF families. Error bars indicate standard deviations from the means. Source data are provided as Source data files.

$auPR_{genome} = 0.53$, $auPR_{promoter} = 0.55$, weighted $auPR_{genome} = 0.66$, weighted $auPR_{promoter} = 0.68$, $auROC_{genome} = 0.89$ and $auROC_{promoter} = 0.89$. Model performance was benchmarked against shuffled controls: a single-nucleotide baseline (S-baseline) and a dinucleotide baseline (D-baseline). Both controls significantly reduced auPR (Mann–Whitney *U* test, *p* < 0.01), indicating disruption of predictive sequence features (Fig. 1b). Genome-wide prediction across tiled windows showed that inferred binding events recapitulate genomic architecture, with enrichment of predicted binding events in proximal promoter regions and immediately downstream of transcription termination sites (TTS) (Supplementary Note 1 and Supplementary Fig. 2).

### Analysing the effects of class distribution on model performance

Despite strong overall performance, class-wise evaluation using the multi-label confusion matrix (MLCM;[20]), Matthews correlation coefficient (MCC), balanced accuracy and F1 scores revealed substantial

variability across TF families (Supplementary Fig. 3, Supplementary Data 1 and 2). Minority classes showed lower sensitivity and higher misclassification rates, and five families (EIL_tnt, RAV_tnt, MADS_tnt, RWPRK_tnt, TCP_tnt) were not learned. Performance correlated with class frequency (Pearson's *r* = 0.43, *p* = 0.002), consistent with known challenges in multi-label classification[21].

To mitigate the observed effects of class imbalance, we explored two complementary strategies aimed at upsampling the minority classes and adjusting learning dynamics. First, we modified the training sets by generating overlapping 250 bp sequence windows using step sizes of 75 bp and 125 bp, thereby increasing the effective representation of labelled regions. Second, we implemented class-weighted loss functions to penalise errors on underrepresented classes more during training. Introducing overlapping windows improved the model performance across several minority classes, suggesting that increasing the sampling density enhanced the model's ability to learn informative patterns from sparsely represented classes (Fig. 2a). Similarly, the use of weighted loss functions improved performance on

some minority classes, e.g. GRF_tnt (MCC = 0.37) and C2C2YABBY_tnt (MCC = 0.31) (Supplementary Fig. 4). However, this improvement came at a cost to performance on other families and a slight overall drop in model performance across all families. Interestingly, combining both strategies, class-weighted loss and 250 bp windows with a 75 bp step size, produced the most favourable balance. This optimised modelling approach not only improved performance on the minority classes but also yielded the best overall performance (Fig. 2b). To further characterise the differences in predictive behaviours of the three approaches, we computed the Pearson correlations across per-class MCC and observed that both approaches to mitigate class imbalance indeed showed no significant correlation to per-class frequencies (Fig. 2c). This suggests that the two approaches influence the learning process and the predictive distribution across families.

Furthermore, to examine the impact of label distribution on predictive performance, we trained sub-models restricted to the top-$k$ ($k = 10$) most abundant families and the bottom-$k$ ($k = 10$) least abundant families. This was to assess whether the minority classes were overshadowed by the more frequent classes. Training on the top-$k$ families resulted in a marginal performance gain relative to the full multi-label model, suggesting that highly abundant families were already being modelled effectively (Fig. 2d). In contrast, the bottom-$k$ model showed substantially stronger improvements across the subset of minority families, learning to predict the RWPRK_tnt family (MCC = 0.49) (Fig. 2e). However, some families (e.g. EIL_tnt, ARF_tnt) remained difficult to predict, indicating that factors beyond class frequency also contribute to performance disparities.

## Model interpretation for de novo identification of TF family-specific binding motifs

To interpret learned regulatory features, we applied SHapley Additive exPlanation (SHAP)[22] and TF-MoDISco[23] to correctly predict sequences from unweighted baseline models, identifying interaction predictive motifs (IPMs). IPMs were derived at the TF family level and clustered using motifStack[24], grouping families with similar binding preferences (Fig. 3a). These IPMs showed significant similarities to known motifs in the JASPAR database[25] and prior DAP-seq studies[26] (Supplementary Data 3). IPMs from the unweighted baseline and optimised models were largely consistent, with the optimised model additionally recovering low-abundance families (EIL_tnt, RAV_tnt, MADS_tnt, TCP_tnt) (Supplementary Fig. 5). Raw contribution scores per IPMs, extracted from the SHAPley scores, are provided in Supplementary Data 4.

Detailed analysis of the bZIP, bHLH and WRKY families showed that the models captured both canonical motifs and their variants (Fig. 3b and Supplementary Fig. 6a, b). WRKY binding sites (core 5′-TTGAC-3′) were consistently identified across 38 members with minimal variation. In contrast, bZIP models (core 5′-ACGT-3′; 42 members) captured diverse motif variants, including sequences associated with factors, such as AREB3 (5′-CACGTGNC-3′) and HY5 (5′-TGACGT-3′). While most bHLH members were predicted to bind the canonical G-box (5′-CACGTG-3′), others showed preference for variant motifs, including bHLH28 (5′-CCGTAC-3′), bHLH122 (5′-CAAGTTG-3′) and bHLH10 (5′-ACCGACA-3′) (Supplementary Fig. 6a).

Consistent with feature attribution highlighting TFBS-like IPMs, the models recovered on average 50.4–94.5% of known binding sites for AP2/EREBP, bHLH, bZIP, G2like, HSF, MYB and WRKY families from the JASPAR database across the genome (Supplementary Data 5), including motifs derived from assays beyond DAP-seq, demonstrating generalisability. For low-abundance families, IPMs closely matched experimental motifs (e.g. ARF2, ARF5, FAR1 and CAMTA2), despite low predictive performance in unweighted models (MCC: ARF = 0.437; FAR1 = 0.407; CAMTA = 0.167), indicating that poor performance was not due to failure in motif recognition. The optimised model resolved this by capturing consensus motifs and likely additional predictive features.

For medium- and high-performing classes, similarity between TF family motifs contributed to false positives, reflected by frequent co-occurrence predictions (Supplementary Fig. 6c and Supplementary Data 6). High false-positive rates were observed among families that fully or partially bind the G-box (bZIP, BES1, BZR, bHLH, FAR1 and CAMTA) and among AT-rich motif-binding families (HB, homeobox, ZFHD, ARID, C2C2YABBY) (Fig. 3c, d). Such redundancy in binding sites, including the shared G-box motif, is known to complicate sequence-based predictions[27].

## Sequence context and motif syntax contribute to predicting the binding specificity of different TF families

Occurrences of IPMs cannot explain the model's predictions alone, indicating that models likely learned extended sequence context (Fig. 4). To estimate the relevance of IPMs in their native sequence context, as utilised sequence features, we compared the model predictions against an alternative scenario where the IPM presence alone indicated binding. Relying solely on IPM occurrence yielded limited predictive power as expected, with an average false discovery rate (FDR) of 0.819 against DAP-seq data (Supplementary Data 7). Here, applying the model reduced the average FDR by 0.314 (Supplementary Data 8). Consequently, we concluded that the model utilises IPMs and combines it with an extended sequence context that is lost when motifs are aggregated to consensus sequences. For the evaluation of sequence context, we used three metrics. With co-occurrence per TF family, we measured how often different TFs appear together in the same sequence window (Supplementary Method 1). The offset independence per TF family determines how often the binding is predicted (above a predicted probability of 0.5) when shifting the input window within the genome (Supplementary Method 2). Essentially, it defines the smallest amount of sequence data the model needs to make a prediction without a change of the surrounding DNA. To evaluate the predictive importance of each IPM, we calculated the IPM's flanking sequence context importance value (IPMciv) (Supplementary Method 3). The IPM predictability for the TF families was inferred from IPMciv as the subtraction of one from the reciprocal of IPMciv. A high IPM predictability indicates that the prediction of binding does not rely on sequence context and vice versa.

Our analysis indicated that the model could detect multiple motifs and their sequence context at once. Most sequence windows predicted to be bound contain more than one TFBS (Fig. 4a). The model's MCC positively correlates with the number of TFBS predicted for one sequence window (Pearson's $r = 0.846$, $p$-value 9.12e-9; Fig. 4b). The average offset independence per TF family values exhibited a strong positive correlation explaining most of the variance between average MCC per TF family, too (Pearson's $r = 0.8$, $p$-value 4.3e-10, Fig. 4c). This indicates that TF families with high performance required less sequence information and were independent of their placement and the specific flanking sequence within an input window, accordingly. In parallel, the weighted average for specific TF co-occurrence shows moderate to strong positive correlation with MCC (Pearson's $r = 0.64$, $p = 2.7e-5$, Fig. 4d). This indicates that the TF family's predictive performance can be explained partially by co-occurrence. Optimal performance was observed with two to three binding sites per window (Fig. 4d). The average IPM context importance and predictability showed moderate positive correlation with MCC (Pearson's $r = 0.58$, $p = 5.4e-4$, Fig. 4e), indicating that sequence context can partially explain variances in the prediction given the IPMs presence. We selected outliers with high MCC and exceptionally high and low IPM context importance (0.1 percentile to the standard deviation) to further understand the model's limitations. The role of extracted IPMs for the low-performance TF-families with low data abundance is described in the Supplementary Note 2.

For instance, IPMs of AP2/EREBP (5′-GCGGC-3′) and BBR/BPC (5′-GAGAGA-3′) show high predictive performance, with an MCC of 0.75

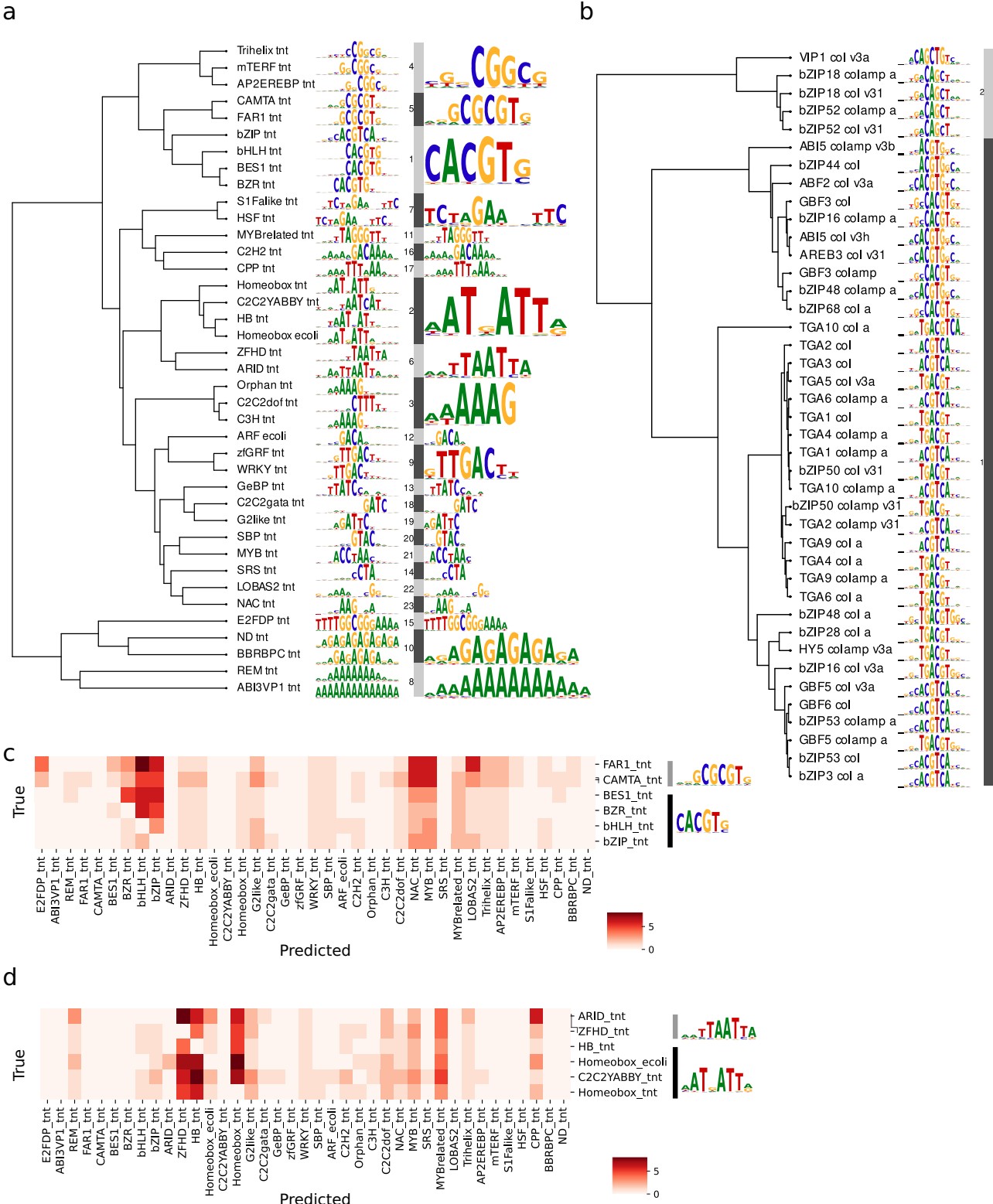

**Fig. 3 | Interaction predictive motifs and prediction of co-occurrence.**
**a** Extracted binding motifs for 39 TF families from the multi-label classification deep learning DNA-TF binding model. The binding motifs are clustered using motifStack. **b** Feature extraction of the multi-label classification deep learning DNA-TF binding model could retrieve 41 binding motifs of individual members of the G-box type bZIP transcription factor family (marked with a black rhombus in (**a**, **b**)). **c** The G-box binding TF families BES1, BZR, bHLH and bZIP, together with FAR1 and

the CAMTA, have similar binding motifs and show high rates of co-occurrence or calls for false positives and negatives, illustrated by the diagonally masked confusion matrix. **d** TF families with similar binding motifs, like the AT-rich ARID, ZFHD, HB, Homeobox, Homeobox_ecoli and C2C2YABBY show high rates of co-occurrence or calls for false positives and negatives, illustrated by the diagonally masked confusion matrix.

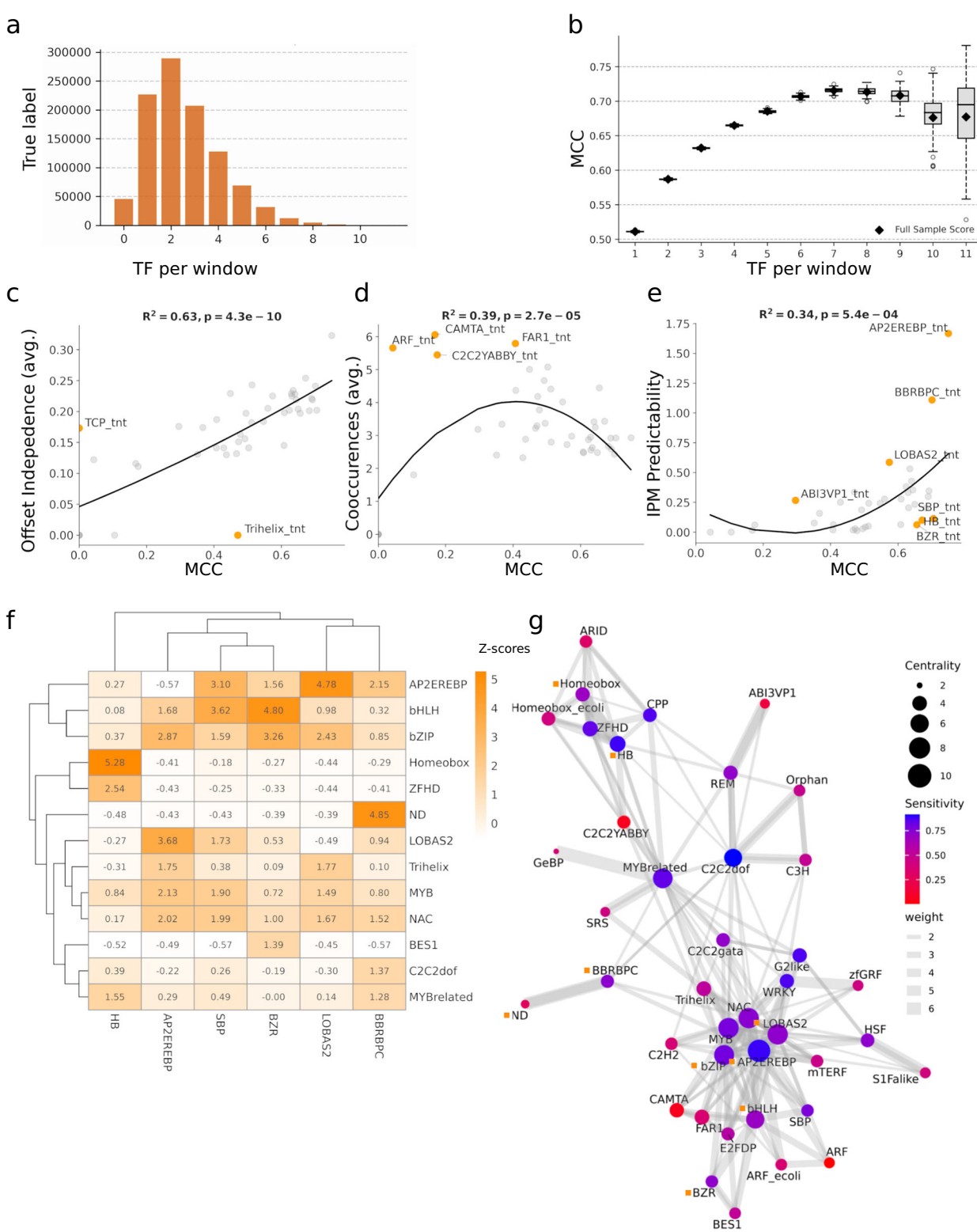

and 0.701, respectively, the highest scores of IPM predictability and the lowest context importance (Fig. 4e). On chromosome 1 of *A. thaliana*, 53,669 sites matched the AP2/EREBP IPM, but only 33,552 of these sites were predicted as bound. Of these, DAP-seq confirmed 24,356 occurrences as true binding sites. Similarly, 61,508 sites matched the BBRBPC IPM, with 32,337 sites predicted as bound (Supplementary Data 7). However, only 4,145 sites were confirmed to be bound in the DAP-seq data (Supplementary Data 7). Accordingly, the multi-label classifier overestimated the motif's importance, leading to false positives. However, the application of the model still increased

precision compared to baseline based on the occurrence of IPMs solely by 1.605 and 3.65-fold for AP2/EREBP and BBRBPC, respectively (Supplementary Data 8). The non-optimised model recovers 94.5% of characterised TFBS of the AP2/EREBP family from previous DAP-seq and PBM experiments, intriguingly, including ERF12, ERF14 and ERF17 that were not present in the training data, with 90%, 97.4% and 92%, respectively.

Similar to AP2/EREBP and BBRBPC, LOBAS2 (binding to 5′-AAANNCGG-3′) and HB (binding to 5′-AATNATT-3′) featured high MCC with 0.574 and 0.672, respectively (Fig. 4f). In the case of HB,

**Fig. 4 | Evaluation of the multi-label classifiers feature importances. a** Number of TF families with correctly predicted binding sites per 250 bp sequence input window. **b** Distribution of Matthews correlation coefficient (MCC) based on the number of correctly predicted TF family binding sites per 250 bp sequence input window. Boxplots represent the variance of MCC scores across 1000 bootstrap iterations, performed to mediate class imbalance across bins. For each boxplot, the central line indicates the median bootstrapped score, the box spans the inter-quartile range, and whiskers extend to the rest of the distribution (excluding outliers). Black rhombus markers denote the exact MCC score calculated on the full, un-resampled dataset for each bin. **c** Polynomial regression for the predictive performance measured in MCC for average input window offset independence per TF family. Outliers outside of the 10% confidence interval were labelled and coloured orange. **d** Polynomial regression for the predictive performance measured in MCC for a weighted average number of co-occurrences with other TF families per TF family. Outliers outside of the 10% confidence interval were labelled

and coloured orange. **e** Polynomial regression for the predictive performance measured in MCC for interaction predictive motif (IPM) predictability. Outliers outside of the 10% confidence interval were labelled and coloured orange. **f** Based on the count of co-occurrence of IPMs within predicted windows, $Z$-scores were calculated to check for IPM enrichment per modelled TF family and displayed as a heatmap for outliers of the 10% confidence interval of IPM predictability and sensitivity above 0.6. Additionally, G-box binding factors and significantly co-occurring TF families BES1, SBP and BZR and LOBA2, BBRBPC and AP2/EREBP are shown. **g** Network graphs were generated for all modelled TF families with edges showing co-enrichment $Z$-score ≥ 1. Edge thickness represents the degree of co-enrichment between nodes. Node colour reflects sensitivity, while node size corresponds to centrality, measured by the degree of connections within the TF family network. The sensitivity of the model to predict binding of TF families is higher the more central these are within the network. Source data are provided as Source data files.

158,699 sites mapped to *A. thaliana* chromosome 1, but only 14,483 were predicted to be bound, shown by high IPMciv of 0.804 and twice as high precision compared to baseline (Supplementary Data 7 and 8). AP2/EREBP, BBRBPC, LOBAS2 and HB all display TF families with high MCC, high offset independence and are on average co-occurring with two to three other TF families (Fig. 4f). AP2/EREBP and LOBAS2 frequently co-occur ($Z$-score 3.68), likely resulting from a partial overlap in IPMs. Similarly, BBRBPC exhibits few distinctive combinations and strongly co-occurs with identical binding sites of ND. Independently, HB forms its own cluster of co-occurring factors generally binding sites with high AT content, e.g. Homeobox, ZFHD, CPP or C2C2YABBY, but also sequentially different MYB and MYB-related sites ($Z$-scores 0.84 and 1.55) (Fig. 4f, g). This indicates that these TF families are predicted to occur in combinatorial patterns without necessarily sharing sequence homology. Overall, the network representation showed that the model's sensitivity in predicting TF family binding increases with their centrality within the network with significant moderate to strong correlation (Pearson's $r = 0.57$, $p = 0.00013$) supporting the notion that the models accurately reflect these complex interdependencies of TF co-binding (Fig. 4g). In addition, using the multi-label classification model to predict on sequences upstream of gene promoter regions, we could confirm that distinct TF families are more likely to co-occur than others based on their genomic location (Supplementary Note 3 and Supplementary Fig. 7).

To understand the model's varied performance for TFs that bind the G-box motif, we analysed the influence of co-occurring sequence features and perturbations flanking its core consensus motif (5'-CACGTG-3') on its predictive power (Supplementary Method 4). We found that BZR is exceptionally difficult to predict by IPM occurrence alone but surprisingly features a high MCC of 0.656 (Fig. 4e). Compared to IPM baseline occurrence, the model improved precision in calling binding of BES1 and BZR by 10.8 and 9.6-fold, and about two-fold for bHLH and bZIP (Supplementary Data 8). However, the predictions of the other G-box factors, BES1, bZIP and bHLH varied significantly from BZR. For example, around 25% of bZIP and bHLH sites were correctly predicted to be bound, while only 15% of BES1 sites were correctly predicted, in contrast to about 74% of BZR cases. The model's MCC for bZIP and bHLH was 0.63 and 0.644, respectively, compared to 0.503 for BES1. In this line, it was found that the model recovered on average 82.1% of previously characterised TFBS, including those for BZR and BES1 identified through DAP-seq and ChIP-seq ($n = 16$). This, for example, included BZR2, which was predicted to be bound in 67.3% of cases of the motifs occurrence, a positive regulator of brassinosteroid signalling, which was not included in the training data as it was characterised using ChIP-seq.

Analysis of IPM predictability and co-occurrence networks (Fig. 4g), in general, indicated that BZR predictions were influenced by combinations of other IPMs. For example, SBP co-occurred more frequently with bHLH, but not with BZR or BES1, despite their similar IPMs

(Supplementary Fig. 8a). BZR and BES1 predictions on windows with matching IPMs also showed distinct co-occurrence patterns with other TFs, such as MYB, LOBAS2 and NAC (Supplementary Fig. 8a). The systematic introduction of perturbations to the G-box core consensus motif flanking regions at +1/−1 bp positions significantly changed predicted TFBS co-occurrence with distinct other TF families (Supplementary Note 4 and Supplementary Fig. 8b). These results support that the model effectively recognised broader TFBS syntax that may represent *cis*-regulatory modules that shall be studied in the future. As for example, the co-binding predictions for bHLH104 were different from the predictions for bHLH13 (Supplementary Note 5).

## TF-binding potential classifies genes to 14 distinct regulatory clusters

With an established sequence-to-binding link, we aimed to generate an unbiased data-driven regulatory gene ontology. We clustered all *A. thaliana* genes for this purpose based on their predicted binding profiles. Predicted profiles were computed from the proximal (both 1.5 kbp) promoter and terminator sequences. Resulting TF-binding profiles were projected onto a lower dimension using uniform manifold approximation and projection for dimension reduction (UMAP) and clustered the genes using the hierarchical density-based spatial clustering of applications with noise (HDBSCAN) (Fig. 5). This analysis revealed 14 distinct clusters (which we refer to as regulatory clusters), indicating a high degree of specificity in the TF-binding potential of *A. thaliana* genes. Within each regulatory cluster, the sum of total predicted binding events per TF family highlighted the enrichment of unique combinations of TF families, with some families shared across the clusters and others occurring in few or specific to individual ones (Supplementary Fig. 9a).

The MYB, MYB-related, NAC, C2C2DOF, HB, G2like and AP2EREBP families of transcription factors were shared across all clusters, while SRS, mTERF, zfGRF, BES1, BZR, ARID and ABI3VP1 were specific to fewer clusters (Supplementary Fig. 9a). Based on the enrichment of different families of TFs, we have assigned names to each cluster using the top four most enriched families without considering the TF families that are shared across all clusters. For example, *c8-ZFHD-CPP-Homeobox-ARID* represents the 8th cluster, which contains genes with high potential to be bound by TFs from the ZFHD, CPP, homeobox and ARID families.

In the next step, we investigated the relationship between the regulatory profile of the genes and their function. Using the Mercator4-derived functional gene annotation[28], we performed functional enrichment analysis on the proteome of *A. thaliana* and examined whether these TF clusters could be associated with specific biological functions. Indeed, we observed a significant and systematic link between regulatory profile and functional annotation in some of the regulatory clusters (Fisher's exact test; FDR ≤ 0.05; Supplementary Fig. 9b). Out of the 14 regulatory clusters, eight were found to be

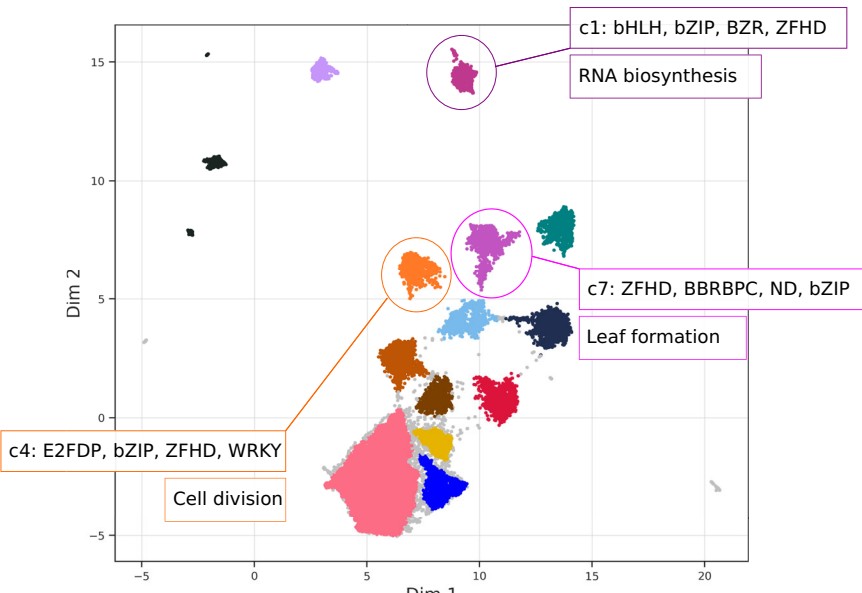

**Fig. 5 | Clustering of genes in *A. thaliana* using their predicted regulatory binding profiles.** UMAP of genes clustered based on the number of predicted binding events per TF family across promoters and terminators combined. Genes regarded as noise by HDBSCAN are in coloured light grey. Significant clusters with signature sets of TF families are highlighted (c1, c4 and c7), with the top enrichment ranked by log2 of Fisher's exact test of regulatory clusters in Mercator 4 biological functional categories.

significantly enriched within certain Mercator4 categories. For example, *c4-E2FDP-bZIP-ZFHD-WRKY* was significantly enriched in genes involved in cell division and chromatin organisation (Fig. 5). The cluster *c7-ZFHD-BBRBPC-ND-bZIP* was significantly enriched in genes involved in plant organogenesis, specifically in the formation of leaves (Fig. 5). This cluster was also enriched in genes involved in protein biosynthesis, protein modification and RNA biosynthesis, specifically ribosome biogenesis, protein phosphorylation and DNA-binding transcriptional regulation of the helix-turn-helix and beta-barrel DNA-binding domains, respectively (Supplementary Fig. 9b).

In addition, we analysed the relationship between the regulatory clusters and gene co-expression to determine whether shared TF-binding potential translates to coordinated transcriptional regulation. Using co-expression data from the ATTED-II database[29], we performed permutation testing ($n = 1000$) on the standardised co-expression logit scores (Supplementary Fig. 10). We observed that genes within eight of the fourteen regulatory clusters were significantly co-expressed (Benjamini–Hochberg (BH) FDR < 0.05). This analysis confirmed that similar TF-binding profiles are likely associated with gene co-regulation.

### Predicted TFBS dynamics explain gene expression changes

To compare the multi-label classifiers predictions for TF binding with experimental results, we used data from a deep mutational scan of the *Pisum sativum* (pea) rbcS-E9 enhancer[30]. The enhancer strength of these sequences was previously measured in tobacco leaves at light and dark conditions[30]. For the rbcS-E9 enhancer, several of the tested single-nucleotide substitutions resulted in a loss of predicted TF binding. According to the predictions, the tested region contains multiple TFBS with sensitivity to the single nucleotide substitutions (e.g. MYB-related, bHLH, bZIP and BZR) which were equally detected in the rbcS-E9 by Jores and colleagues[30] (Supplementary Fig. 11a). Consistent with the predicted loss of TF binding, the enhancer strength of these variants was significantly lower than the enhancer strength of variants that retained transcription factor binding (Supplementary Fig. 11a).

To further test the cross-species applicability of the TF model and for less well characterised enhancers, we predicted transcription factor

binding for a set of ~100 core promoters and variants thereof with one or two nucleotide substitutions[31]. The strength of WT and mutant core promoters was previously measured in maize protoplasts[31]. Core promoter variants with mutations that led to a loss of predicted HSF or S1Fa-like TFBS were significantly weaker than the corresponding wildtype core promoters (Supplementary Fig. 11b). This was not the case for the core promoter variants that lost binding sites for other TFs, however. These findings are in line with the original study, which demonstrated that HSF, but not bHLH, bZIP or WRKY, transcription factors are associated with increased promoter strength in the maize protoplast system[31]. In addition, we tested if regulatory effects are determined by combinations of TF families. We found a significant change in gene expression for the loss of PIF (a bHLH TF family member) in tobacco dark conditions (Supplementary Fig. 12). In addition, the model predicted complex combinatorial interactions, where predicted co-gain, co-loss, retained or unbound status of other TFs came with significant changes in promoter activity (Supplementary Fig. 12). For instance, there was no significant difference in expression between sequences with and without HSF binding in maize protoplasts. However, variants that resulted in parallel gain of bZIP, bHLH and AP2/EREBP, together with the loss of HSF, showed significantly higher rates of gene expression (permutation test $p \leq 0.05$) (Supplementary Fig. 12).

### Genetic variation affects TF-binding potential in loci associated with phenotypic traits

We demonstrated how single-nucleotide polymorphisms (SNPs) alter predicted TF-binding profiles that influence gene expression. Using 7364 AraGWAS SNPs associated with phenotypes like pathogen resistance and flowering time[32,33], we found that 20.72% of variants led to a gain or loss of predicted binding sites (Fig. 6a). To ensure these perturbations were not artifacts of the classification threshold, we performed two validations (Supplementary Note 6). First, we generated 30 in silico SNPs for each site, mirroring the specific nucleotide substitution at random positions. For every SNP that changed predicted binding, more than half of the in silico SNPs resulted in no perturbation, about 13% of cases resulted in the expected perturbation, and about 33% of cases resulted in a de novo perturbation (Fig. 6b).

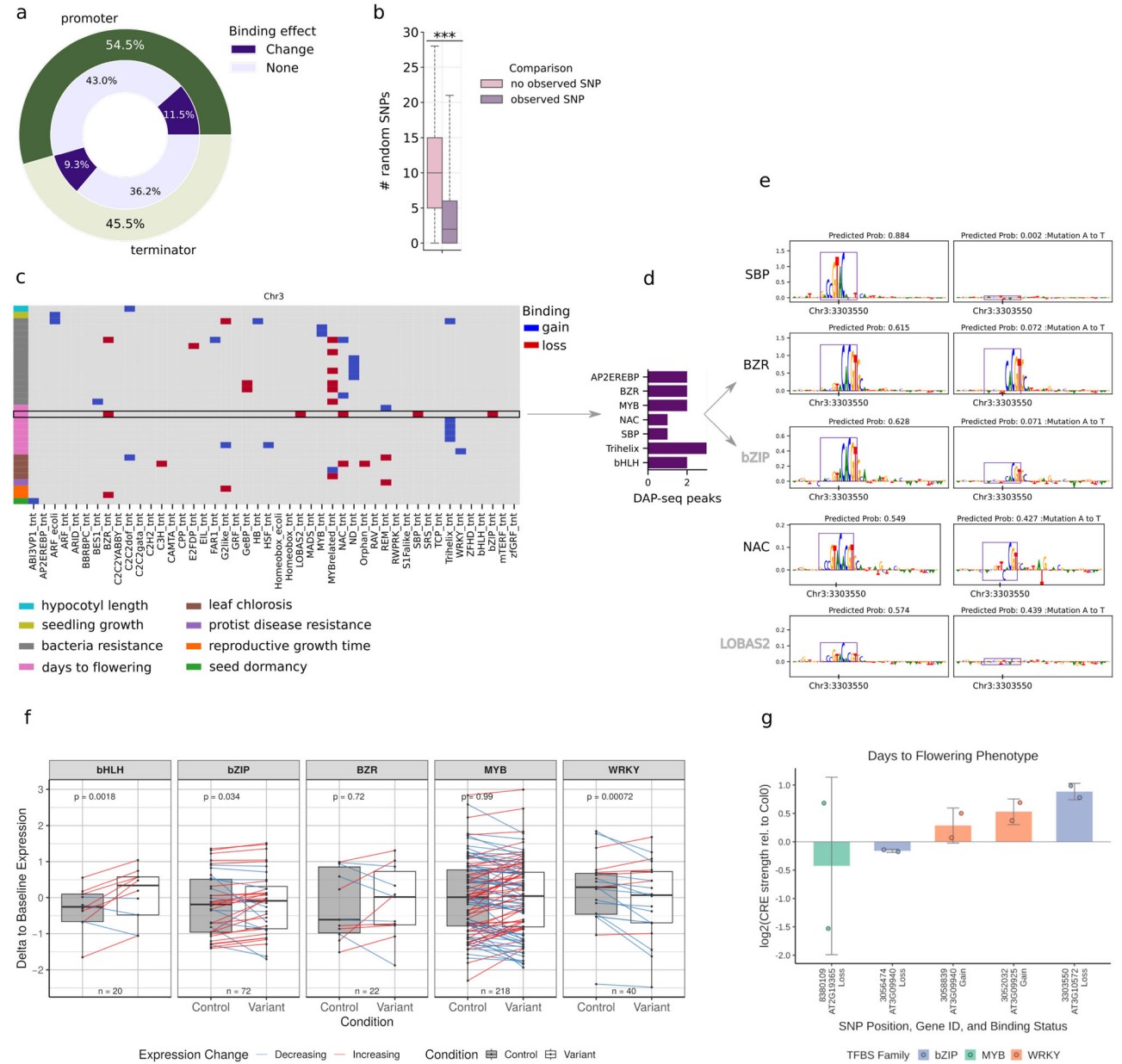

**Fig. 6 | Investigating the effects of single-nucleotide polymorphisms (SNPs) on the binding profiles of genes. a** SNPs found within the promoters and terminator regions of genes in *A. thaliana*, which were significantly associated with phenotypes across GWAS studies. **b** The effects of random mutations on predicted binding profiles, compared to a reference sequence with no mutations and a sequence containing a naturally observed mutation, across (*n* = 1433) SNPs. Statistical significance was assessed using a two-sided Mann–Whitney *U* rank test (*p* = 6.21e-168). Boxplots show the median and interquartile range. **c** A subset of mutations within the promoter regions of genes in chromosome 3 of *A. thaliana* and their effects on TF-binding. The rows represent individual significant SNPs-to-phenotype associations from GWAS. **d** DAP-seq observed binding events of transcription factors within the promoter of *AT3G10572*. **e** Saliency scores highlighting the loss in binding sites caused by an SNP within the promoter of *AT3G10572*. **f** Relative enhancer strength was assessed for *A. thaliana* Col-0 controls, and GWAS SNP

variants of bHLH, bZIP, BZR, MYB and WRKY transcription factors predicted to cause a loss in TF binding using plant STARR-seq in tobacco dark conditions. Boxplots display the median, with upper and lower 25th percentiles indicated by whiskers and outliers as individual dots. Positive or negative changes in measured transcript levels highlight differences between control and variant samples. *P*-values are calculated from a two-sided Wilcoxon rank-sum test. **g** Changes in relative promoter strength following predicted gain and loss of TF binding for GWAS variant constructs associated with the days to flowering phenotype, as measured by plant STARR-seq. Plotted loci include AT3G10572. Data are presented as the mean log2 relative strength of the variant compared to the Col0 baseline across *n* = 2 paired biological replicates from the high-throughput plant STARR-seq screening assay. Error bars represent standard deviation, with individual replicate pairs overlaid as distinct data points. Source data are provided as Source data files.

Second, to ensure that the predicted gains and losses in TF binding were not artifacts of minor statistical fluctuations around the classification threshold (e.g. marginal shifts between 0.49 and 0.51), we systematically quantified the change in predicted probability for all variants. Indeed, 41.73% of the binding changes exhibited a shift in probability larger than 0.2 (Supplementary Fig. 13).

We highlighted the SNPs within the promoters of genes found in chromosome 3 of *A. thaliana*, inducing gain and losses, which have been significantly associated with phenotypes related to leaf chlorosis, bacteria resistance, seedling growth, seed dormancy, hypocotyl length and 'days to flowering' (Fig. 6c). For example, an A-to-T mutation in the promoter of *AT3G10572* (linked to flowering time) altered predicted

binding for five families: bZIP, BZR, NAC, LOBAS2 and SBP (Fig. 6c, d). Saliency scores revealed this mutation flanks an adjacent G-Box motif essential for BZR/bZIP binding and disrupts a 5′-GTAC-3′ SBP core sequence (Fig. 6e).

We verified these biological impacts via plant STARR-seq on 340 control and variant enhancer constructs. We observed a significant increase in reporter gene expression for variants predicting a loss of bHLH or bZIP, and a significant decrease in reporter gene expression upon predicted loss of WRKY ($p < 0.05$). On the other hand, a significant increase of reporter gene expression came with predicted gain of MYB binding (Fig. 6f and Supplementary Note 7). Notable extremes included a 1.27-fold decrease in reporter gene expression for a BZR loss in *AT5G33439* (other RNA) and a 1.92-fold increase in reporter gene expression for a MYB gain in *AT5G67470* (Formin Homologue 6, FH6), both associated with climate adaptation. Further, we highlight our predicted regulatory effects against natural genetic variation and GWAS-associated traits, such as bacterial resistance and flowering time (Supplementary Note 7, Fig. 6g, e and 14). For the 'days to flowering' phenotype, we identified several natural SNPs that significantly altered both predicted TF binding and measured transcript levels. For instance, SNP chr.3: 3,303,550 resided within the coding region of *AT3G10570* (CYP77A6). While this variant results in a synonymous valine codon, it triggers a regulatory change in TF-binding potential.

Additional flowering-time SNPs, changing predicted TF profiles with measured change in gene expression, were found near *AT2G19365* (encoding a MYB/SANT-like DNA binding domain), *AT3G09940*, and *AT3G09925*. *AT3G09940* is affected by two natural variants: a bZIP loss that decreases transcript levels and a WRKY gain that increases them (Fig. 6g). This gene encodes MDAR3, a monodehydroascorbate reductase involved in mutualistic root interactions and ROS signalling downstream of ERF6 (AP2/EREBP family)[34,35]. The C-to-T mutation at *AT3G09925* increased transcript levels via a predicted WRKY gain. This variant also causes a non-synonymous proline-to-serine exchange in a flanking pollen ole e 1 and extensin family protein, a known master regulator of root regeneration[36]. These cases showed that natural selection often acts on variants with dual impacts; thus, predicting regulatory effects alongside gene function is essential to fully characterize the molecular drivers of phenotypic diversity.

## Model-based peak annotation enables genome-scale annotation from untargeted DNA binding assays

To independently demonstrate the applicability of our model, we performed TF-footprinting using MNase-defined cistrome-occupancy analysis (MOA-seq)[37] in *A. thaliana*. MOA-seq is a scalable approach to capture TF-binding sites globally in their native chromatin context within the genome[37]. It helps mitigate some of the limitations of methods like ChIP-seq, such as the need for TF-specific antibodies and the cost of performing a ChIP-seq experiment for each TF. However, the scalability of MOA-seq comes with trade-offs, such as the lack of annotation for captured binding peaks. We generated a set of genomic footprints and peak regions applying MOA-seq in Col-0 leaves and used them as input for the genome-wide analysis. In total, we identified about 60,000 significant putative TF-footprint peaks (MACS3, adjusted $p < 0.05$).

The *A. thaliana* MOA-seq peaks (Supplementary Data 9) were overlapped with DAP-seq peaks, which were then assigned to experimentally observed TF families. Then, our multi-label classifiers were used to annotate these peaks, and predicted TF-family assignments were compared to the overlapped experimental assignments. We observed that our model assigned about 90% of these peaks to at least one of the 46 TF families. When compared to the overlapped experimental assignments, our models were able to correctly assign about 74% of peaks to at least one TF family. Models trained on shuffled versions of sequences could only assign at most 8.3% of peaks to a TF family (Fig. 7a). Expanding our model annotations into the individual

TF families showed varying accuracies in annotating peaks from the different families, with our models being unable to assign peaks to the families that it did not learn a binding motif, such as the RAV, EIL, MADS and RWPRK families (Fig. 7b).

## *A. thaliana*-trained TF-binding model annotates stress-induced TF-binding in *Z. mays*

In the next step, we demonstrated the applicability of our models in annotating TF binding from homologous and heterologous experiments in another species. To evaluate the applicability of transfer learning, we assessed the model's performance on DAP-seq, ChIP-seq and MOA-seq data from the crop plant *Z. mays*[38]. Publicly available *Z. mays* ChIP-seq[39] and DAP-seq peaks[40,41] were preprocessed by centreing and extending them to 250 bp, an approach consistent with our modelling approach for *A. thaliana*. The TF family aggregated experimental results, specifically selecting those present in our models' training data. In addition, we processed MOA-seq data for *Z. mays* containing regions of the genome that showed differential binding activities between heat-stressed and control conditions[38]. The performance of our models on *Z. mays* DAP-seq datasets was similar to what we observed for *A. thaliana,* with a Pearson's $r$ of 0.797 for the average prediction accuracy for shared TF families. This demonstrated that the model's predictions can be transferred across species. However, predicting individual TF binding profiles across species proved challenging. For instance, while the AP2/EREBP family generally performed well, achieving average DAP-seq accuracies of 0.849 in *A. thaliana* and 0.741 in *Z. mays*, certain members displayed significant variability in the latter species. EREB71 and EREB49 in *Z. mays*, for example, showed very different accuracies of 0.839 and 0.374, respectively. The most probable explanation for these differences is the divergence in the evolution of TF family members, which likely resulted in altered binding profiles or a changed composition of the respective TF family.

Despite the high prediction accuracy of our models on DAP-seq datasets, we observed a decrease in performance across multiple families for their respective ChIP-seq data, with a Pearson's $r$ of 0.423 (Supplementary Fig. 15 and Supplementary Data 10). By comparison of shared individual TFs (EREB71, MYBR17, NAC49, WRKY53 and WRKY82) between the *Z. mays* ChIP-seq datasets and the datasets, we found prediction accuracies of 0.596 and 0.793, respectively. Interestingly, however, average accuracies were equally high for EREB71 from both data sources. Accordingly, we explain the differences in performance between DAP-seq and ChIP-seq by TF binding that is heavily dependent on cellular context, mediated by e.g. the chromatin contexts.

To assess the performance of the model on condition specific data, *Z. mays* MOA-seq peaks of control and heat-stress conditions were divided into two categories: peaks with negative and positive fold changes (Fig. 7c). Peaks with positive fold changes were those with a decrease in binding activity under heat stress, while those with negative fold changes showed an increase in binding activities under heat stress when compared to the control conditions. We framed a binary classification problem to classify peaks into either the positive or negative fold change category, with the expectation that our binary classifiers will focus on the heat-responsive TF families to distinguish the two classes of peaks. Binary classifiers were fitted using peaks with extreme response to heat (absolute fold change ≥1), mild response to heat (absolute fold change <1) and all peaks. For each model trained on *A. thaliana*, we used this model to predict binding profiles for the MOA-seq peaks in *Z. mays*. These predicted binding profiles were fitted to logistic regression models to generate binary classifiers of heat response. We fitted 25 (5 *A. thaliana* models × 5-fold cross-validation) logistic regression models for each subset of the dataset: mild, extreme and all. Our models achieved average accuracies of about 80% in all three subsets (Fig. 7d). Finally, we assigned importance scores to each TF family using in silico perturbation of the predicted binding

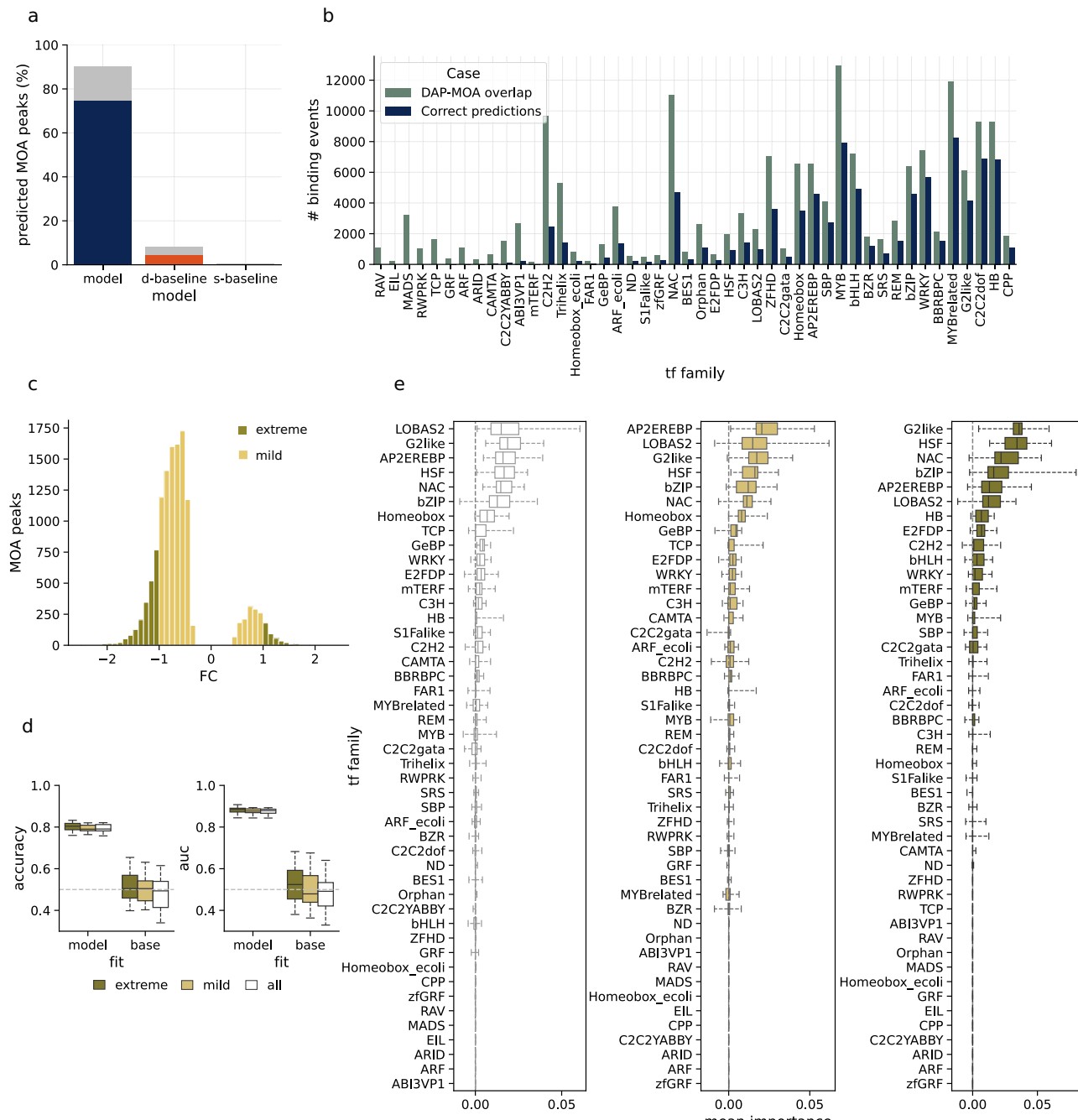

**Fig. 7 | Annotation of MOA-seq peaks from *A. thaliana* and *Z. mays*. a** Annotation of MOA-seq peaks from *A. thaliana* using models trained on real and shuffled data. The total percentage of peaks assigned to at least one TF family (grey), correctly assigned to at least one family (navy, dark orange and light orange). **b** The number of MOA-seq peaks assigned to each family (navy) alongside the observed number of experimental binding events based on DAP-seq overlap (green). **c** Distribution of fold changes for MOA-seq peaks captured in *Z. mays* under heat stress conditions. Peaks with a mild response to heat stress are considered to have an absolute fold change <1, while the rest are considered to be extremely responsive to heat. **d** Performance of logistic regression binary classifiers built using predicted binding

profiles for each family (Fig. 7e). The permutation importance scores highlighted the importance of families such as the HSF, G2like, NAC, bZIP, LOBAS2 and AP2EREBP in heat stress response. It also highlighted the increased importance played by TFs from the HSF family in response to heat stress as we move from mild to extreme responses.

profiles from *A. thaliana* multi-label classifiers, evaluated across ($n = 5$) classifiers. Binary classifiers were trained on predicted binding profiles from four *A. thaliana* chromosomes and evaluated on the left-out chromosome. Classifiers assign peaks to either negative or positive fold changes in binding in response to heat stress. Box plots mark the median, upper and lower quartiles and 1.5× interquartile range (whiskers). **e** Feature importance for each TF family shown as boxplots of mean importance scores across 30 permutations for ($n = 5$) binary classifiers. Importance scores are defined as the change in model accuracy when the predicted profile for each TF family is independently permuted. Box plots mark the median, upper and lower quartiles and 1.5× interquartile range (whiskers).

## Discussion

By shifting from traditional single-motif scanning and binary classification to a deep learning multi-label model[13,18], we provided a highly scalable approach for annotating untargeted, genome-wide footprinting assays (such as MOA-seq) and facilitated cross-species

transfer learning. Using this model, we made three primary biological findings.

The first finding is capturing the importance of regulatory grammar and context for TF binding. We provide an in-depth explanation of the deep learning model, with a detailed description of limitations regarding motif detection and how these biological features guide prediction. Our study indicated that isolated sequence motifs are not the only critical features for predicting DNA-TF binding; factors such as the 3D shape of DNA[42], DNA methylation[19], and complex combinatorics heavily influence these interactions[27]. The importance of understanding of *cis*-regulatory modules, assemblies of TFBS, has been considered paramount in crop improvement[9]. In this line, a relatively simple, computationally inexpensive, deep learning architecture effectively captures the regulatory grammar to accurately resolve complex interactions among highly similar TF families. For example, resolving the binding combinatorics of G-box recognising families (such as BES1, bHLH and BZR) is particularly challenging due to their high sequence similarities and genome-wide abundance. While in vitro DAP-seq experiments often miss in vivo binding events that require partner proteins - detecting only ~8.5% of BZR1 targets in *A. thaliana*[19,43]- our model inferred these dynamics, similar to findings from in vivo hybrid allele-specific ChIP-seq in maize[44]. This aligns with previous knowledge that factors like bHLH and bZIP can act antagonistically or as competing homo/heterodimers for identical binding sites[43,45,46], highly dependent on specific flanking sequences[27,47].

The second major finding is a gene co-regulation analysis driven not by co-expression but by the similarity of the TF binding profiles. This method grouped all *A. thaliana* genes into just 14 quite distinct functionally and co-regulated clusters, effectively linking promoter sequence with gene function and gene expression. For example, the *c4-E2FDP-bZIP-ZFHD-WRKY* cluster was significantly enriched in genes linked to cell division and chromatin organisation[48–50]. Similarly, the *c7-ZFHD-BBRBPC-ND-bZIP* cluster was enriched in categories involved in mRNA silencing through miRNA pathways[51]. The low number of regulatory clusters indicated that a wide range of possible tissue, development, and stimulus-related gene expression patterns emerge from a limited number of TF binding combinations.

The third major finding of this study is attributed to predicting variant effects on phenotypes. Our variant effect analysis provided a complementary tool to GWAS by linking associated signals to molecular phenotypes, specifically changes in predicted DNA-TF binding activities[52]. By defining the functional impact of a non-coding SNP as its induced change on expected binding sites, we established a simple method to functionally annotate regulatory gene variants. In our example, a natural SNP associated with flowering time was predicted to alter the binding profiles of BZR and NAC transcription factors, which are known to play core roles in floral transition[53,54]. This approach can be easily extended to other phenotypes and associations and performed on any other population and GWAS data.

This brings us to the utility of the model, which we believe could be exploited mostly in cross-species transfer learning for species with limited genomic and DNA-binding data. This applies in particular to untargeted DNA binding assays. Methods like MOA-seq or ATAC-seq provide an affordable, high-throughput approach to capture genome-wide chromatin accessibility in native contexts[37,55], but inherently lack annotations regarding which specific TF is bound to a recorded peak. Since MOA-seq is a relatively recent technique, ATAC-seq data is an alternative assay, providing a larger body of available data, and a large overlap with MOA-seq across different species (Supplementary Note 8)[37,55]. Integrating our model with these assays creates a reciprocal refinement: the assay's tempo-spatial specificity filters out the model's potential false positives, while the model resolves the identity of TFs within MOA-seq peaks. This synergy is expected to not only lower FDRs but also anchor the model's predictions in a native, in vivo context, bypassing the inherent biases of in vitro protein–DNA assays

like DAP-seq. In a transfer learning setup, our model–trained exclusively on *A. thaliana*–functionally annotated MOA-seq peaks in the distantly related crop plant *Z. mays* in heat stress conditions and highlighted differential binding and gene targets of key transcription factors in heat stress responses, including HSF, bZIP and AP2EREBP families[56–58] and specific heat shock regulators[59,60].

Finally, it is important to mention the limitations of the model. Most of all, we observed weak predictive performance across certain TF families, which can be largely attributed to data scarcity and biological data imbalance[61]. Moreover, transcription factor families bind CREs at intrinsically different rates[19,62], leading to an uneven representation of binding sites within experimental datasets. We recognise that this inherent class imbalance leads to severe underfitting for low-abundance TFs in the long tail of the dataset, while conversely inflating FDRs for over-represented TFs coupled with TFBS homologies. While grouping TFs into families and employing offset training partially mitigated these issues, the challenge of capturing the full spectrum of TF interdependencies remains. To further address this, we provide extensive model explainability metrics–such as IPM context importance–allowing researchers to track and interpret potential misclassifications. Ultimately, we demonstrate that refining training data composition significantly addresses these shortcomings, and, consequently, the systematic integration of future genomic datasets will likely resolve remaining gaps in model coverage.

A final limitation of the model is its reliance on a fixed 250-bp input window, which can complicate variant effect predictions in regions containing multiple redundant TFBS of the same family. In such cases, a localised mutation might be masked by intact neighbouring sites within the same window. We observed this with the AB80 and Cab-1 enhancers[30], which contain multiple MYB binding sites. Because that study tested only single-nucleotide mutations, our model did not predict a loss of MYB binding for the overall 250-bp window, even though experimentally, mutating these individual sites decreased enhancer strength. To overcome this limitation in spatial resolution, we highly recommend pairing the model's broad predictions with genome-wide IPM annotations and calculation of importance scores, all outlined here, which pinpoint the exact sub-window motifs driving the model's binding classifications.

## Methods

### Data source

The reference genome, proteome and annotation of *A. thaliana* (TAIR10) were downloaded from the Ensembl Plants database (release 59). The reference genome RefGenV4 of *Z. mays* was downloaded from the Ensembl Plants database (release 50). We also downloaded processed DAP-seq peaks for 568 TFs grouped into 46 families belonging to *A. thaliana* from http://neomorph.salk.edu/PlantCistromeDB. The whole genome of *A. thaliana* was split into 250 bp windows without overlaps using bedtools[63]. The windows were further overlapped with peaks from each of the TF families to generate an overlap matrix using bedtools. A window was considered to be bound by a specific family if it covered at least 70% of a peak belonging to that family. The sequence windows were accordingly classified and split into training and validation sets for a multi-level classification task. Sequence windows from four chromosomes were used for training, while windows from the remaining chromosome were used for validation.

### Model architecture and training strategy

We modelled TF-DNA binding as a multi-label classification problem with the help of CNNs. The main CNN architecture consisted of four convolutional blocks, followed by two blocks of fully connected layers and a final output layer for multi-label classification. Each convolutional block was made up of a convolutional layer, followed by batch normalisation, ReLU activation and pooling (MaxPooling) layers. The final convolutional block was followed by two fully connected blocks.

Each fully connected block contained a dense layer, followed by batch normalisation, ReLU activation and dropout layers. Finally, the output layer used a sigmoid activation function with 46 output units. Model training was done using the Adam optimising algorithm, with a learning rate of 0.002. In addition, we applied early stopping of the training procedure if the model's validation loss did not improve after five epochs. We performed 5-fold cross-validation such that, in each cross-validation step, windows from four chromosomes were used for training the model, while windows from the left-out chromosome were used for validation.

## Feature extraction

We used a combination of several methods to assess our models, to understand what they have learnt and identify possible drawbacks to their performance. Firstly, a multi-label confusion matrix (MLCM) was created for our models using the MLCM implementation of ref. [20]. This helped us elucidate the TF families for which our models learnt a potential binding motif and showed possible sources of false positives and negatives. Secondly, for each family, we used a combination of SHAP and TF-MoDisco to generate importance scores and motifs, respectively[22,23]. Precisely, for each TF family, nucleotide-resolution importance scores were computed for all windows that were correctly predicted to be bound by transcription factors belonging to that family. These scores were then used to generate interaction predictive motifs (IPMs) using TF-MoDisco[23]. Interaction predictive motifs were further clustered using the motifStack[24] package in R to highlight families sharing similar binding motifs. A masked-diagonal multi-label confusion matrix, clustered based on similarities of IPMs, was also generated to highlight the relationship between motif similarity and false positive predictions.

## Model explanation

We designed specific methods to systematically analyse and interpret the predictions of the TFBS predictive model based on sequence information. We used the IPMs occurrence and generalisation of TFBS Family motif consensus as baseline for interpretation. The framework integrates three metrics to explain the specific prediction sensitivity: (i) window offset independence, which quantifies the stability of a IPM prediction across shifting sequence windows to assess its locational certainty and context-dependency; (ii) weighted average co-occurrence, a metric that characterises the potential importance of TFBS-like IPM combination by calculating the average density of binding sites within a given factor's predictive context; and (iii) IPM predictability, which measures the predictive power of a canonical sequence motif relative to its broader genomic context. By correlating these quantitative properties with model performance, our approach provides a mechanistic explanation for prediction outcomes.

The window offset independence of TF families was estimated for the upstream regulatory regions of 7156 genes on chromosome 1 of *A. thaliana*. These regions were defined as 1.5 kbp sequences spanning 1 kbp upstream and 0.5 kbp downstream of the annotated TSS. IPM predictions were generated across these regions using a 250 bp sliding window with a 10 bp step size, ensuring that a single IPM can be detected in a maximum of 24 consecutive windows. The offset independence score for each predicted TF family was calculated as the fraction of continuous positive predictions within this sliding window framework. For further explanation, please see the Supplementary Method 1. Quantifying TF co-occurrence evaluates the potential of combinatorial binding on model performance. Predicted binding events within each 250 bp window were first categorised into discrete bins based on the total number of distinct TF families predicted to bind (from 0 to 11). To account for the substantial imbalance in the number of samples per bin, performance metrics (sensitivity, F1 score, micro and Matthews Correlation Coefficient) for each bin were derived from 100 bootstrap replicates. To further quantify the specific co-

occurrence tendency for each TF family, we calculated a weighted average co-occurrence score. For further explanation, please see the Supplementary Method 2. To evaluate the importance for predictions of IPMs alone and their context, we developed the IPM context importance value, where the number of windows predicted for binding and containing the IPM is divided by the number of total IPM occurrences of the same TF family. For further explanation, please see the Supplementary Method 3. We mapped and extracted IPMs against chromosome 1 of *A. thaliana* using BLAMM[64] and filtered the results using custom R scripts. To systematically investigate whether a model's predictive performance for a given transcription factor can be explained by other calculated properties of that factor, such as its positional independence, co-occurrence and reliance on the cognate IPMs presence, we performed second-degree polynomial regressions (Fig. 5). We further tested for positive enrichments of co-occurring IPMs, calculating $Z$-scores and Fisher's exact test to generate a graph network highlighting prominent co-occurrence of TFs and highlighted the TFs sensitivities, respectively. To assess the sensitivity of our models to TF co-occurrences, in silico mutations were introduced next to canonical TFBS sites on *A. thaliana* chromosome 1 identified by the mapping of IPM. For further explanation, please see the Supplementary Method 4.

To evaluate the added value of the deep learning model over traditional motif-scanning approaches, we utilised annotated motifs from JASPAR2026, including TFBS from different experimental sources (e.g. DAP-seq, ChIP-seq and SELEX etc). The *A. thaliana* genome was scanned on both forward and reverse-complement strands, and genomic 'hits' were recorded whenever a sequence achieved a score larger than 85% of the motif's maximum possible theoretical score using FIMO[65]. We matched predictions of the presented model with biological metadata from the JASPAR database (including TF families, classes, and associated PubMed IDs), limiting the analyses to a subset of TF families (see Supplementary Data 5). We calculated a recovery rate for each motif, representing the ratio of predicted binding sites correctly identifying a database motif.

## Linking regulatory clusters to gene co-expression

Co-expression data were obtained from the ATTED-II database[29] to assess the relationship between regulatory binding clusters and gene co-expression. Permutation testing ($n = 1000$) was performed on standardised logit-transformed co-expression scores. The null distribution was generated by randomly sampling gene pairs from the 14 defined clusters to control for background composition. Empirical $p$-values were calculated as the proportion of permutations with mean co-expression greater than or equal to the observed cluster mean, followed by Benjamini–Hochberg FDR correction.

## Model validation

The model's predictions were experimentally validated using promoter activity assays (STARR-seq), and its predictive performance was additionally assessed via transfer learning on genetic variants in *A. thaliana*, MOA-seq and DAP-seq data from the distantly related monocot *Z. mays*.

To validate the model's predictive performance, its predictions were compared against two published experimental datasets containing functional measurements of sequence variants. First, to assess predictions within *A. thaliana* the model was used to predict the loss, gain or retention of TFBS for single-nucleotide variants from a deep mutational scan of the rbcS-E9 enhancer and multi-nucleotide variants from -100 promoter region[30,31]. These predictions were then correlated with the experimentally measured enhancer strength for each variant. Of note, the other two enhancers tested in the experimental study, AB80 and Cab-1, tested by Jores and colleagues[30] contain multiple binding sites for MYB transcription factors. Since only single-nucleotide mutations were tested in that study, none of the variants

was predicted to lose binding by MYB transcription factors. Mutations targeting the individual MYB TFBS did, however, lead to decreased enhancer strength.

Second, to test the cross-species applicability of the model, predictions were generated for a set of approximately 100 maize core promoters and their corresponding variants containing one or two nucleotide substitutions. The predicted loss of binding for specific TF families (e.g. HSF, S1Fa-like, bHLH, bZIP and WRKY) was compared against previously measured promoter strengths from a maize protoplast assay[31].

Third, to evaluate the model's ability to predict the functional impact of natural genetic variation, we used single-nucleotide polymorphisms (SNPs) identified in trait-associated GWAS in *A. thaliana*, obtained from the AraGWAS catalogue[33]. We focused our analysis on 7364 SNPs located within 1.5 kbp gene flanking regions (1 kbp upstream and 0.5 kbp downstream of the TSS). The model was applied to predict gain or loss of TFBS resulting from these specific SNPs. We evaluated the significance of predicted binding changes using a background perturbation model. For each natural SNP, we generated 30 controls in silico mutations by introducing identical nucleotide substitutions at random positions within the prediction window. The SNP's impact was then assessed against the distribution of effects from these random mutations to identify significant binding perturbations.

A library comprising 369 control and variant sequence pairs, yielding 351 pairs after experimentation, was constructed for selected phenotypes identified from AraGWAS SNPs. These sequences exhibited relevance to the transcription factor binding site (TFBS) families bHLH, bZIP, BZR, MYB and WRKY. Sequences containing BsaI recognition sites were excluded to preclude interference with library preparation for STARR-seq experiments. In instances of identical or overlapping nucleotide sequence windows, only one representative with less than 50% overlap was used for synthesis, and a mapping was established to facilitate subsequent analysis of both versions. Sequences were trimmed from 250 bp to 170 bp for STARR-seq experiments, and the dynamics of gain/loss were assessed both prior to and following trimming. Sequences were ordered as an oligo pool from Twist Biosciences and inserted into the plasmid pPSntF (Addgene no. 256230; https://www.addgene.org/256230/) by Golden Gate cloning using BsaI-HFv2 (NEB).

Plant STARR-seq experiments were performed with this library in *Nicotiana benthamiana* leaves under dark conditions[66]. Log2-scaled transcript enrichment analysed regulatory activity changes in comparison to a minimal CAMV 35S promoter construct. Two replicates per construct were tested, and the variance formed a background distribution. The relative change in regulatory activity was calculated from the median delta of control and variant construct replicates enrichment scores before and after inoculation. Variants that cause a gain or loss of TF binding sites impact gene expression, and were tested for an overall effect using a paired Wilcoxon test for significance.

### Plant cultivation and *A. thaliana* MOA-seq

*A. thaliana* Col-0 plants were grown under controlled greenhouse long day (LD) (16 h light, 8 h dark, 35000 lux) conditions for four weeks. Rosette leaf tissue of three biological replicates (>12 plants each) were harvested and immediately frozen in liquid nitrogen. Collected tissues were pooled and homogenised and processed for MOA-seq library constructions following a previous protocol[37] except for the following modifications: 150 mg of leaves were used to prepare 70 μL of nuclei, to which 7.8 μL of 50 U/mL MNase (Worthington) were added. During nuclei preparation, additional protease inhibitors, Pepstatin A (2 μg/mL) and Aprotinin (2 μg/mL), were added to the fixation solution. RNA was digested just after MNase treatment by adding 2.5 μL of RNAse A (20 mg/mL) to the stop EGTA solution used to stop the MNase reaction and incubating at room temperature for 30 min. DNA purification was performed by extracting once with chloroform:isoamyl alcohol (24:1) followed by purification with the NEB Monarch. PCR and DNA cleanup kit, following the instructions for oligonucleotide cleanup.

We prepared the library and performed sequencing on a NovaSeq 6000 S4 flow cell in 100 bp paired-end mode at the Cologne Center for Genomics (CCG)[37]. The three biological replicates yielded a total of 127,818,588 merged reads. For MOA peak calling, we determine the average fragment length using samtools (v1.16)[67], and the effective genome size with unique-kmers.py (https://github.com/dib-lab/khmer/). MACS3 (v3.0.0a7)[68] was used to determine significant peaks with parameters '-q 0.05 -g {effective genome size} -s {fragment length} --min-length {fragment length} -max-gap {2x fragment length} --nomodel --extsize {fragment length} --keep-dup all'. Bigwig files containing normalised MOA-signal (counts per million) were generated using the bam Coverage function of deeptools (v 3.5.0)[69].

### Reporting summary

Further information on research design is available in the Nature Portfolio Reporting Summary linked to this article.

## Data availability

The data (DAP-seq datasets for *A. thaliana*) used for training and validation of our models were adopted from O'Malley et al.[19]. The data for validation on *Z. mays* ChIP-seq and DAP-seq data used for cross-species evaluation were downloaded from the NCBI GEO under accession GSE137972, GSE120304, and GSE275897. The data for validation on MOA-seq raw data for *Arabidopsis* leaves was submitted to the SRA archive under BioProject PRJNA1225493. Peaks called from the MOA-seq data have been made available in narrowPeak format (Supplementary Data 9). Source data are provided with this paper.

## Code availability

Custom code and trained models used in this study are available at the GitHub repository [https://github.com/NAMlab/DeepCistrome].

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

## Acknowledgements

The authors want to thank CEPLAS for the financing of S.M.Z., the IPK Gatersleben IT team for providing and maintenance of the computing cluster, and all members of the Network Analysis and Modelling lab at the IPK Gatersleben for discussions and suggestions. J.S., S.M.Z. were funded by the Deutsche Forschungsgemeinschaft (DFG, German Research Foundation) under Germany's Excellence Strategy—(EXC-2048/1–project ID 390686111). T.H. was funded by DFG, ID:458854361, as part of DFG Sequencing call 3 and the European Union's Horizon Europe program BOOSTER (ID: 101081770 to T.H.).

## Author contributions

F.F.P. designed and trained the deep learning models, evaluated their performance, and performed feature selection and transfer learning. S.M.Z. characterised TFBS syntax and conducted the selection and analysis of plant STARR-seq data. G.S. curated sequences for the plant STARR-seq experiments, and D.S. conducted the validation experiments of AraGWAS variants. S.M.Z., T.J. performed the underlying bioinformatic analyses. T.H., J.E. and S.W. generated MOA-seq data for *A. thaliana*. F.F.P., S.M.Z., T.J. and J.S. designed figures and wrote the manuscript. A.O.S. and T.H. consulted on data interpretation and critically revised the manuscript. J.S. designed and supervised the study.

## Funding

## Competing interests

The authors declare no competing interests.

## Additional information

**Supplementary information** The online version contains Supplementary material available at https://doi.org/10.1038/s41467-026-73634-8.

