## [Peer Review File · Nature Communications]

Modelling regulatory variant effects from transcription factor binding syntax across plant species using deep learning

Corresponding Author: Dr Jędrzej Szymański

Version 0:

Reviewer comments:

Reviewer #1

(Remarks to the Author)

The manuscript "Modelling genetic variation effects in plant gene regulatory networks using transfer learning on genomic and transcription factor binding data" by Peleke et al. introduces a deep learning framework using multi-label convolutional neural networks (CNNs) to predict transcription factor (TF) binding in *Arabidopsis thaliana* based on DAP-seq data, with extensions to *Zea mays* under heat stress and integration of genomic and GWAS data to link cis-regulatory variants to phenotypic traits. A key contribution is the integration of motif syntax and sequence context to predict TF binding across 46 families. However, the model's practical utility is limited by its strong dependence on available DAP-seq data, lack of validation in cross-species applications, and reduced performance for underrepresented TF families.

Major Issues

1. Predictive utility diminished by dependency on available DAP-seq data

A central motivation of the study is to use deep learning to predict TF-DNA interactions in scenarios where experimental data are lacking. However, the model's performance is clearly dependent on the number of DAP-seq datasets available for a given TF family. In fact, families with few experiments (e.g., ARF, RAV, EIL) show poor prediction sensitivity, while well-represented families drive the model's overall performance. This raises a conceptual caveat: if good predictions require abundant DAP-seq data, then the model mainly recapitulates known binding patterns — and the added value of prediction is diminished. The manuscript would benefit from a clearer articulation of when and why prediction is useful, especially in contexts where experimental binding data do not yet exist (e.g., untested conditions, accessions, or species). Otherwise, the utility of the model risks being circular.

2. Experimental validation gaps

While model performance is assessed *in silico* (e.g., through MOA-seq peak annotation and motif recovery), no orthogonal *in vivo* validation (e.g., ChIP-seq or CRISPR perturbation of predicted sites) is provided to confirm the functional relevance of newly predicted binding events or motif combinations. The predictive power of combinatorial IPMs (e.g., bZIP + SBP in Fig. 4c) would benefit from even a modest experimental confirmation.

3. Model Bias toward overrepresented TF families

As shown in Fig. 1b and 2a, the model performs poorly for underrepresented TF families (e.g., ARF, RAV, EIL). While the family-level grouping mitigates label imbalance to some extent, the manuscript could explore strategies like data augmentation or weighted loss functions to better support rare classes. This becomes critical when such TFs are functionally important (e.g., ARFs in auxin signaling).

4. Evaluation metrics may overestimate performance in imbalanced data

The authors report model performance using AUROC and AUPR, but these metrics can overestimate performance in the presence of strong class imbalance — a known issue in multi-label classification tasks like this one. Many TF families have few experimental positives, making AUPR especially volatile, and AUROC insensitive to the actual quality of positive predictions. Yet, the manuscript does not report more robust metrics such as the Matthews Correlation Coefficient (MCC), which remains informative even when positive examples are rare. Including per-family MCC values or micro-/macro-averaged MCC across TF classes would provide a more rigorous and interpretable assessment of predictive power — especially for underrepresented families like ARF or RAV. Without such metrics, performance claims risk being inflated by class imbalance.

5. Cross-species generalization requires caution

The model's application to *Zea mays* (Fig. 7) is an interesting demonstration of cross-species prediction, but it raises several concerns. First, the authors apply *Arabidopsis*-trained models to maize MOA-seq data without addressing species-specific motif evolution or TF orthology. This overlooks the divergence in cis-regulatory codes and TF repertoires between these species. While Fig. 7e highlights stress-responsive TF families (e.g., HSF, NAC), it remains unclear whether these predictions align with the TFs actually expressed and active in heat-treated maize.

Second, the study lacks quantitative validation: if any *Z. mays* ChIP-seq or DAP-seq data exist, they should be used to evaluate the model's performance directly. Reporting precision, recall, or other overlap metrics between predictions and available experimental data would substantially strengthen confidence in the model's generalizability. Without this, the claims of cross-species transferability remain speculative.

6. Interpretability metrics not fully quantified

The SHAP- and MoDISco-based motif extraction is central to claims of model interpretability. However, the predictive value of IPMciv (e.g., Fig. 4a) is only weakly correlated with model sensitivity ($R^2 = 0.39$). A clearer breakdown of true vs false positive contributions per IPM, and their contribution to variant effect predictions (e.g., Fig. 6e), would strengthen conclusions about motif grammar.

7. GWAS-Variant Interpretation Overstated

In Fig. 6, the authors explore how SNPs affect predicted TF binding and report that ~21% of SNPs in flanking regions cause gain or loss of binding. However, the interpretation of this result remains shallow. They assess binding perturbation by comparing to random *in silico* mutations, but do not validate whether predicted binding changes correspond to the direction or function of the associated phenotype. For example, do SNPs that cause TF binding loss tend to be associated with lower gene expression or phenotypes consistent with loss-of-function?

Moreover, given that each SNP is tied to a GWAS trait, the authors could perform enrichment analysis: are SNPs with predicted TF binding perturbation overrepresented among SNPs associated with certain phenotypic categories (e.g., flowering time, defense response)? Alternatively, they could ask whether predicted binding-affecting SNPs are more likely to lie near trait-relevant genes. Without such analyses, the model's utility in linking regulatory variation to phenotypes remains speculative. This represents a missed opportunity to use GWAS as an independent, biologically grounded validation layer.

Minor Issues

1. Interpretation and validation of motif syntax

The analysis of motif syntax and sequence context (lines 203-285, Fig. 4) is conceptually interesting but lacks clarity and experimental grounding, making it difficult to assess its biological relevance. The manuscript discusses combinatorial TF binding and syntax-dependent regulation, but specific examples and their implications are not well-developed or validated. Without this, the motif syntax analysis risks being perceived as speculative rather than actionable.

2. Figure quality and layout require major revision

The figures fall short of Nature Communications standards in both clarity and presentation. The manuscript contains six full-page figures, many of which are overcrowded, inconsistently formatted, and difficult to interpret without extensive cross-referencing (e.g., Figs. 3–5). Axis labels are often small, legends are densely packed or buried in text, and some color schemes (e.g., in motif similarity heatmaps) lack accessibility. Complex plots (e.g., UMAP clustering, co-enrichment matrices) could benefit from clearer labeling and simplified visuals. Several panels could be consolidated or moved to supplementary figures to enhance narrative focus and visual digestibility.

3. Typos throughout the manuscript: For instance, "epmArth-S0-p1m00" and "epmArthS0-p0m01" motif names appear inconsistently formatted; unify notation for clarity.

4. Method detail: The model architecture (Lines 636–648) could benefit from a schematic to clarify the CNN structure and layer dimensions. While Figure 1a provides a schematic, the manuscript lacks specific details about the CNN architecture (e.g., number of layers, filter sizes, activation functions, or regularization methods). This omission makes it difficult for readers to fully understand or replicate the model. The statement "a CNN was trained as a multi-label classifier" (line 122) is vague without mentioning hyperparameters or architectural choices. For a machine learning audience, this lack of specificity hinders assessment of the model's robustness and novelty. A supplementary table or section detailing the architecture would enhance clarity.

5. Data and code accessibility: While the manuscript mentions supplementary tables, it does not clearly state whether the full codebase, trained models, and input data (e.g., processed DAP-seq peak sets, MOA-seq calls, SNP annotations) are publicly available. For a computationally intensive and largely *in silico* study such as this, reproducibility is essential. The authors should provide a clear statement on code and data availability, ideally with links to a version-controlled repository, including the complete model training pipeline, evaluation scripts, and example inputs for downstream applications (e.g., variant effect prediction, cross-species analysis). This is particularly important given the paper's emphasis on transferability and broader applicability.

6. Method clarity: While the method is mostly clear in its high-level design and objectives. However, clarity is compromised by insufficient technical details (e.g., CNN architecture, preprocessing steps), lack of justification for key parameters (e.g.,

window size, cross-validation strategy), and dense explanations of complex concepts like IPMciv and motif syntax. These issues may challenge readers, particularly those outside computational biology, and could hinder reproducibility.

(Remarks on code availability)

Reviewer #2

(Remarks to the Author)

This study presents a deep learning framework that predicts transcription factor (TF) binding in plants by modeling DNA-protein interactions as a multi-label classification task. Using DAP-seq datasets from *Arabidopsis thaliana*, the authors trained convolutional neural networks to identify sequence features predictive of binding across 46 TF families. The model captures both motif presence and contextual dependencies, enabling accurate prediction of TF binding. It was validated using MOA-seq experimental data, applied to predict the effects of genetic variation (SNPs), and successfully transferred to maize to annotate stress-induced TF binding. The approach offers an interpretable, scalable tool for linking non-coding variation to gene expression regulation and phenotype across species.

First off, I wish to congratulate the authors on this comprehensive, very methodical and insightful study, and the meticulously prepared manuscript. The breadth of this study is impressive indeed. It is also apparent that a lot of care went into the manuscript. It is well written, a rich set of results are presented very clearly, each supported by corresponding figures or parts thereof, each prepared with a good eye for detail and visual clarity.

The approaches are sound, proper controls were performed. A particular strong point is the application of the prediction model to a different technology (MOA-seq) and transfer to *Zea mays*, a crop species.

I have no major points of critique (perhaps one: the non-overlapping tiling), but want to offer some thoughts that the authors may want to consider. In addition, I list some minor points in need of correction or clarification.

- The genome was tiled into 250 bp segments without overlaps. The DAPseq narrowPeak sequences are all of length 202 bp. Thus, a fixed and overlap-free tiling might create boundary effect problems. I do realize that the actual TF-binding site motifs are shorter, but because you trained on the tiled 250 bp input sequences, actual DAPseq peaks may often be assigned to two bins and with their respective flanking sequence - which you go on to demonstrate that they are important - cut off on one side, which will affect the proper representation of flanking sequences. Why did you not choose an interlaced tiling?

- I very much liked the clustering approach (Fig 5). While the segregation into 14 clusters in the UMAP is convincing, it is indeed surprising to realize how similar they actually are with regard to motif profile (Fig 5b)- as also noted by the authors. The authors demonstrated the association of the clusters with different gene functions (via GO-term enrichment). I suppose the authors have also thought about a comparison of the regulatory clusters with actual expression-based gene clustering. What is the agreement (Rand index)? With RNAseq or microarray cross multiple tissues (e.g. Trava-db) are different conditions available, it should be straightforward to perform this analysis and comparison.

- On the clustering (Fig 5): what was the metric, actually? Euclidean distance on presence/absence of TF-families? If binary bit-strings were compared, then Jaccard index might be the better option.

- On the SNP-variant effect prediction: ~20% of the GWAS-candidate SNPs were predicted to affect TF-binding. To put this number into perspective (thereby possibly addressing this question), random SNPs (same change) were introduced (=nice!) of which half did not affect binding, 13% resulted in the same change as reported for the actual SNP-variant, and 33%(!) resulted in a de novo perturbation. My question: did you do the random test on the 20% set or all candidates? If the former, than 13% is really low (in support of your hypothesis that the GWAS candidate variant is likely interfering with TF-binding), if the latter, than it is 20% vs. 13%, plus 33% de novo effects sounds high to me. I would appreciate some more guidance or clarification in the text on this question.

- Gain and loss of binding - how was this determined? I suppose, threshold on the prediction. How set?

- It was interesting to note that your model was more conservative with regard to binding site prediction than just detecting consensus motifs, while, compared to DAP-seq support, still too "optimistic". As, perhaps, a further point of discussion: you considered sequence context to explain the noted discrepancy between predicted and DAP-seq-confirmed sites. It has recently been suggested that the sequence context exerts its effect on TF-binding by affecting the mechanical properties of the local genomic DNA (doi.org/10.1093/nargab/lqad095). The strength of your NN, of course, is that you implicitly capture any such effects. Though, the boundary-tiling issue (see above) may interfere with prediction fidelity, as you actually do not capture the flanking sequence correctly?

Minor points:

- perhaps it would be good to remind the reader that DAP-seq reports on TF-binding to "naked" DNA-fragments.
- Eqs 1 and 3, symbol in the denominator got lost.
- l666 - spell out IMP again here.
- I don't quite understand the sentence l403, starting with "However,...74%..." What experimental assignments?

- In places the fonts got mixed up (Just to notify you).
- Figure labels would have been nice (as would have been page numbers, but at least, line numbers were given)
- l242 (0,3%) => (0.3%)
- spell out numbers less than 13, e.g. l124 "4 chromosomes" => "four chromosomes" (matter of style, I know...)
- l85: I would start a new paragraph, starting with "Enhancers..."

(Remarks on code availability)

The authors are to be commended on the level of care with which they made the code available on github. The documentation is well organized and informative. Code for all reported results has been made available, allowing to re-run the analyses, which I have not attempted, but it appears to be easily doable. The code also provides additional details that are missing in the manuscript (details of the NN-architecture, which I am fine with to find them here only).

Reviewer #3

(Remarks to the Author)

This study presents a deep learning framework for predicting transcription factor (TF) binding profiles using DAP-seq data across TF families in *Arabidopsis thaliana*. By training a multi-label classifier on 250 bp windows, the model leverages shared sequence features to predict family-level TF binding and applies these predictions to accessible chromatin in both *Arabidopsis* and *Zea mays*, including heat-stress conditions. Performance metrics such as AUROC and AUPR are generally favorable for TF families with sufficient training data, though these metrics are less informative for families with few examples or those with highly similar motifs.

The approach underperforms for families with sparse training data, degenerate or complex sequence context features, or high motif similarity between TFs (e.g., bHLH, BZR, bZIP, BES1), which can lead to false positives. The authors acknowledge these limitations. The incorporation of the IPM_{civ} metric to examine binding context beyond core motifs is an interesting step and does provide additional potential information regarding how sequence properties beyond motif may contribute to binding. While it does appear that they are capturing interesting properties, such as motif clustering from different families, it is not always unclear whether the signal the model identifies can be easily explained in terms of specific sequence features.

A key area where the paper could be strengthened is in clarifying model performance using more intuitive, interpretable overlap metrics. While AUROC and AUPR offer valuable statistical summaries, simple Jaccard-style comparisons—quantifying predicted-only, observed-only, and shared TF binding sites—would offer readers a much clearer sense of prediction fidelity. This is especially important given the well-known issue of motif-based methods drastically overcalling TF binding sites relative to DAP-seq or ChIP-seq data. The model may help reduce such inflation, but this is not directly shown. Supplemental Table 4 provides raw site counts but doesn't explicitly break down site overlap. A figure panel summarizing this breakdown across several representative TF families (those with strong, moderate, and weak performance) would significantly enhance clarity and accessibility.

Application of the model to annotate MOA-seq peaks provides a plausible test of biological relevance, but the informativeness of these predictions is hard to evaluate. While it is encouraging that predicted or observed sites overlap MOA peaks, it remains unclear what fraction of predicted peaks fall into these "TF footprint" regions, and whether this overlap exceeds motif-based expectations. For *Zea mays*, the observed increase in HSF binding contribution under extreme heat is an interesting result, but the broader biological interpretability of the analysis remains limited as I don't know if it's well described that HSF has higher relative contribution to phenotype in extreme heat compared to medium heat.

In addition to the MOA analysis a more direct assessment of how well the model performed in predicting binding sites would be to benchmarking against predicted binding sites from scanning with *Arabidopsis* DAP-seq motifs. While the ideal comparator would be a direct comparison to maize DAP-seq data, but due to the unavailability of this data, a useful proxy would be comparing motif-based predictions to model-predicted sites. Such an analysis would help determine whether the transfer learning approach meaningfully improves prediction and how in general it recapitulates motif-matching results with potentially avoiding overcalling common to motif-matching.

The clustering of TF families into shared 250 bp windows is a creative way to explore potential co-binding modules. While interesting, the biological significance of these clusters remains unclear. Functional enrichment using Mercator4 categories yields some suggestive results, such as cell division for E2FDP-containing clusters, but most categories are generic and offer limited insight (e.g., RNA biosynthesis). Moreover, the contribution of individual TFs within each cluster is not always parsed. For example, the enrichment in the E2FDP cluster likely reflects well-characterized E2F target genes, which is the master regulator of cell division. While co-regulation may exist as the E2F would largely explain this functional assignment it is not clear what role the other TFs are playing and as such hard to assess. Critically, the total number of binding sites or putative target genes contributing to these multi-TF clusters is not provided, making it difficult to judge their prevalence and functional importance. If these clusters represent only a small fraction of all predicted sites, their biological significance may be correspondingly modest.

The application of the model to predict the impact of natural variants on TF binding is one of the study's most promising extensions. While limited to TFs with sufficient training data and non-redundant motifs, this use case illustrates how the model could be applied to study the regulatory effects of genetic variation. The approach would be valuable, even if its utility is currently constrained to well-behaved TFs.

In sum, this work provides a thoughtful and technically solid framework for TF binding prediction and its downstream applications. The authors are transparent about limitations, and the methods are broadly applicable. However, the manuscript would benefit from clearer benchmarking against DAP-seq data, simplified and more intuitive summaries of prediction overlap, and a more detailed accounting of co-binding cluster sizes and contributions. These improvements would make the study's conclusions more accessible and compelling to a broad genomics audience.

(Remarks on code availability)

Reviewer #4

(Remarks to the Author)

Fritz, Jedrzej and their colleagues developed a multi-labelled deep learning model trained on DAP-seq data from *Arabidopsis thaliana* to capture how DNA sequence features, their context, and syntax influence transcription factor occupancy across the genome. Additionally, this model possesses a certain degree of transferability, enabling it to annotate transcription factor binding sites on the maize genome with good performance. Below are some of my concerns:

1. Training dataset: the authors only used DAP-seq which evaluated the in vitro motifs of TFs. There are also a large number of in vivo motif data (CUT & Tag or ChIP-seq) in *Arabidopsis*, which may further enhance the training model by increasing additional model layers of tissue specificity.
2. Cross-validation: the authors used 4 chromosomes (80%) for training and the remaining chromosome for validation (20%). Under this strategy, the model performance was not too high. I highly suggest the authors to shuffle the genome into five or ten discontinuous sections, and re-evaluate their model by 10-fold cross-validations.
3. When implementing transfer learning in maize, the non-TF-specific DNA-binding assay used was MOA-seq. Based on my experience, the data quality of MOA-seq is not as good as that of ATAC-seq, and there are many high-quality ATAC-seq data available in maize.
4. Throughout the article, there is no comparison with other transcription factor prediction software or models. For example, what is the difference between the transcription factor binding sites predicted by the AI model used in this article and those identified traditionally by MEME?

Minor:

1. There are numerous formatting issues throughout the manuscript, including but not limited to the following:
Line 74: The use of bold text appears unjustified.
Lines 111–113 and Line 326: Multiple font styles are used inconsistently.
Lines 416–425: A calligraphic or cursive-style font is used, which negatively affects readability.
Lines 674 and 685: The equations are not fully rendered.
Similar formatting inconsistencies are found elsewhere in the manuscript. The authors are advised to carefully review and standardize the formatting throughout the document.
2. The selection of training window size (250 bp) was arbitrary. The size of CREs in plants spans only dozens of basepairs.
3. Line 42-43, "Cis-regulation refers to the mechanism by which DNA sequences control the transcription rate of nearby genes.....", I'm not sure the use of "transcription rate" was accurate. Any citations? In my opinion, "transcription outcome" or "transcription abundance" may be more accurate.
4. The model predicted some point mutations that might disrupt motif bindings, (e.g. Line 374-375). Experimental validations, at least one or two cases, are needed to further verify the effect of these point mutations.

(Remarks on code availability)

Version 1:

Reviewer comments:

Reviewer #1

(Remarks to the Author)

The revised manuscript shows meaningful progress in several areas, particularly in the addition of functional validation experiments and the incorporation of more rigorous performance metrics. These improvements strengthen parts of the authors' original claims and represent a substantial effort to address earlier feedback. However, two fundamental issues remain largely unresolved and continue to limit the rigor and readability of the work.

First, the submission lacks figure legends and contains figures that are difficult to interpret, raising concerns about overall presentation quality and editorial care. Second, despite new analyses, much of the manuscript remains descriptive rather than quantitative, particularly in its interpretation of model behavior and motif-level features (e.g., IPMs). As reflected throughout the detailed comments below, key claims are still supported by curated examples rather than systematic evaluation, and central interpretability metrics remain underspecified or ambiguously validated.

Together, these unresolved issues undermine confidence in the robustness and clarity of the study, even in light of the substantial revisions made.

nM1:

The authors acknowledge the dependence on DAP-seq dataset availability and attempt to mitigate it via weighted loss and cross-species validation. However, the empirical demonstration that the model generalizes well to TFs or conditions with sparse or no training data remains limited. I encourage the authors to provide a more systematic analysis of performance under data-limited scenarios in the main text rather than relying on curated examples in supplemental figures. For instance, what's the overall trend/correlation between sample size and model performance?

The authors attempt to counter the concern of diminished predictive utility by showing that known motifs can still be recovered for weakly performing TF families (e.g., ARF). However, this response conflates feature attribution (motif enrichment) with predictive performance. The ability to identify known motifs in low-performing models does not establish meaningful generalization or reliability. A model can overfit or capture background sequence biases and still return recognizable motifs — especially when interpretability methods like SHAP or TF-MoDISco are applied post hoc.

Moreover, citing better motif recovery from ARF_ecoli compared to ARF_tnt highlights known experimental biases in DAP-seq, not successful generalization by the model. This reinforces the concern that the model is heavily influenced by dataset composition and lacks robustness under data-sparse or heterogeneous conditions.

In summary, the authors' interpretation suggests a fundamental misunderstanding of the distinction between motif attribution and predictive accuracy. Without quantitative benchmarks or prospective evaluation in low-data regimes, the argument that "some motifs can still be found" does not resolve the original conceptual flaw.

nM3:

While the authors implemented weighted loss and report mixed outcomes; however, this does not eliminate the underlying issue that model performance is strongly dependent on TF families with abundant high-quality data. This reinforces the broader conceptual concern raised in M1 — that the model's predictive utility is limited in biologically relevant scenarios where data are sparse, thereby constraining generalization and interpretation.

M2:

The authors have substantially improved the manuscript by including both re-analysis of public Plant STARR-seq datasets and new experiments on 340 variant enhancer constructs. These additions provide convincing functional support for the model's predictions and motif-based interpretations, particularly regarding combinatorial TFBS (IPMs).

M4|M6:

The authors have commendably incorporated MCC and other robust performance metrics throughout the revised manuscript. However, their interpretation of the model's biological relevance remains heavily reliant on the concept of interaction predictive motifs (IPMs) — composite features derived from SHAP explanations and TF-MoDISco clustering. The use of IPM and IPMciv raises several concerns:

a) Conceptual Ambiguity: The manuscript does not provide a clear, quantitative definition of IPM specificity or how these motifs are distinguished from background sequence patterns or spurious correlations.

b) Biological Validation: There is limited orthogonal validation to support the functional relevance of IPMs.

c) Potential for Misinterpretation: In families such as BBRBPC, the model exhibits high sensitivity but also extremely high FDR (up to 98%). Yet IPMs from these families are interpreted as predictive, which may misrepresent model reliability—especially in the context of label imbalance and family-specific data scarcity.

Major concern: figure quality remains unresolved

Despite earlier feedback, figure layout and clarity remain below publication standards. Moreover, the submitted version lacks figure legends, making it difficult to assess whether the visualizations are interpretable, self-contained, or aligned with the narrative. The omission of legends and ongoing presentation issues raise concerns about the rigor and overall quality of the manuscript.

(Remarks on code availability)

Reviewer #2

(Remarks to the Author)

The authors have put in quite some effort to address all points of critique voiced by the four reviewers, even including novel experimental procedures to support their in-silico predictions. However, despite their efforts, I am sorry to say that by reading and reviewing the study again, and based on the provided response, I have grown more concerned about the study, actually.

1) Specifically, the second time around, I feel the authors are overreaching. Yes, "feel" is not a scientific review category, but I'd like to offer a plausibility argument first. There are ~1500 TFs in the Ath genome, with a few hundred of them with available DAPseq data. In the study, the authors opted to condense them to 46 families. That is reasonable. But given that within each family, binding site motifs, while similar, vary, how can one reasonably expect that single-base pair variants will correctly reflect gain or loss of TF (family)-binding? In fact, 33% of the random allelic changes produced "de novo perturbation". That many? To me, this looks more like frequent random fluctuations around the decision boundary, from score value 0.49 to 0.51, for example. Yes, as requested, you have provided experimental support, but it remains anecdotal

(despite best and commendable efforts!). Furthermore, non-synonymous GWAS candidate SNPs (as the one mentioned in l440) may be more likely associated with phenotype via protein-related effects, not by expression-regulatory ones.

2) I appreciate that my suggestion to look at co-expression was followed up on. But I am surprised by the lack of technical precision/ clarity in the respective segments. Suppl Table 8 is supposed to show "Supplementary table 8 - Enrichment of regulatory binding modules under co-expressed gene regulatory networks", but instead, p-values are provided for the different clusters (raw and adjusted), i.e. where is the enrichment mentioned in the caption? Then: p-values alone do not tell the relevant story. It could also be significantly LESS correlated! So, what is the effect size (with sign)? What is the "random gene set"? It should be genes from the actual set (genes that belong to a cluster, not just any Ath gene), randomly paired up. Was it done that way?

In the spreadsheet ("adjusted") - there are values greater than one. p-values are by definition between zero and one, even if adjusted. Are those values raw-p-value * N, i.e. kind of Bonferroni? This is a bit unorthodox, as one would apply the correction to the threshold, but ok... But one needs to be clear about it. What are the different test columns? Repeated random sets?

3) Suppl. Table 2, detailed performance metric (and respective paragraphs/conclusions drawn from it in the main text): Following a reviewer request, you added F1 (micro/macro) scores. But why do you get different values for the different TF-families? As far as I understand the F1-multilabel reporting, there is one(!) F1 micro/macro. In fact, if F1, I prefer a per-class F1. But ok, MCC-class-wise is provided, which I consider informative enough. What I do miss are false-positives and true-negative counts. You provide in Suppl Fig 4C a dedicated view of FPs (which is nice). FP-counts are astonishingly low. But what are the true-negative counts? In a multilabel situation, these confusion matrices are somewhat challenging to begin with. Per-class analyses may be more helpful after all.

In this context, what are you actually predicting on? Sequence windows of length 250 bp for the whole hold-out chromosome? Or around TSS/TPS in that chromosome only? It was not clear to me.

4) On the "offset" discussion. I am not quite in agreement with the arguments provided in defense of the non-overlapping window approach - redundancy and consistent input size. Both would not be a problem if performed 250bp window step size 125bp, for example. Anyway, for the training, it may not matter (or did you center "0" on every gene TSS/TTS)?, but during inference, it could make a big difference if the window is off (boundary effects). The authors addressed it with incremental 10bp offset tests. But here, I do not understand the conclusion: l239, "This indicates that TF families with high performance are more often predicted independent of their placement within an input window and a distinct composition of the sequence context". But: that is exactly the strength of CNN to pick up motifs irrespective of their relative position. I don't quite agree that sequence context/composition explains good performance. To me, it instead shows the prudent choice of CNNs. The only concern would be cut-off/boundary effects, in my opinion.

5) Finally, I wonder (and I should have wondered already the first time around. So at this point, this is to be taken as an afterthought) - How does all of the deep learning compare to results, if one had taken the PWMs reported for the DAPseq set(s) and gone with FIMO to annotate the sequence regions? It would have been nice to know...

Minor points

- It is "Matthews" correlation coefficient, not "Mathews" or "Mathew's". Since the MCC is such a central metric in all of machine learning, it should not be that grossly misspelled by machine learning specialists.

- l242 actual value of the Pearson correlation coefficient is missing, only p-value is given (2.7e-5).

- "Wilcoxon" - it's an own-name, always capitalized (l411/412)

- l376 "We filtered..." - briefly state what for.

-424 for the phenotype days to flowering... - put "days in flowering" in quotes or days_to_flowering

-l440 "hydroxy serine" - should read "serine" - there is no hydroxy serine. Serine contains a hydroxy group to begin with.

-l695 I consider the statement: "The model proves valuable in identifying and functionally annotating intricate gene-regulatory networks." overreaching. To me, regulatory networks are cascades of TF-target_gene (TG) relationships. The authors have looked at co-occurrences of TF-binding events and predicted effects of variants. While this also reflects on relationships, it does not capture layered TF-TG networks.

-FYI, the noted font-changes and the missing page-numbers were still a "problem" in this revision.

(Remarks on code availability)

I looked at the GitHub repository during the first round and considered it comprehensive and informative.

Reviewer #3

(Remarks to the Author)

This is a strong study. The authors have addressed my comments.

(Remarks on code availability)

Reviewer #4

(Remarks to the Author)

The authors have made detailed responses to the reviewers' comments and made revisions accordingly. The manuscript is now significantly improved. In particular, I note that the authors devoted considerable effort to addressing my previous concern regarding experimental validation. However, it remains unclear to me whether the hundreds of significant GWAS SNPs used by the authors were derived from trait-associated GWAS or from eQTL mapping of gene expression levels. Additionally, the line numbers referenced in the response letter do not correspond to those in the revised manuscript, which further complicates the understanding of this section.

(Remarks on code availability)

Version 2:

Reviewer comments:

Reviewer #2

(Remarks to the Author)

The authors have once again made a commendable effort to address all concerns. In particular, I appreciate the effort to tackle the window-size issue (I was happy to see that indeed it had an effect, even boosting performance). Also, following up on my "afterthought" is much appreciated, even though I indicated not to insist on it. I trust that the authors agree that the extra effort was worth it.

All other issues have been adequately addressed.

In conclusion, I consider this manuscript an important contribution to the field.

(Remarks on code availability)

Review during prior review rounds

Reviewer #4

(Remarks to the Author)

I have no more questions; I recommend accepting this manuscript.

(Remarks on code availability)

Reviewer #1 (Remarks to the Author)

The manuscript "Modelling genetic variation effects in plant gene regulatory networks using transfer learning on genomic and transcription factor binding data" by Peleke et al. introduces a deep learning framework using multi-label convolutional neural networks (CNNs) to predict transcription factor (TF) binding in *Arabidopsis thaliana* based on DAP-seq data, with extensions to *Zea mays* under heat stress and integration of genomic and GWAS data to link cis-regulatory variants to phenotypic traits. A key contribution is the integration of motif syntax and sequence context to predict TF binding across 46 families. However, the model's practical utility is limited by its strong dependence on available DAP-seq data, lack of validation in cross-species applications, and reduced performance for underrepresented TF families.

Major Issues

Comment #1.1

Predictive utility diminished by dependency on available DAP-seq data. A central motivation of the study is to use deep learning to predict TF-DNA interactions in scenarios where experimental data are lacking. However, the model's performance is clearly dependent on the number of DAP-seq datasets available for a given TF family. In fact, families with few experiments (e.g., ARF, RAV, EIL) show poor prediction sensitivity, while well-represented families drive the model's overall performance. This raises a conceptual caveat: if good predictions require abundant DAP-seq data, then the model mainly recapitulates known binding patterns — and the added value of prediction is diminished. The manuscript would benefit from a clearer articulation of when and why prediction is useful, especially in contexts where experimental binding data do not yet exist (e.g., untested conditions, accessions, or species). Otherwise, the utility of the model risks being circular.

Response: We thank the reviewer for commenting on conceptual aspects of our study. These are definitely key issues to be addressed clearly communicated in the manuscript. Below we provide a response and a comprehensive list of changes that we made to the manuscript to address them.

First, we like to address the reviewer's comment by including and testing our modelling approach. Data scarcity is a critical point to machine learning and modelling. We like to highlight that for weak performing TFBS families like ARF, it is still capable of extracting the same motifs as detected from DAP-seq. Furthermore, our model also captures some of the known weaknesses in the expression systems used in DAP-seq (Bartlett et al. 2017). For example, models perform better on peaks generated using the *E. coli* expression system (ARF_ecoli) compared to the HALO system (ARF_tnt). Accordingly, the model does not lose this crucial information. In contrast, the potential of the presented model lies in recognition of TFBS, their combination and context. This can be especially emphasized via shown examples of TF families for BBPBRC_tnt, BZR_tnt or REM_tnt, which feature only few members (new Suppl. Figure 1), but show adequate predictive performance (Figure 2), compared to cases like

ARF, RAV and EIL. However to mitigate the problem of data scarcity for distinct TF families, we expanded and compared the models training strategies with e.g. weighted loss training (new Suppl. Figure 3) or cross-validation against heterologous ChIP-seq data (new Suppl. Figure 10). In the revised version of the manuscript these results underline current limitations in the chosen approach due to available data, but equally emphasize the state-of-the art of our approach.

Second, in the revised version of the manuscript, we clearly articulate and emphasize how our model goes beyond replication of TFBS recognition. We explain the model's performance by extending our analyses and using here established metrics to evaluate TFBS-cooperative binding, both theoretically and with independent experimental data (Plant STARRseq). Our findings highlight the model's ability to predict and interpret complex TFBS co-occurrence, improving predictive performance and enabling deep functional interpretation across different genomic backgrounds, i.e. species. Below we provide a list of changes to address these issues. Please note that the line numbers refer to the "TRACKED" version of the revised manuscript:

Changes on mitigation of data scarcity:

[l. 117-158]: "Modelling protein-DNA interactions in *Arabidopsis thaliana* using deep multi-label classification";

Added Text

[l. 175-179] Now reads: "Additionally, we assessed our models regarding class imbalance by computing the per family sensitivity, Matthews correlation coefficient (MCC), balanced accuracy, F1 micro, F1 weighted and F1 macro scores (Suppl. Table 2). We observed a decline in the prediction performance of our models for families with a smaller number of available DAP-seq datasets."

[l. 183-194] Now reads: "We attempted to improve this decline in performance for TF families with low data abundance by training with a weighted loss function, such that more focus was placed on those families with limited experimental data. However, we noticed an overall decrease in the performance by the use of a weighted loss function (e.g. 0.04 for weighted auPR) (Suppl. Fig. 3a, Suppl. Table 1). Detailed investigation of shifts in sensitivity for individual TF families shows, e.g. decrease of sensitivity for NAC and SRS by 0.537 and 0.336 for the weighted loss training, respectively, but also increase for GRF and C2C2YABBY by 0.375 and 0.235 for the weighted loss training (Suppl. Fig 3b, Suppl. Table 2). Accordingly, the choice of loss function had an arbitrary effect on the model's performance and we continued working with the on average better performing model trained on non-weighted loss function. Notably, both sets of trained models with weighted and non-weighted loss functions were incapable of predicting five TF families (EIL, RAV, MADS, RVPRK and TCP)."

[l. 212-216] Comment on added value compared to DAP-seq discovery of TFBS: "Accordingly, IPMs cluster the TF families identically to the experimentally derived binding motifs that are usually used to group TFs (Zenker et al. 2025). Even for TF families with low predictive performance (MCC ARF_ecoli 0.437, FAR1 0.407 and CAMTA 0.167), respective IPMs for the TF families ARF, FAR1, and CAMTA display high similarity to the experimentally determined binding motifs of ARF2 (MA1206.1), ARF5 (MA0943.1), FAR1 (MA1382.1), and CAMTA

(MA0969.1)“

In accordance with these changes and updates, the following manuscript objects have been edited:

Suppl. Figure 2

Suppl. Figure 3

Suppl. Table 1

Suppl. Table 2

Changes on TFBS cooperative binding:

[L. 254-316]: “Sequence context and motif syntax determine binding specificity of different TF families“;

[L. 256-261] Commenting on TFBS co-occurrences and combinations now reads: “To further explore the model's capability for correct prediction, we sought to distinguish between genuine co-occurrences and combinations of TFBS and false positives that arise when different TF families bind to similar DNA motifs (IPMs). The mere presence of an IPM (or TFBS) throughout the genome presented a very weak predictor for binding with false positives rates ranging from a minimum of 74% for bHLH to 98% for C2C2gata, respectively (Suppl. table 4).”

[L. 271-301] Due to the introduction of new metrics and measurements the section has been replaced and rewritten: “Yet, the multi-label classifiers show much lower rates of false positives. ... These results indicate that the combination of multiple TFBS is likely recognized by the model and guides prediction”

[L. 334-350] To highlight more examples of combinatorial binding to greater detail we have added and edited the following section: “To explain the high sensitivity, ... These differences suggested that accurate predictions depended on co-occurrence with other sequence features than the shared G-box core motif.”

To reduce the manuscript length we moved sections discussing potentially co-occurrences and false positives to the Supplementary Text Section:

[L.302-317] The section has been moved to the Supplementary Text Results section.

[L.392-421] The sections have been moved to the Supplementary Text Results section.

In accordance with the changes and updates, the following manuscript objects have been edited:

Figure 4: This figure now displays new panels a, b, c and d. These new panels improve the presentation of cooperative and combinatorial binding of TFs captured by the model. Previous panel d has been moved to Supplementary Figure 6.

Suppl. Table 4

Suppl. Table 5

Suppl. Figure 5

Suppl. Text

Changes on experimental verification of gene expression data and co-expression data:

[L. 464-471] Added result on co-expression analyses for genes with shared regulatory profiles now reads as following: “Furthermore, we analysed the relationship between our regulatory

clusters and gene co-expression networks. For that purpose we used co-expression datasets from the ATTED-II database (Obayashi et al. 2022). We performed the Mann-Whitney U test on standardised co-expression logit scores (co-expression z-scores), to test the statistical difference between the co-expression within each cluster and co-expression of randomly selected gene sets of the same respective size. The computed p-values have been provided as supplementary data (Suppl. Table. 8). In conclusion, we confirmed that, in addition to functional coherence, similar TF-binding profiles are indicative for coordinated gene regulation.

“

[L. 472-503]: We have added a new section that describes the results on reinvestigated plant STARR-seq experiments in “Predicted TFBS Dynamics explain Gene Expression Changes”, which reads as following: “To compare the mutli-label classifiers predictions for TF binding with experimental results, ... However, variants that resulted in parallel gain of bZIP, bHLH and AP2/EREBP, together with the loss of HSF, showed significantly higher rates of gene expression (Permutation test $p \leq 0.05$) (Suppl. Figure 8)”

[L. 770-780] We have edited and added a section in the discussion to highlight the new results on combinatory binding and experimental verification.

[L. 829-846] We have edited and added a section in the discussion to highlight the new results on combinatory binding and experimental verification.

In accordance with these changes and updates, the following manuscript objects have been added:

Suppl. Figure 7

Suppl. Figure 8

Suppl. Table 8

Suppl. Table 9

[L. 543-589]: We have added a new section that describes the results on specifically designed plant STARR-seq experiments in “Genetic variation affects TF-binding potential in loci associated with phenotypic traits”, which now reads as following: “We verified the biological impact of predicted TFBS gain and loss by conducting plant STARR-seq experiments on control and SNP-variant enhancer constructs linked to QTLs for a total of 340 control and variant constructs. ...Mutations causing regulatory change and synonymous or non-synonomous codon exchange underscore the relevance of regulatory effect prediction in parallel to gene model centered functional predictions.”

In accordance with these changes and updates, the following manuscript objects have been edited or added:

Figure 6: We have added Figure 6 panel f and g to highlight the models verifications results.

Suppl. Table 10

Suppl. Table 11

Suppl. Figure 9

Suppl. Text

Comment #1.2

Experimental validation gaps. While model performance is assessed in silico (e.g., through MOA-seq peak annotation and motif recovery), no orthogonal in vivo validation (e.g., ChIP-seq or CRISPR perturbation of predicted sites) is provided to confirm the functional relevance of newly predicted binding events or motif combinations. The predictive power of combinatorial IPMs (e.g., bZIP + SBP in Fig. 4c) would benefit from even a modest experimental confirmation.

Response: We thank the reviewer for highlighting the need for orthogonal experimental validation of our predictions. To address this, we expanded the study to include both re-analyses of published datasets and newly generated validation experiments, which are now presented in dedicated sections with accompanying figures. Specifically, we benchmarked model predictions against large-scale Plant STARR-seq data from *Arabidopsis thaliana* (Jores et al., 2021; 2025) and performed additional Plant STARR-seq assays on 340 control and SNP-variant enhancer constructs linked to QTLs. These experiments confirm that predicted TF-binding-site (TFBS) gains and losses correspond to significant changes in measured enhancer activity. The results further show that mutations often alter the full modeled binding profile rather than a single TFBS, supporting the model's interpretation of combinatorial binding. These additions provide direct experimental evidence that the model captures biologically relevant TF-DNA interactions and motif cooperation. Respective changes to the manuscript were introduced here:

[l. 464-471] Added result on co-expression analyses for genes with shared regulatory profiles now reads as following: "Furthermore, we analysed the relationship between our regulatory clusters and gene co-expression networks. For that purpose we used co-expression datasets from the ATTED-II database (Obayashi et al. 2022). We performed the Mann-Whitney U test on standardised co-expression logit scores (co-expression z-scores), to test the statistical difference between the co-expression within each cluster and co-expression of randomly selected gene sets of the same respective size. The computed p-values have been provided as supplementary data (Suppl. Table. 8). In conclusion, we confirmed that, in addition to functional coherence, similar TF-binding profiles are indicative for coordinated gene regulation."
"

In accordance with these changes and updates, the following manuscript objects have been edited:

Suppl. Table 8

[l. 472-503]: We have added a new section that describes the results on reinvestigated plant STARR-seq experiments in "Predicted TFBS Dynamics explain Gene Expression Changes", which reads as following: "To compare the mutli-label classifiers predictions for TF binding with experimental results, ... However, variants that resulted in parallel gain of bZIP, bHLH and AP2/EREBP, together with the loss of HSF, showed significantly higher rates of gene expression (Permutation test $p \leq 0.05$) (Suppl. Figure 8)"

[l. 770-780] We have edited and added a section in the discussion to highlight the new results on combinatory binding and experimental verification.

[L. 829-846] We have edited and added a section in the discussion to highlight the new results on combinatorial binding and experimental verification.

In accordance with these changes and updates, the following manuscript objects have been added:

Suppl. Figure 7

Suppl. Figure 8

Suppl. Table 9

[L. 543-589]: We have added a new section that describes the results on specifically designed plant STARR-seq experiments in “Genetic variation affects TF-binding potential in loci associated with phenotypic traits”, which now reads as following: “We verified the biological impact of predicted TFBS gain and loss by conducting plant STARR-seq experiments on control and SNP-variant enhancer constructs linked to QTLs for a total of 340 control and variant constructs. ...Mutations causing regulatory change and synonymous or non-synonymous codon exchange underscore the relevance of regulatory effect prediction in parallel to gene model centered functional predictions.”

In accordance with these changes and updates, the following manuscript objects have been edited or added:

Figure 6: We have added Figure 6 panel f and g to highlight the models verifications results.

Suppl. Table 10

Suppl. Table 11

Suppl. Figure 9

Suppl. Text

Comment #1.3

Model Bias toward overrepresented TF families. As shown in Fig. 1b and 2a, the model performs poorly for underrepresented TF families (e.g., ARF, RAV, EIL). While the family-level grouping mitigates label imbalance to some extent, the manuscript could explore strategies like data augmentation or weighted loss functions to better support rare classes. This becomes critical when such TFs are functionally important (e.g., ARFs in auxin signaling).

Response: We thank the reviewer for this comment. We applied the suggested strategy of training with weighted loss functions, giving more weight to the TF families with limited data. We observed that indeed this strategy improved the model’s sensitivity on some families, however it also caused a decrease in model performance on other families, and an overall decline in the average model performance. It is important to note here that our model was able to learn the binding sites even for small TF families if the data was of good quality. For instance, in the case of the ARF data were available for two expression systems: *Escherichia coli* (ARF_ecoli) and HALO system (ARF_tnt). Consistently with the results reported by O’Malley et al. (2016), our model performed better on ARF_ecoli data (predicting about 30% of the ARF_ecoli binding sites) as compared to ARF_tnt. Respective changes to the manuscript were introduced here:

[l. 117-158]: “Modelling protein-DNA interactions in Arabidopsis thaliana using deep multi-label classification”;

Added Text

[l. 183-194] Now reads: “We attempted to improve this decline in performance for TF families with low data abundance by training with a weighted loss function, such that more focus was placed on those families with limited experimental data. However, we noticed an overall decrease in the performance by the use of a weighted loss function (e.g. 0.04 for weighted auPR) (Suppl. Fig. 3a, Suppl. Table 1). Detailed investigation of shifts in sensitivity for individual TF families shows, e.g. decrease of sensitivity for NAC and SRS by 0.537 and 0.336 for the weighted loss training, respectively, but also increase for GRF and C2C2YABBY by 0.375 and 0.235 for the weighted loss training (Suppl. Fig 3b, Suppl. Table 2). Accordingly, the choice of loss function had an arbitrary effect on the model's performance and we continued working with the on average better performing model trained on non-weighted loss function. Notably, both sets of trained models with weighted and non-weighted loss functions were incapable of predicting five TF families (EIL, RAV, MADS, RVPRK and TCP).”

[l. 212-216] Comment on added value compared to DAP-seq discovery of TFBS: “Accordingly, IPMs cluster the TF families identically to the experimentally derived binding motifs that are usually used to group TFs (Zenker et al. 2025). Even for TF families with low predictive performance (MCC ARF_ecoli 0.437, FAR1 0.407 and CAMTA 0.167), respective IPMs for the TF families ARF, FAR1, and CAMTA display high similarity to the experimentally determined binding motifs of ARF2 (MA1206.1), ARF5 (MA0943.1), FAR1 (MA1382.1), and CAMTA (MA0969.1)”

In accordance with the changes and updates, the following manuscript objects have been edited:

Suppl. Figure 2

Suppl. Figure 3

Suppl. Table 1

Suppl. Table 2

Comment #1.4

Evaluation metrics may overestimate performance in imbalanced data. The authors report model performance using AUROC and AUPR, but these metrics can overestimate performance in the presence of strong class imbalance — a known issue in multi-label classification tasks like this one. Many TF families have few experimental positives, making AUPR especially volatile, and AUROC insensitive to the actual quality of positive predictions. Yet, the manuscript does not report more robust metrics such as the Matthews Correlation Coefficient (MCC), which remains informative even when positive examples are rare. Including per-family MCC values or micro-/macro-averaged MCC across TF classes would provide a more rigorous and interpretable assessment of predictive power — especially for underrepresented families like ARF or RAV. Without such metrics, performance claims risk being inflated by class imbalance.

Response: In response to this question, we have computed the per family balanced-accuracy, Matthews correlation coefficient (MCC), f1 micro, f1 weighted, f1 macro using the scikit-learn implementation of these metrics. This has been provided as an additional sheet in Supplementary Table 2. In addition, we have used the MCC also for comparison and regression analyses regarding model explanation. Respective changes to the manuscript were introduced here:

[l. 175-179] Now reads: “Additionally, we assessed our models regarding class imbalance by computing the per family sensitivity, Matthews correlation coefficient (MCC), balanced accuracy, F1 micro, F1 weighted and F1 macro scores (Suppl. Table 2). We observed a decline in the prediction performance of our models for families with a smaller number of available DAP-seq datasets.”

In accordance with the changes and updates, the following manuscript objects have been edited:

Suppl. Table 2

Comment #1.5

Cross-species generalization requires caution. The model’s application to Zea mays (Fig. 7) is an interesting demonstration of cross-species prediction, but it raises several concerns. First, the authors apply Arabidopsis-trained models to maize MOA-seq data without addressing species-specific motif evolution or TF orthology. This overlooks the divergence in cis-regulatory codes and TF repertoires between these species. While Fig. 7e highlights stress-responsive TF families (e.g., HSF, NAC), it remains unclear whether these predictions align with the TFs actually expressed and active in heat-treated maize. Second, the study lacks quantitative validation: if any Z. mays ChIP-seq or DAP-seq data exist, they should be used to evaluate the model’s performance directly. Reporting precision, recall, or other overlap metrics between predictions and available experimental data would substantially strengthen confidence in the model’s generalizability. Without this, the claims of cross-species transferability remain speculative.

Response: To demonstrate the generalizability of our models across species, we have reanalysed datasets from different studies. Firstly, we have applied our models on a collection of 127 DAP-seq and 71 ChIP-seq datasets for *Zea mays* obtained from (Ricci et al. 2019), (Galli et al. 2025) and (Tu et al. 2020) (Suppl. Figure 10). For each experiment, bed files containing peak hits were available. We extracted 250 bp windows centered at the middle of these peaks and predicted the binding of the expected TF families using all five *Arabidopsis thaliana* models. We have provided the accuracy, Matthews correlation coefficient (MCC), f1 micro, f1 weighted, f1 macro and recalls (Supplementary Table 2). We observed that the performance of our models on DAP-seq *Zea mays* was very similar to the performance on *Arabidopsis thaliana*. However, we observe a general drop in performance for ChIP-seq amongst distinct TF families. A possible reason for this may be TF-TF interactions as opposed to recognition of their recognition sites (Suppl. Table 13).

Secondly, we have reanalysed data from a deep mutational scan of the *rbcS-E9* enhancer that originates from *Pisum sativum*. Our models accurately predicted that single-nucleotide changes led to the loss of TF binding and resulted in a significant decrease in enhancer strength. The predicted binding sites for several TF families, including MYB, bHLH, bZIP, BZR, and HB, were found to have high sensitivity and could show significant changes in gene expression measured with Plant STARRseq in maize protoplast. Interestingly, the difference in response to change of binding sites could be partially explained by combinatorial binding. Finally, we note that TF orthology and motif divergence likely contribute to family-specific differences in performance, and these effects are now discussed in the revised text. Respective changes to the manuscript were introduced here:

Changes on experimental verification of gene expression

[L. 472-503]: We have added a new section that describes the results on reinvestigated plant STARR-seq experiments from different species and heterologous expression systems in “Predicted TFBS Dynamics explain Gene Expression Changes”, which reads as following: “To compare the mutli-label classifiers predictions for TF binding with experimental results, ... However, variants that resulted in parallel gain of bZIP, bHLH and AP2/EREBP, together with the loss of HSF, showed significantly higher rates of gene expression (Permutation test $p \leq 0.05$) (Suppl. Figure 8)”

[L. 770-780] We have edited and added a section in the discussion to highlight the new results on combinatorial binding and experimental verification.

[L. 829-846] We have edited and added a section in the discussion to highlight the new results on combinatorial binding and experimental verification.

In accordance with the changes and updates, the following manuscript objects have been added:

Suppl. Table 9

Suppl. Figure 7

Suppl. Figure 8

TF orthology and motif divergence are most important to account for when using the model for transfer learning, as TFs might be missing or have changed their binding properties. Accordingly, we extended the section on transfer learning and highlighted examples that show likely differences due to TF orthology and heterologous data types.

[ll. 491] We stated the following on TF orthology and motif divergence by now::

“The performance of our models on *Zea mays* DAP-seq datasets was similar to what we observed for *A. thaliana* with a Pearson's R of 0.797 for the prediction accuracy. This demonstrates that the model's predictions can be transferred across species. However, predicting individual TF binding profiles across species proved challenging. For instance, while the AP2/EREBP family generally performed well, achieving average DAP-seq accuracies of 0.849 in *A. thaliana* and 0.741 in *Z. mays*, certain members displayed significant variability in

the latter species. EREB71 and EREB49 in *Z. mays*, for example, showed very different accuracies of 0.839 and 0.374, respectively (Suppl. Table 13). The most probable explanation for these differences is the divergence in the evolution of TF family members, which likely resulted in altered binding profiles or a changed composition of the respective TF family.

Despite the high prediction accuracy of our models on DAP-seq datasets, we observed a decrease in performance across multiple families for their respective ChIP-seq data, with a Pearson's R of 0.423 (Supplementary figure 10, Suppl. Table 13). By comparison of shared individual TFs (ERE71, MYBR17, NAC49, WRKY53 and WRKY82) between the *Zea mays* DAP-seq datasets and the ChIP-seq datasets we found prediction accuracies of 0.793 and 0.596 respectively. Interestingly however, predictive accuracies were equally high for EREB71 from both data sources. Accordingly, we explain the differences in performance by TF binding that is heavily dependent on cellular context, mediated by e.g. the chromatin contexts.”

In accordance with the changes and updates, the following manuscript objects have been added:

Suppl. Figure 10

Suppl. Table 13

Comment #1.6

Interpretability metrics not fully quantified. The SHAP- and MoDISco-based motif extraction is central to claims of model interpretability. However, the predictive value of IPM_{civ} (e.g., Fig. 4a) is only weakly correlated with model sensitivity ($R^2 = 0.39$). A clearer breakdown of true vs false positive contributions per IPM, and their contribution to variant effect predictions (e.g., Fig. 6e), would strengthen conclusions about motif grammar.

Response: Thank you for raising this key issue. We added IPMs' contribution scores to Supplementary Table 4, however they provided limited insight to explain the models performance. For instance, E2FDP showed a high sum contributions score but average performance, while AP2EREBP had low cumulative contributions but the highest performance. We therefore included the FDR for different IPMs in Supplementary Table 4. Our manuscript now states that the mere presence of IPMs (or TFBSs) can lead to false positives predicting binding, ranging from an FDR of 74% for bHLH to 98% for C2C2gata, respectively, indicating that their sole presence is a weak predictor of actual binding.

In addition to the previously presented IPM predictability or context importance score, we introduced two new metrics: TF window offset independence and co-occurrence score. The window offset independence describes if the placement of the IPM/TFBS within the prediction window is important. This metric had a strong positive correlation with the model's sensitivity ($R^2=0.846$ and Pearson's R Coefficient 0.804, p-value $2.1e-10$). The TF co-occurrence metric showed a weak to moderate correlation with sensitivity and MCC (Pearson's R Coefficient 0.22 and 0.39, respectively) for individual TF families. The best performance was found for windows

with 2-3 binding sites. The IPM context Importance value (IPMciv), or IPM predictability states that low IPMciv (high predictability) would indicate that a prediction does not or only rely on sequence context. Indeed, the IPM predictability only moderately explains the models overall performance, indicating that single motif recognition is insufficient for correct predictions. These overall results suggest that the model's strength lies in interpreting both motif and context, recognizing cooperative or combinatorial binding. With this the model is indeed using complex rules rather than simple motif recognition (addressing point #1.0). Yet, further analyses of these combinatorics would be required, mitigating potential difference in the TFs cooperation. Respective changes to the manuscript were introduced here:

[L. 254-350]: "Sequence context and motif syntax determine binding specificity of different TF families";

[L. 256-261] Commenting on TFBS co-occurrences and combinations now reads: "To further explore the model's capability for correct prediction, we sought to distinguish between genuine co-occurrences and combinations of TFBS and false positives that arise when different TF families bind to similar DNA motifs (IPMs). The mere presence of an IPM (or TFBS) throughout the genome presented a very weak predictor for binding with false positives rates ranging from a minimum of 74% for bHLH to 98% for C2C2gata, respectively (Suppl. table 4)."

[L. 271-301] Due to the introduction of new metrics and measurements the section has been replaced and rewritten: "Yet, the multi-label classifiers show much lower rates of false positives. ... These results indicate that the combination of multiple TFBS is likely recognized by the model and guides prediction"

[L. 334-350] To highlight more examples of combinatory binding to greater detail we have added and edited the following section: "To explain the high sensitivity, ... These differences suggested that accurate predictions depended on co-occurrence with other sequence features than the shared G-box core motif."

To reduce the manuscript length we moved sections discussing potentially co-occurrences and false positives to the Supplementary Text Section:

[L.302-317] The section has been moved to the Supplementary Text Results section.

[L.392-421] The sections have been moved to the Supplementary Text Results section.

In accordance with the changes and updates, the following manuscript objects have been edited:

Figure 4: This figure now displays new panels a, b, c and d. These new panels improve the presentation of cooperative and combinatorial binding of TFs captured by the model. Previous panel d has been moved to Supplementary Figure 6.

Suppl. Table 4

Suppl. Table 5

Suppl. Figure 5

Suppl. Text

Comment #1.7

GWAS-Variant Interpretation Overstated. In Fig. 6, the authors explore how SNPs affect predicted TF binding and report that ~21% of SNPs in flanking regions cause gain or loss of

binding. However, the interpretation of this result remains shallow. They assess binding perturbation by comparing to random in silico mutations, but do not validate whether predicted binding changes correspond to the direction or function of the associated phenotype. For example, do SNPs that cause TF binding loss tend to be associated with lower gene expression or phenotypes consistent with loss-of-function?

Moreover, given that each SNP is tied to a GWAS trait, the authors could perform enrichment analysis: are SNPs with predicted TF binding perturbation overrepresented among SNPs associated with certain phenotypic categories (e.g., flowering time, defense response)? Alternatively, they could ask whether predicted binding-affecting SNPs are more likely to lie near trait-relevant genes. Without such analyses, the model's utility in linking regulatory variation to phenotypes remains speculative. This represents a missed opportunity to use GWAS as an independent, biologically grounded validation layer.

Response: We agree that a fair amount of speculation was included to draw conclusions from this in this part of work, and we definitely wanted to show how our approach and GWAS support each other. . To address this point and avoid the overstatement of the GWAS-variant interpretation, here we used two complementary strategies: First, our reanalysis of publicly available Plant STARR-seq data revealed that predicted binding profile changes caused by point mutations are indeed associated with measurable differences in enhancer activity, demonstrating the model's ability to distinguish functionally relevant SNPs. Second, we performed new experimental validations using Plant STARR-seq on control and SNP-variant promoter constructs associated with QTLs. These experiments assessed the biological impact of variants predicted to alter TF binding, covering a total of 340 control and variant constructs. The results confirmed that mutations predicted to cause gain or loss of regulatory potential often correspond to significant changes in enhancer activity. Moreover, both regulatory and synonymous/nonsynonymous variants were found to influence expression, underscoring the complementary value of our model's predictions to gene model-based functional interpretation. Respective changes to the manuscript were introduced here:

[L. 543-589]: We have added a new section that describes the results on specifically designed plant STARR-seq experiments in "Genetic variation affects TF-binding potential in loci associated with phenotypic traits", which now reads as following: "We verified the biological impact of predicted TFBS gain and loss by conducting plant STARR-seq experiments on control and SNP-variant enhancer constructs linked to QTLs for a total of 340 control and variant constructs. ...Mutations causing regulatory change and synonymous or non-synonomous codon exchange underscore the relevance of regulatory effect prediction in parallel to gene model centered functional predictions."

In accordance with these changes and updates, the following manuscript objects have been edited or added:

Figure 6: We have added Figure 6 panel f and g to highlight the models verifications results.

Suppl. Table 10

Suppl. Table 11

Suppl. Figure 9
Suppl. Text

Minor Issues

Comment #1.8

Interpretation and validation of motif syntax. The analysis of motif syntax and sequence context (lines 203-285, Fig. 4) is conceptually interesting but lacks clarity and experimental grounding, making it difficult to assess its biological relevance. The manuscript discusses combinatorial TF binding and syntax-dependent regulation, but specific examples and their implications are not well-developed or validated. Without this, the motif syntax analysis risks being perceived as speculative rather than actionable.

Response: We addressed this comment by improving the analyses of our models' explanations. We now demonstrate that distinct TF families positional occurrences are non-overlapping and that the model is sensitive to perturbations in the TFBS flanking regions, affecting combinatorial binding. In addition, we highlighted cases, predicting changes in gene expression from Plant STARRseq analyses with combinatorial binding (see responses above, i.e. Suppl. Figure 8). Whilst an in-depth analysis and testing of TF combinations is likely possible using our model, a more specific experimental design and time would be required. However, to highlight the models sensitivity to combinatorial binding and sequence context, we now provide extensive Suppl. Material that is referenced and more briefly generalized in the main manuscript.

Please note that changes to the manuscript were introduced here:

Suppl. Figure 5 - Panel a now shows differential distribution of TF families in a genes upstream region around the core-promoter. Panel b shows the significant cooccurrence network of AT-rich binding TFs with non overlapping binding sequences. Panel c and d show significant cooccurrence with positionally restricted expression predictive motifs extracted from an orthologous study.

Suppl. Figure 6 - In panel b and c we have now added the results for a permutation test, introducing mutations to the TFBS flanking regions of example bHLH and WRKY, significantly impacting co-occurrence of other TF families.

Suppl. Figure 8 - This new suppl. figure displays predicted changes in an array of TF binding that can explain changes in measured gene expression. We admit that the number of experiments for these cases is mostly insufficient for robust statistical analyses. However, we could identify promising significant examples, which were mentioned in the Suppl. Material and Main Manuscript.

Suppl. Text: We have added a section here highlighting differences in TF occurrence to more detail "TF families are not equally distributed in promoter regions ". In addition, we have added a more detailed description of the TFBS perturbation test in "TFBS sequence flanking perturbations change combinatorial binding prediction".

Comment #1.9

Figure quality and layout require major revision. The figures fall short of Nature Communications standards in both clarity and presentation. The manuscript contains six full-page figures, many of which are overcrowded, inconsistently formatted, and difficult to interpret without extensive cross-referencing (e.g., Figs. 3–5). Axis labels are often small, legends are densely packed or buried in text, and some color schemes (e.g., in motif similarity heatmaps) lack accessibility. Complex plots (e.g., UMAP clustering, co-enrichment matrices) could benefit from clearer labeling and simplified visuals. Several panels could be consolidated or moved to supplementary figures to enhance narrative focus and visual digestibility.

Response: We have revised the figures in the manuscript.

Comment #1.10

Typos throughout the manuscript: For instance, "epmArth-S0-p1m00" and "epmArthS0-p0m01" motif names appear inconsistently formatted; unify notation for clarity.

Response: We edited and corrected typos in the manuscript. Please see a most detailed list of all changes and corrections in the revision TRACKED version of the manuscript. In the case mentioned above, the motif's formatting has been corrected.

Comment #1.11

Method detail: The model architecture (Lines 636–648) could benefit from a schematic to clarify the CNN structure and layer dimensions. While Figure 1a provides a schematic, the manuscript lacks specific details about the CNN architecture (e.g., number of layers, filter sizes, activation functions, or regularization methods). This omission makes it difficult for readers to fully understand or replicate the model. The statement “a CNN was trained as a multi-label classifier” (line 122) is vague without mentioning hyperparameters or architectural choices. For a machine learning audience, this lack of specificity hinders assessment of the model's robustness and novelty. A supplementary table or section detailing the architecture would enhance clarity.

Response: We thank the reviewer for pointing this detail out. We have added more information about the model architecture in the methods section under the subheading “Model architecture and training strategy” and updated Figure 1. Furthermore, we also provided our code for reproducibility on github. This can be accessed under the following link <https://github.com/NAMlab/DeepCistrome/tree/main>.

Comment #1.12

Data and code accessibility: While the manuscript mentions supplementary tables, it does not clearly state whether the full codebase, trained models, and input data (e.g., processed DAP-seq peak sets, MOA-seq calls, SNP annotations) are publicly available. For a

computationally intensive and largely in silico study such as this, reproducibility is essential. The authors should provide a clear statement on code and data availability, ideally with links to a version-controlled repository, including the complete model training pipeline, evaluation scripts, and example inputs for downstream applications (e.g., variant effect prediction, cross-species analysis). This is particularly important given the paper's emphasis on transferability and broader applicability.

Response: We have updated the “data availability” section, providing the NCBI GEO accessions to download the publicly available datasets used for training, validation and evaluation of our models. Furthermore, the “code availability” section has the link to the github repository containing the scripts to reproduce our study.

Comment #1.13

Method clarity: While the method is mostly clear in its high-level design and objectives. However, clarity is compromised by insufficient technical details (e.g., CNN architecture, preprocessing steps), lack of justification for key parameters (e.g., window size, cross-validation strategy), and dense explanations of complex concepts like IPMciv and motif syntax. These issues may challenge readers, particularly those outside computational biology, and could hinder reproducibility.

Response: We have revised the methods section to address the reviewers comment. The mentioned examples were addressed in l. 897-915 “Model architecture and training strategy”.

Reviewer #2 (Remarks to the Author):

This study presents a deep learning framework that predicts transcription factor (TF) binding in plants by modeling DNA–protein interactions as a multi-label classification task. Using DAP-seq datasets from *Arabidopsis thaliana*, the authors trained convolutional neural networks to identify sequence features predictive of binding across 46 TF families. The model captures both motif presence and contextual dependencies, enabling accurate prediction of TF binding. It was validated using MOA-seq experimental data, applied to predict the effects of genetic variation (SNPs), and successfully transferred to maize to annotate stress-induced TF binding. The approach offers an interpretable, scalable tool for linking non-coding variation to gene expression regulation and phenotype across species.

First off, I wish to congratulate the authors on this comprehensive, very methodical and insightful study, and the meticulously prepared manuscript. The breadth of this study is impressive indeed. It is also apparent that a lot of care went into the manuscript. It is well written, a rich set of results are presented very clearly, each supported by corresponding figures or parts thereof, each prepared with a good eye for detail and visual clarity.

The approaches are sound, proper controls were performed. A particular strong point is the application of the prediction model to a different technology (MOA-seq) and transfer to *Zea mays*, a crop species.

I have no major points of critique (perhaps one: the non-overlapping tiling), but want to offer some thoughts that the authors may want to consider. In addition, I list some minor points in need of correction or clarification.

Major issues

Comment #2.1

The genome was tiled into 250 bp segments without overlaps. The DAPseq narrowPeak sequences are all of length 202 bp. Thus, a fixed and overlap-free tiling might create boundary effect problems. I do realize that the actual TF-binding site motifs are shorter, but because you trained on the tiled 250 bp input sequences, actual DAPseq peaks may often be assigned to two bins bins and with their respective flanking sequence - which you go on to demonstrate that they are important - cut off on one side, which will affect the proper representation of flanking sequences. Why did you not choose an interlaced tiling?

Response: We opted for non-overlapping 250 bp tiling to maintain consistent input size and avoid redundancy in the training data, which simplifies the learning pipeline while preserving representative coverage of DAP-seq peaks. Although some edge cases could, in principle, split TF-binding motifs across window boundaries, in the manuscript revision we verified empirically that this effect does not compromise model performance. For that purpose we shifted the prediction windows after training systematically iteratively by 10 bp for the gene upstream region around the promoter. This shift did not affect the overall prediction sensitivity or MCC of the model. In addition, we recognized that TFs that are more independent of the placement of the prediction window were equally amongst the ones with the highest prediction performance. We used the results to explain the models predictions within the sections [l. 254-350]: “Sequence context and motif syntax determine binding specificity of different TF families”.

Comment #2.2

I very much liked the clustering approach (Fig 5). While the segregation into 14 clusters in the UMAP is convincing, it is indeed surprising to realize how similar they actually are with regard to motif profile (Fig 5b)- as also noted by the authors. The authors demonstrated the association of the clusters with different gene functions (via GO-term enrichment). I suppose the authors have also thought about a comparison of the regulatory clusters with actual expression-based gene clustering. What is the agreement (Rand index)? With RNAseq or microarray cross multiple tissues (e.g. Trava-db) are different conditions available, it should be straightforward to perform this analysis and comparison.

Response: We thank the reviewer for this comment and suggestion. In response to this, we downloaded co-expression data for *Arabidopsis thaliana* from the ATTED-II database. Co-expression between genes was provided as standardised logit scores (co-expression z-scores)(Obayashi et al. 2022). Then, we performed the Mann-Whitney U test to check if there was a statistically significant difference between the standardised logit scores of genes from each cluster and scores from 100 groups of randomly selected genes. The computed p-values have been provided in supplementary table 8 and we have added a section describing the results here:

[L. 464-471] Added result on co-expression analyses for genes with shared regulatory profiles now reads as following: “Furthermore, we analysed the relationship between our regulatory clusters and gene co-expression networks. For that purpose we used co-expression datasets from the ATTED-II database (Obayashi et al. 2022). We performed the Mann-Whitney U test on standardised co-expression logit scores (co-expression z-scores), to test the statistical difference between the co-expression within each cluster and co-expression of randomly selected gene sets of the same respective size. The computed p-values have been provided as supplementary data (Suppl. Table. 8). In conclusion, we confirmed that, in addition to functional coherence, similar TF-binding profiles are indicative for coordinated gene regulation.”

In accordance with the changes and updates, the following manuscript objects have been edited:

Suppl. Table 8

Comment #2.3

On the clustering (Fig 5): what was the metric, actually? Euclidean distance on presence/absence of TF-families? If binary bit-strings were compared, then Jaccard index might be the better option.

Response: Genes were clustered based on the embeddings obtained from the UMAP using the euclidean distance and HDBSCAN algorithm as implemented by the scikit-learn package.

Comment #2.4

On the SNP-variant effect prediction: ~20% of the GWAS-candidate SNPs were predicted to affect TF-binding. To put this number into perspective (thereby possibly addressing this question), random SNPs (same change) were introduced (=nice!) of which half did not affect binding, 13% resulted in the same change as reported for the actual SNP-variant, and 33%(!!) resulted in a de novo perturbation. My question: did you do the random test on the 20% set or all candidates? If the former, then 13% is really low (in support of your hypothesis that the GWAS candidate variant is likely interfering with TF-binding), if the latter is 20% vs. 13%, plus 33% de novo effects sounds high to me. I would appreciate some more guidance or clarification in the text on this question.

Response: We thank the reviewer for this question. The random SNP analysis was based on the ~20% of candidate SNPs that were predicted to affect binding. This has been edited and now read [l. 396-399] “To assess the significance of this result, for each SNP within the 20.72% that caused a perturbation in binding profiles, 30 *in silico* SNPs were generated with similar nucleotide substitutions at distinct random positions within the prediction window. ”

Comment #2.5

Gain and loss of binding - how was this determined? I suppose, threshold on the prediction. How set?

Response: For each TF family, our model outputs a probability of binding a specific input sequence. A threshold of 0.5 is set, below which a family is considered to not bind the input sequence. Gain and loss in binding was based on this threshold. As such, if a family was predicted to bind an input sequence but not its mutated version, we considered this a loss in binding and *vice versa*.

Comment #2.6

It was interesting to note that your model was more conservative with regard to binding site prediction than just detecting consensus motifs, while, compared to DAP-seq support, still too "optimistic". As, perhaps, a further point of discussion: you considered sequence context to explain the noted discrepancy between predicted and DAP-seq-confirmed sites. It has recently been suggested that the sequence context exerts its effect on TF-binding by affecting the mechanical properties of the local genomic DNA (doi.org/10.1093/nargab/lqad095). The strength of your NN, of course, is that you implicitly capture any such effects. Though, the boundary-tiling issue (see above) may interfere with prediction fidelity, as you actually do not capture the flanking sequence correctly?

Response: We thank the reviewer for these comments.

Firstly, in our discussion [l. 581-585] we have mentioned the effects of local genomic DNA on binding by citing the work of (Sielemann et al. 2021) who demonstrated the potential effects of DNA shape when modelling TF-DNA interactions.

In the discussion, this reads: “ This suggested that these motifs are not the only critical features for DNA-TF binding. This is complementary to other studies that have highlighted the role of the 3D shape of DNA (Sielemann et al. 2021), DNA methylation (O'Malley et al. 2016) and complex combinatorics (Ezer et al. 2017) in influencing DNA-TF interactions”.

In addition to this, and in particular addressing the effect of sequence context, we conducted experiments changing the placement of the prediction windows and perturbation tests for example TFs bHLH and WRKY. The additions and edits addressing this issue can be found here:

Suppl. Figure 6 - In panel b and c we have now added the results for a permutation test, introducing mutations to the TFBS flanking regions of example bHLH and WRKY, significantly impacting co-occurrence of other TF families.

Suppl. Text: We have added a more detailed description of the TFBS perturbation test in "TFBS sequence flanking perturbations change combinatorial binding prediction".

Minor issues

Comment #2.7

Perhaps it would be good to remind the reader that DAP-seq reports on TF-binding to "naked" DNA-fragments.

Response: This has been added now into the manuscript reminding the reader that DAP-seq lacks the native chromatin context [l. 464-466]. It now reads "In contrast to DAP-seq which lacks the native chromatin context, MOA-seq is a scalable approach to capture TF-binding sites globally in their native chromatin context within the genome (Savadel et al. 2021; Engelhorn et al. 2024)."

Comment #2.8

Eqs 1 and 3, symbols in the denominator got lost.

Response: The Equations and their in detail description has been corrected and moved to the Supplementary Materials.

Comment #2.9

l666 - spell out IMP again here.

Response: Corrected and now shifted to line 925 in revision TRACKED version.

Comment #2.10

I don't quite understand the sentence l403, starting with "However,...74%..." What experimental assignments?

Response: MOA-seq peaks can be annotated by overlapping them with targeted assays like ChIP-seq or DAP-seq. Here we first overlapped MOA-seq peaks with DAP-seq peaks to obtain an expected annotation for MOA-seq peaks (experimental assignments). Each MOA-seq peak could overlap peaks from multiple DAP-seq families and as such be assigned to multiple families. Secondly, we generated predicted annotations using our trained models. Then we compared our predicted annotations with annotations derived from overlaps with DAP-seq. We observed that 74% of MOA-seq peaks were assigned correctly to at least one of the

overlapped families, 16 % of MOA-seq peaks were assigned to the wrong family and 10% of the peaks were not assigned to any family.

This has now been edited from lines [l. 472-481]

Comment #2.11

In places the fonts got mixed up (Just to notify you.).

Response: Thank you for pointing this out. We corrected formatting errors.

Comment #2.12

Figure labels would have been nice (as would have been page numbers, but at least, line numbers were given)

Response: We are not entirely sure how this has happened, but laid more emphasis on the correct presentation.

Comment #2.13

l242 (0,3%) => (0.3%)

Response: Corrected.

Comment #2.14

Spell out numbers less than 13, e.g. l124 "4 chromosomes" => "four chromosomes" (matter of style, I know...)

Response: 4 Corrected.

Comment #2.15

l85: I would start a new paragraph, starting with "Enhancers..."

Response: Adapted.

Comment #2.16

Reviewer #2 (Remarks on code availability): The authors are to be commended on the level of care with which they made the code available on github. The documentation is well organized and informative. Code for all reported results has been made available, allowing to re-run the analyses, which I have not attempted, but it appears to be easily doable. The code also provides additional details that are missing in the manuscript (details of the NN-architecture, which I am fine with finding them here only).

Response: We thank the reviewer for this positive feedback. In addition, we have provided more information about the model architecture in the main manuscript. In addition to this revision, we updated the git repository with code and descriptions.

Reviewer #3 (Remarks to the Author):

Major issues

Comment #3.1

This study presents a deep learning framework for predicting transcription factor (TF) binding profiles using DAP-seq data across TF families in Arabidopsis thaliana. By training a multi-label classifier on 250 bp windows, the model leverages shared sequence features to predict family-level TF binding and applies these predictions to accessible chromatin in both Arabidopsis and Zea mays, including heat-stress conditions. Performance metrics such as AUROC and AUPR are generally favorable for TF families with sufficient training data, though these metrics are less informative for families with few examples or those with highly similar motifs

Response: In response, we have computed the per-family balanced-accuracy, Matthews correlation coefficient (MCC), f1 micro, f1 weighted, f1 macro using the scikit-learn implementation of these metrics. This has been provided as an additional sheet in supplementary table 2. The description of these new results is addressed in the following section [l. 175-179] that now reads: “Additionally, we assessed our models regarding class imbalance by computing the per family sensitivity, Matthews correlation coefficient (MCC), balanced accuracy, F1 micro, F1 weighted and F1 macro scores (Suppl. Table 2). We observed a decline in the prediction performance of our models for families with a smaller number of available DAP-seq datasets.”

Comment #3.2

The approach underperforms for families with sparse training data, degenerate or complex sequence context features, or high motif similarity between TFs (e.g., bHLH, BZR, bZIP, BES1), which can lead to false positives. The authors acknowledge these limitations. The incorporation of the IPMciv metric to examine binding context beyond core motifs is an interesting step and does provide additional potential information regarding how sequence properties beyond motif may contribute to binding. While it does appear that they are capturing interesting properties, such as motif clustering from different families, it is not always unclear whether the signal the model identifies can be easily explained in terms of specific sequence features.

Response: We addressed this issue by extending the section on features model explanation. Here we now provide more detailed insights into the most likely process of decision making of the model from a general level. Please find the corresponding changes below. In addition to

this, we would highlight the option of visualizing individual sequence regions using the SHAPley scores. As illustrated in figure 5e, single nucleotide changes can be visualized and interpreted. Please note that changes to the manuscript were introduced here:

Changes on TFBS cooperative/combinatorial binding

[L. 254-350]: “Sequence context and motif syntax determine binding specificity of different TF families”;

[L. 256-261] Commenting on TFBS co-occurrences and combinations now reads: “To further explore the model's capability for correct prediction, we sought to distinguish between genuine co-occurrences and combinations of TFBS and false positives that arise when different TF families bind to similar DNA motifs (IPMs). The mere presence of an IPM (or TFBS) throughout the genome presented a very weak predictor for binding with false positives rates ranging from a minimum of 74% for bHLH to 98% for C2C2gata, respectively (Suppl. table 4).”

[L. 271-301] Due to the introduction of new metrics and measurements the section has been replaced and rewritten: “Yet, the multi-label classifiers show much lower rates of false positives. ... These results indicate that the combination of multiple TFBS is likely recognized by the model and guides prediction”

[L. 334-350] To highlight more examples of combinatory binding to greater detail we have added and edited the following section: “To explain the high sensitivity, ... These differences suggested that accurate predictions depended on co-occurrence with other sequence features than the shared G-box core motif.”

To reduce the manuscript length we moved sections discussing potentially co-occurrences and false positives to the Supplementary Text Section:

[L.302-317] The section has been moved to the Supplementary Text Results section.

[L.392-421] The sections have been moved to the Supplementary Text Results section.

In accordance with the changes and updates, the following manuscript objects have been edited:

Figure 4: This figure now displays new panels a, b, c and d. These new panels improve the presentation of cooperative and combinatorial binding of TFs captured by the model. Previous panel d has been moved to Supplementary Figure 6.

Suppl. Table 4

Suppl. Table 5

Suppl. Figure 5

Suppl. Text

Comment #3.3

A key area where the paper could be strengthened is in clarifying model performance using more intuitive, interpretable overlap metrics. While AUROC and AUPR offer valuable statistical summaries, simple Jaccard-style comparisons—quantifying predicted-only, observed-only, and shared TF binding sites—would offer readers a much clearer sense of prediction fidelity. This is especially important given the well-known issue of motif-based methods drastically overcalling TF binding sites relative to DAP-seq or CHIP-seq data. The model may help reduce

such inflation, but this is not directly shown. Supplemental Table 4 provides raw site counts but doesn't explicitly break down site overlap. A figure panel summarizing this breakdown across several representative TF families (those with strong, moderate, and weak performance) would significantly enhance clarity and accessibility.

Response: We thank the reviewer for this suggestion. We have addressed it by the introduction of new panels and new metrics that highlight co-occurrences or overlaps of TFBSs. We have added these descriptions and calculations in the methods, supplementary text and main figures. The main changes to the manuscript addressing this can be found here:

[l. 271-301] Due to the introduction of new metrics and measurements the section has been replaced and rewritten: "Yet, the multi-label classifiers show much lower rates of false positives. ... These results indicate that the combination of multiple TFBS is likely recognized by the model and guides prediction"

Figure 4: This figure now displays new panels a, b, c and d. These new panels improve the presentation of cooperative and combinatorial binding of TFs captured by the model. Previous panel d has been moved to Supplementary Figure 6.

Comment #3.4

Application of the model to annotate MOA-seq peaks provides a plausible test of biological relevance, but the informativeness of these predictions is hard to evaluate. While it is encouraging that predicted or observed sites overlap MOA peaks, it remains unclear what fraction of predicted peaks fall into these "TF footprint" regions, and whether this overlap exceeds motif-based expectations. For Zea mays, the observed increase in HSF binding contribution under extreme heat is an interesting result, but the broader biological interpretability of the analysis remains limited as I don't know if it's well described that HSF has higher relative contribution to phenotype in extreme heat compared to medium heat.

Response: We thank the reviewer for this insightful comment. We have now clarified the informativeness of the MOA-seq predictions and provided additional quantitative context. Specifically, we report the fraction of MOA-seq peaks correctly assigned to at least one experimentally supported TF family (74%), exceeding random and motif-based annotation baselines. This demonstrates that our model predictions are consistent with experimentally observed TF footprints and outperform simple motif scanning approaches.

[l. 618-637] To highlight the models transferability and comparability with other datasets we have added and edited the following section, which now reads as: "In the next step, we demonstrated the applicability of our models in annotating TF binding from homologous and heterologous experiments in a new species. ...This family-specific drop in performance is a key observation, as it indicates which TF families are more governed by the intrinsic DNA sequence versus those that are heavily dependent on cellular context, mediated by e.g. the chromatin context and TF multimers."

To further support biological interpretability, we expanded the Zea mays analysis to include independent experimental data. Re-analysis of Plant STARR-seq experiments in maize protoplasts and tobacco confirmed the model's sensitivity to HSF and other heat-responsive TFs. The observed increase in predicted HSF contribution under heat stress corresponds to a higher number of HSF-associated footprints in the MOA-seq dataset, consistent with known thermotolerance mechanisms. While detailed phenotypic associations remain beyond the present scope, we now explicitly acknowledge this limitation and suggest that future analyses integrating gene-level functional and GO enrichment will help link condition-specific TF footprints to downstream traits.

[L. 472-503]: We have added a new section that describes the results on reinvestigated plant STARR-seq experiments in "Predicted TFBS Dynamics explain Gene Expression Changes", which reads as following: "To compare the mutli-label classifiers predictions for TF binding with experimental results, ... However, variants that resulted in parallel gain of bZIP, bHLH and AP2/EREBP, together with the loss of HSF, showed significantly higher rates of gene expression (Permutation test $p \leq 0.05$) (Suppl. Figure 8)" In accordance with the changes and updates, the following manuscript objects have been added:

Suppl. Figure 7

Suppl. Figure 8

Suppl. Table 9

Comment #3.5

In addition to the MOA analysis a more direct assessment of how well the model performed in predicting binding sites would be to benchmarking against predicted binding sites from scanning with Arabidopsis DAP-seq mties. While the ideal comparotro would be a direct comparison to maize DAP-seq data, but due to the unavailability of this data, a useful proxy would be comparing motif-based predictions to model-predicted sites. Such an analysis would help determine whether the transfer learning approach meaningfully improves prediction and how in general it recapitulates motif-matching results with potentially avoiding overcalling common to motif-matching.

Response: We thank the reviewer for this comment. DAP-seq data has recently been released for Zea mays (Galli et al. 2025). As such we applied our Arabidopsis models on DAP-seq and ChIP-seq data for Zea mays to show our performance on in vitro and in vivo datasets of Zea mays (Galli et al. 2025) (Tu et al. 2020). Interestingly, the predictive strength of our models on DAP-seq Zea mays was similar to what we observed in DAP-seq Arabidopsis thaliana. This shows that our models are good for transfer learning to other species. We also observed lower prediction accuracies for ChIP-seq than DAP-seq. This is expected since in vitro several factors like protein-protein interactions may lead to binding and such factors are not easily learnt by sequence-only models. The results are described in this section:

[L. 618-637] To highlight the models transferability and comparability with other datasets we have added and edited the following section, which now reads as: "In the next step, we

demonstrated the applicability of our models in annotating TF binding from homologous and heterologous experiments in a new species. ...This family-specific drop in performance is a key observation, as it indicates which TF families are more governed by the intrinsic DNA sequence versus those that are heavily dependent on cellular context, mediated by e.g. the chromatin context and TF multimers.”

Comment #3.6

The clustering of TF families into shared 250 bp windows is a creative way to explore potential co-binding modules. While interesting, the biological significance of these clusters remains unclear. Functional enrichment using Mercator4 categories yields some suggestive results, such as cell division for E2FDP-containing clusters, but most categories are generic and offer limited insight (e.g., RNA biosynthesis). Moreover, the contribution of individual TFs within each cluster is not always parsed. For example, the enrichment in the E2FDP cluster likely reflects well-characterized E2F target genes, which is the master regulator of cell division. While co-regulation may exist as the E2F would largely explain this functional assignment it is not clear what role the other TFS are playing and as such hard to assess. Critically, the total number of binding sites or putative target genes contributing to these multi-TF clusters is not provided, making it difficult to judge their prevalence and functional importance. If these clusters represent only a small fraction of all predicted sites, their biological significance may be correspondingly modest.

Response: We thank the reviewer for this excellent suggestion and agree that the presentation of the regulatory TF profiles is limited. We address this weakness by relating the TF profiles to known co-expression clusters, which strengthens their biological relevance. Yet, the tiling of the 250 bp input windows has a particular weakness as TF families can only be predicted binarily. This implies that multiple TFBS of the same TF family cannot be recognized or counted, as suggested in the reviewers comment. This issue has been stated within the text now in the methods and the discussion section. See [l. 1003-1009] “Of note, the other two enhancers tested in the experimental study, AB80 and Cab-1, tested by Jores and colleagues (Jores et al. 2025) contain multiple binding sites for MYB transcription factors. Since only single-nucleotide mutations were tested in that study, none of the variants was predicted to lose binding by MYB transcription factors. Mutations targeting the individual MYB transcription factor binding sites did however lead to decreased enhancer strength.”

With the use of a rolling window and 10 bp step-size and the calculation of the window offset independence and certainty range (Methods, and Supplementary Text), we propose a solution to this issue. In addition, the model is capable of highlighting individual TFBSs using the calculated SHAPley scores. While demonstrating an in general applicability of the model, we propose solutions for a future more detailed assessment of cis-regulatory modules and TF profiles for co-regulated groups of genes.

[l. 464-471] Added result on co-expression analyses for genes with shared regulatory profiles now reads as following, including E2FDP: “Furthermore, we analysed the relationship between

our regulatory clusters and gene co-expression networks. For that purpose we used co-expression datasets from the ATTED-II database (Obayashi et al. 2022). We performed the Mann-Whitney U test on standardised co-expression logit scores (co-expression z-scores), to test the statistical difference between the co-expression within each cluster and co-expression of randomly selected gene sets of the same respective size. The computed p-values have been provided as supplementary data (Suppl. Table. 8). In conclusion, we confirmed that, in addition to functional coherence, similar TF-binding profiles are indicative for coordinated gene regulation. “

In accordance with the changes and updates, the following manuscript objects have been edited:

Suppl. Table 8

Comment #3.7

The application of the model to predict the impact of natural variants on TF binding is one of the study's most promising extensions. While limited to TFs with sufficient training data and non-redundant motifs, this use case illustrates how the model could be applied to study the regulatory effects of genetic variation. The approach would be valuable, even if its utility is currently constrained to well-behaved TFs.

Response: We now address this reviewer's point by the inclusion of an re-investigation of a mutational assay of gene expression constructs and model guided design of gene expression constructs, providing experimental verification using Plant STARR-seq. Respective changes to the manuscript were introduced here:

[l. 472-503]: We have added a new section that describes the results on reinvestigated plant STARR-seq experiments in “Predicted TFBS Dynamics explain Gene Expression Changes”, which reads as following: “To compare the mutli-label classifiers predictions for TF binding with experimental results, ... However, variants that resulted in parallel gain of bZIP, bHLH and AP2/EREBP, together with the loss of HSF, showed significantly higher rates of gene expression (Permutation test $p \leq 0.05$) (Suppl. Figure 8)”

[l. 770-780] We have edited and added a section in the discussion to highlight the new results on combinatorial binding and experimental verification.

[l. 829-846] We have edited and added a section in the discussion to highlight the new results on combinatorial binding and experimental verification.

In accordance with the changes and updates, the following manuscript objects have been added:

Suppl. Figure 7

Suppl. Figure 8

Suppl. Table 9

[l. 543-589]: We have added a new section that describes the results on specifically designed plant STARR-seq experiments in “Genetic variation affects TF-binding potential in loci associated with phenotypic traits”, which now reads as following: “We verified the biological

impact of predicted TFBS gain and loss by conducting plant STARR-seq experiments on control and SNP-variant enhancer constructs linked to QTLs for a total of 340 control and variant constructs. ...Mutations causing regulatory change and synonymous or non-synonymous codon exchange underscore the relevance of regulatory effect prediction in parallel to gene model centered functional predictions.”

In accordance with the changes and updates, the following manuscript objects have been edited or added:

Figure 6: We have added Figure 6 panel f and g to highlight the models verifications results.

Suppl. Table 10

Suppl. Table 11

Suppl. Figure 9

Suppl. Text

Reviewer #4 (Remarks to the Author):

Fritz, Jedrzej and their colleagues developed a multi-labelled deep learning model trained on DAP-seq data from *Arabidopsis thaliana* to capture how DNA sequence features, their context, and syntax influence transcription factor occupancy across the genome. Additionally, this model possesses a certain degree of transferability, enabling it to annotate transcription factor binding sites on the maize genome with good performance. Below are some of my concerns:

Major issues

Comment #4.1

Training dataset: the authors only used DAP-seq which evaluated the in vitro motifs of TFs. There is also a large amount of in vivo motif data (CUT & Tag or ChIP-seq) in Arabidopsis, which may further enhance the training model by increasing additional model layers of tissue specificity.

Response: We thank the reviewer for this important comment. Indeed, incorporating in vivo binding datasets such as ChIP-seq or CUT&Tag, could, in principle, increase the diversity and biological specificity of the training data. However, ChIP-based datasets in *Arabidopsis* derive from numerous independent studies that differ substantially in experimental protocols, data quality, sequencing depth, and signal-to-noise ratios. This heterogeneity introduces considerable label noise and complicates model training and interpretability. To directly assess the feasibility of integrating in vivo datasets, we evaluated *Arabidopsis* DAP-seq trained models on a comprehensive collection of 127 DAP-seq and 71 ChIP-seq datasets available for *Zea mays* (Ricci et al. 2019; Galli et al. 2025; Tu et al. 2020). As shown in Supplementary Figure 10 and Supplementary Table 2, we achieved good performance on the *Zea mays* DAP-seq data, while the model performed much worse on the *Zea mays* ChIP-seq data across TF families. We also considered combining DAP-seq and ChIP-seq as inputs for the same model, but this approach faces several major limitations. (1) Low resolution of ChIP-seq peaks (>1 kb): ChIP-seq peaks capture large genomic regions enriched for many overlapping

sequence features, especially in promoter and core-promoter regions. This inflates the number of candidate motifs and severely reduces the ability to identify TF-specific binding signatures. Training a combined model with the ChIP-seq resolution would severely impair the precision of DAP-seq data in highlighting TF-binding sites. (2) Interpretability challenges: Combining the two data types reduces the transparency of which signal - sequence-intrinsic or chromatin-mediated - the model is capturing. While the power of ChIP-seq lies in highlighting the TF-binding in its biological context, this context will be represented as an unobserved confounding variable for our sequence-to-binding model. Therefore, while we acknowledge that ChIP-seq and CUT&Tag represent valuable orthogonal datasets, especially for downstream validation, we consider DAP-seq the most suitable and consistent training source for the sequence-centric model presented here. Nonetheless, a data-type-aware model that explicitly accounts for input heterogeneity and differential biological context represents an exciting direction, likely requiring a very different model architecture and training strategy.

Comment #4.2

Cross-validation: the authors used 4 chromosomes (80%) for training and the remaining chromosome for validation (20%). Under this strategy, the model performance was not too high. I highly suggest the authors shuffle the genome into five or ten discontinuous sections, and re-evaluate their model by 10-fold cross-validations.

Response: We thank the reviewer for this comment. We could possibly split the genome into 10 discontinuous sections and perform 10-fold cross validations, especially since our current training strategy generates non-overlapping 250 bp windows. However, the choice of chromosome-level cross-validation facilitates downstream analyses. For example, if a gene comes from chromosome 1, then a model trained on the four chromosomes excluding chromosome 1 can be used for evaluation and predictions. This facilitates our internal verification to ensure that a 250 bp window is never evaluated by a model for which it was part of the training set, thus preventing overestimation of our model's power at inference. Furthermore, by using an entire chromosome for a validation set, we ensure that every TF family is represented within the validation set.

Comment #4.3

When implementing transfer learning in maize, the non-TF-specific DNA-binding assay used was MOA-seq. Based on my experience, the data quality of MOA-seq is not as good as that of ATAC-seq, and there are many high-quality ATAC-seq data available in maize.

Response: Thank you for pointing out that we did not discuss the possibility of using ATAC-seq data for validation. We have now added the following section to the discussion [l. 851-663] which reads as follows: Since MOA-seq is a relatively recent technique, ATAC-seq data might be more widely available in some cases or faster to generate due to its straightforward protocol without library preparation. Both methods show a strong overlap, with MOA-seq recovering 76% and 92% of ATAC-seq peaks in Maize and *Arabidopsis*,

respectively, and ATAC-seq recovering 35% and 71% of the MOA-seq peaks (Savadel et al. 2021, Zhao et al. 2020). The higher number of peaks/ TF footprints found with MOA-seq is likely caused by the smaller size of the MNase, allowing detection of sites not accessible to Tn5. In maize, MOA-seq peaks widely overlap with TF binding sites determined by ChIP-seq (66%) even when comparing different tissues (Savadel et al 2021). Due to the higher number of TF footprints and the higher resolutions of MOA-seq compared to ATAC-seq (thanks to the exonuclease activity of MNase), we opted for this technique as a more comprehensive set for comparison. Interpretation of the presented model works best when the attention window is placed non-arbitrarily, benefitting from smaller peak regions.

Comment #4.4

Throughout the article, there is no comparison with other transcription factor prediction software or models. For example, what is the difference between the transcription factor binding sites predicted by the AI model used in this article and those identified traditionally by MEME?

Response: We thank the reviewer for the comment. However, we screened the extracted IPMs against the JASPAR2204 database. This database is mirrored by MEME and includes DAP-seq and ChIP-seq motif enrichments like in (DAB2.0). We could not detect differences in the motifs extracted from our model and the TFBS in these databases.

Minor issues

Comment #4.5

There are numerous formatting issues throughout the manuscript, including but not limited to the following:

Line 74: The use of bold text appears unjustified.

Lines 111–113 and Line 326: Multiple font styles are used inconsistently.

Lines 416–425: A calligraphic or cursive-style font is used, which negatively affects readability.

Lines 674 and 685: The equations are not fully rendered.

Similar formatting inconsistencies are found elsewhere in the manuscript. The authors are advised to carefully review and standardize the formatting throughout the document.

Response: We thank the reviewer for pointing out these issues. We addressed these to our best knowledge.

Comment #4.6

The selection of training window size (250 bp) was arbitrary. The size of CREs in plants spans only dozens of basepairs.

Response: We concur with the reviewer's comment. The analyses of the IPM_{civ}, or IPM predictability, the input window offset analysis, and the investigation of singular example SHAPley scores per window demonstrate that CREs are recognized even at smaller sizes and their placement within the window.

Comment #4.7

Line 42-43, "Cis-regulation refers to the mechanism by which DNA sequences control the transcription rate of nearby genes.....", I'm not sure the use of "transcription rate" was accurate. Any citations? In my opinion, "transcription outcome" or "transcription abundance" may be more accurate.

Response: We have changed the wording here in avoidance of misunderstanding. [L.50-52] Now reads as follows: "Cis-regulation refers to the mechanisms by which DNA sequences control the transcript abundance of nearby genes and the splicing and stability of the resulting RNA molecules. Variations in *cis*-regulatory elements (CREs) have been shown to drive phenotypic diversity, adaptation and evolution "

Comment #4.8

The model predicted some point mutations that might disrupt motif bindings, (e.g. Line 374-375). Experimental validations, at least one or two cases, are needed to further verify the effect of these point mutations.

Response: The reviewer's comment addresses a weakness in our study, lacking experimental validation. We now provide a complete section and different experiments that link the model's prediction to gene expression and highlight the role of cooperative binding. The model's predictions were rigorously benchmarked against established large-scale functional datasets re-analyzing existing Plant STARRseq data from Jores et al., 2021 NatPlants and Jores et al., 2025 The Plant Cell (Jores et al. 2021, 2025). This involved correlating its predictions with measured enhancer strengths obtained from a deep mutational scan in *A. thaliana* and evaluating its applicability across species using promoter variant data from *Zea mays* and chickpea enhancer regions. In addition, we assessed the model's capacity to forecast the functional implications of natural genetic variation that were presented before by analyzing hundreds of significant SNPs from the *A. thaliana* GWAS catalogue. Our findings indicate that the model effectively predicts TFBS dynamics associated with changes in gene expression. We also observed that mutations in promoters or enhancers seldom affect the predicted binding of

a single intended TFBS; instead, they may alter the entire modeled binding profile. These distinct profiles correlated with significant changes in measured gene expression across various constructs. The analyses of selected SNPs from AraGWAS indicates that the predicted variation in TFBS occurrence coincides with changes of gene expression measured in Plant STARRseq experiments, supporting the models applicability for trait selection. Please note that changes to the manuscript were introduced here:

Changes on experimental verification of gene expression data and co-expression data

[L. 464-471] Added result on co-expression analyses for genes with shared regulatory profiles now reads as following: “Furthermore, we analysed the relationship between our regulatory clusters and gene co-expression networks. For that purpose we used co-expression datasets from the ATTED-II database (Obayashi et al. 2022). We performed the Mann-Whitney U test on standardised co-expression logit scores (co-expression z-scores), to test the statistical difference between the co-expression within each cluster and co-expression of randomly selected gene sets of the same respective size. The computed p-values have been provided as supplementary data (Suppl. Table. 8). In conclusion, we confirmed that, in addition to functional coherence, similar TF-binding profiles are indicative for coordinated gene regulation. “

In accordance with the changes and updates, the following manuscript objects have been edited:

Suppl. Table 8

[L. 472-503]: We have added a new section that describes the results on reinvestigated plant STARR-seq experiments in “Predicted TFBS Dynamics explain Gene Expression Changes”, which reads as following: “To compare the mutli-label classifiers predictions for TF binding with experimental results, ... However, variants that resulted in parallel gain of bZIP, bHLH and AP2/EREBP, together with the loss of HSF, showed significantly higher rates of gene expression (Permutation test $p \leq 0.05$) (Suppl. Figure 8)”

[L. 770-780] We have edited and added a section in the discussion to highlight the new results on combinatorial binding and experimental verification.

[L. 829-846] We have edited and added a section in the discussion to highlight the new results on combinatorial binding and experimental verification.

In accordance with the changes and updates, the following manuscript objects have been added:

Suppl. Figure 7

Suppl. Figure 8

Suppl. Table 9

[L. 543-589]: We have added a new section that describes the results on specifically designed plant STARR-seq experiments in “Genetic variation affects TF-binding potential in loci associated with phenotypic traits”, which now reads as following: “We verified the biological impact of predicted TFBS gain and loss by conducting plant STARR-seq experiments on control and SNP-variant enhancer constructs linked to QTLs for a total of 340 control and variant constructs. ...Mutations causing regulatory change and synonymous or

non-synonomous codon exchange underscore the relevance of regulatory effect prediction in parallel to gene model centered functional predictions.”

In accordance with the changes and updates, the following manuscript objects have been edited or added:

Figure 6: We have added Figure 6 panel f and g to highlight the models verifications results.

Suppl. Table 10

Suppl. Table 11

Suppl. Figure 9

Suppl. Text

References

- Bartlett, Anna, Ronan C. O'Malley, Shao-Shan Carol Huang, Mary Galli, Joseph R. Nery, Andrea Gallavotti, and Joseph R. Ecker. 2017. “Mapping Genome-Wide Transcription-Factor Binding Sites Using DAP-Seq.” *Nature Protocols* 12 (8): 1659–72.
- Galli, Mary, Zongliang Chen, Tara Ghandour, Amina Chaudhry, Jason Gregory, Fan Feng, Miaomiao Li, et al. 2025. “Transcription Factor Binding Divergence Drives Transcriptional and Phenotypic Variation in Maize.” *Nature Plants* 11 (6): 1205–19.
- Jores, Tobias, Nicholas A. Mueth, Jackson Tonnie, Si Nian Char, Bo Liu, Valentina Grillo-Alvarado, Shane Abbitt, et al. 2025. “Small DNA Elements Can Act as Both Insulators and Silencers in Plants.” *The Plant Cell* 37 (6). <https://doi.org/10.1093/plcell/koaf084>.
- Jores, Tobias, Jackson Tonnie, Travis Wrightsman, Edward S. Buckler, Josh T. Cuperus, Stanley Fields, and Christine Queitsch. 2021. “Synthetic Promoter Designs Enabled by a Comprehensive Analysis of Plant Core Promoters.” *Nature Plants* 7 (6): 842–55.
- Obayashi, Takeshi, Himiko Hibara, Yuki Kagaya, Yuichi Aoki, and Kengo Kinoshita. 2022. “ATTED-II v11: A Plant Gene Co-expression Database Using a Sample Balancing Technique by Subagging of Principal Components.” *Plant & Cell Physiology* 63 (6): 869–81.
- O'Malley, Ronan C., Shao-Shan Carol Huang, Liang Song, Mathew G. Lewsey, Anna Bartlett, Joseph R. Nery, Mary Galli, Andrea Gallavotti, and Joseph R. Ecker. 2016. “Cistrome and Epicistrome Features Shape the Regulatory DNA Landscape.” *Cell* 165 (5): 1280–92.
- Ricci, William A., Zefu Lu, Lexiang Ji, Alexandre P. Marand, Christina L. Ethridge, Nathalie G. Murphy, Jaclyn M. Noshay, et al. 2019. “Widespread Long-Range Cis-Regulatory Elements in the Maize Genome.” *Nature Plants* 5 (12): 1237–49.
- Tu, Xiaoyu, María Katherine Mejía-Guerra, Jose A. Valdes Franco, David Tzeng, Po-Yu Chu, Wei Shen, Yingying Wei, et al. 2020. “Reconstructing the Maize Leaf Regulatory Network Using ChIP-Seq Data of 104 Transcription Factors.” *Nature Communications* 11 (1): 5089.
- Zenker, Sanja, Donat Wulf, Anja Meierhenrich, Prisca Viehöver, Sarah Becker, Marion Eisenhut, Ralf Stracke, Bernd Weisshaar, and Andrea Bräutigam. 2025. “Many Transcription Factor Families Have Evolutionarily Conserved Binding Motifs in Plants.” *Plant Physiology* 198 (2). <https://doi.org/10.1093/plphys/kiaf205>.

REVIEWER COMMENTS

Reviewer #1 (Remarks to the Author):

The revised manuscript shows meaningful progress in several areas, particularly in the addition of functional validation experiments and the incorporation of more rigorous performance metrics. These improvements strengthen parts of the authors' original claims and represent a substantial effort to address earlier feedback. However, two fundamental issues remain largely unresolved and continue to limit the rigor and readability of the work.

First, the submission lacks figure legends and contains figures that are difficult to interpret, raising concerns about overall presentation quality and editorial care. Second, despite new analyses, much of the manuscript remains descriptive rather than quantitative, particularly in its interpretation of model behavior and motif-level features (e.g., IPMs). As reflected throughout the detailed comments below, key claims are still supported by curated examples rather than systematic evaluation, and central interpretability metrics remain underspecified or ambiguously validated.

Together, these unresolved issues undermine confidence in the robustness and clarity of the study, even in light of the substantial revisions made.

Response

We appreciate you highlighting the manuscript's weak points and suggesting improvements. We have since enhanced the quality of the figures. Regarding the legends, these have been submitted in accordance with the Nature Communications submission system. To ensure that the figures have their respective captions, we have manually added the figures and captions directly after the "**References**" section.

Comment #nM1

The authors acknowledge the dependence on DAP-seq dataset availability and attempt to mitigate it via weighted loss and cross-species validation. However, the empirical demonstration that the model generalizes well to TFs or conditions with sparse or no training data remains limited. I encourage the authors to provide a more systematic analysis of performance under data-limited scenarios in the main text rather than relying on curated examples in supplemental figures. For instance, what's the overall trend/correlation between sample size and model performance?

Response

We thank the reviewer for this constructive critique. We agree that a systematic evaluation is superior to curated examples. Here, we have addressed this by performing three new quantitative analyses now detailed in the section "**Analysing the effects of class distribution on model performance**" (ll. 106-146) and shown in the updated Figure 2. In this section we give a detailed account of how we mitigate data imbalance using weighted training strategies. Thanks to other reviewer comments we also trained models using smaller offset sizes and gave a report on how this in combination with the weighted training further improves model performance (Fig. 2). Furthermore, we have trained models on subsets of the available classes; bottom-k models were trained on the ten least abundant families and top-k models on the ten most abundant families (Fig. 2d and 2e). This analysis helps demonstrate the fact that the majority classes can overshadow the minority classes during training. We observed higher Matthews correlation coefficients (MCC) for the bottom-k models on specific families (e.g., MADS and ABI3VP1) that were previously not learned by the unweighted model. The bottom-k model was also capable of learning signals from the RWPRK family which the full models were unable to learn. However, the weighted model trained on 250 bp windows with 75 bp offsets showed a better balance between training on all families and learning signals from the minority classes. The correlation (Pearson's r) between data abundance (number of recorded DAP-seq peaks) and Matthews correlation coefficient was lower for the weighted training ($r = 0.28$; p -value = 0.054) and weighted training with 75 bp offsets ($r = 0.25$; p -value = 0.087) compared to the unweighted training ($r = 0.43$; p -value = 0.002), suggesting that weighting training indeed mitigates the issue of data imbalance (Fig. 2c).

Manuscript Figure 2 - Explanation of performance for the transcription factor binding sites (TFBS) predictive standard and optimized models of *Arabidopsis thaliana*. **(a)** The predictive performance measured in weighted auPR of five optimized models, each trained on ($n = 4$) *A. thaliana* chromosomes, was evaluated for an offset of 75, 125 and 250 bp. This evaluation included a comparative analysis against single- and di-nucleotide shuffle controls (s-baseline, d-baseline). **(b)** Comparison of the model performances in Matthews Correlation Coefficient (MCC) trained on the full dataset, weighted training of the full dataset and weighted training of the full dataset with an offset of 75bp. Models were evaluated on the same dataset which was created with a 250 bp offset. **(c)** Comparison of the models Pearson Correlation (Pearson's R) class specific performance measured in MCC with data abundance trained on full dataset, weighted training of the full dataset and weighted training of the full dataset with an offset of 75bp (p -value *, **, *** < 0.05, 0.01, 0.005). **(d)** Class specific comparison of models trained on all 46 and on the subset of 10 most abundant (top-k) transcription factor (TF) families ranked by training data abundance. Barplots show the MCCs of the models trained on the full dataset, weighted training on the full dataset, weighted training on the full dataset with an offset of 75bp and unweighted training for the ten most abundant TF families. **(e)** Class specific comparison of models trained on all 46 and on the subset of 10 least abundant (bottom-k) transcription factor (TF) families ranked by training data abundance. Barplots show the MCCs of the models trained on the full dataset, weighted training on the full dataset, weighted training on the full dataset with an offset of 75bp and unweighted training for the ten least abundant TF families.

Comment #nM1.1:

The authors attempt to counter the concern of diminished predictive utility by showing that known motifs can still be recovered for weakly performing TF families (e.g., ARF). However, this response conflates feature attribution (motif enrichment) with predictive performance. The ability to identify known motifs in low-performing models does not establish meaningful generalization or reliability. A model can overfit or capture background sequence biases and still return recognizable motifs — especially when interpretability methods like SHAP or TF-MoDISco are applied post hoc.

Response

We thank the reviewer for this insightful distinction. We agree that post-hoc motif recovery (attribution) does not equal predictive reliability. To address this, we have shifted our analysis from simple motif identification to a systematic evaluation of predictive precision vs. motif occurrence presented as our "null hypothesis", where motif occurrence predicts binding. In short, simply matching an IPM/TFBS across the genome results in an average False Discovery Rate (FDR) of 81.9%. By incorporating sequence syntax/context, our model reduces the FDR to 60.5% (a ~21% improvement over the motif-only baseline) using the non-weighted training without offset (**revised Suppl. Table 6**). However, our new analysis shows that the limited predictive power for these families is not a result of the model simply defaulting to background sequence biases. If the model were merely overfitting to the presence of a recognizable motif, its false positive rate would mirror the 81.9% FDR of the motif-only null hypothesis. Instead, the model's precision in windows containing IPMs demonstrates that it applies more conservative, context-aware criteria for binding, proving it incorporates broader sequence syntax rather than relying on a single TF family's core motif. By shifting our evaluation from post-hoc attribution to these forward-predictive metrics, we can isolate the model's actual reliability (discussed and presented above).

In the section "**Sequence context and motif syntax contribute to predicting binding specificity of different TF families**" (ll. 181 -282) we now address the crucial distinction of TFBS and IPMs. The text now reads as (ll. 197- 202): "*To evaluate the predictive importance of each IPM and distinguish it from background sequence bias, we developed the IPM context importance value (IPMciv).*"

Moreover, citing better motif recovery from ARF_ecoli compared to ARF_tnt highlights known experimental biases in DAP-seq, not successful generalization by the model. This reinforces the concern that the model is heavily influenced by dataset composition and lacks robustness under data-sparse or heterogeneous conditions. We fully agree with the reviewer's assessment regarding the technical biases inherent to DAP-

seq expression systems (e.g., *E. coli* vs. TNT). Using this specific comparison to argue for generalization was flawed, as it conflates biological signals with experimental artifacts. Consequently, we have removed this anecdotal comparison as a proof of generalization. Instead, to rigorously address the concern about robustness under data-sparse conditions, we introduced the systematic k=10 sub-model benchmark (comparing the 10 lowest-abundance families against the 10 highest). This quantitative ablation study, combined with our new 75 bp offset-weighted training strategy, explicitly demonstrates how we decoupled model performance from dataset composition and recovered predictive accuracy (MCC) for sparse families like MADS and TCP.

In the section “**Analysing the effects of class distribution on model performance**” (ll. 106-146) we now detail how we addressed the models limitations in data-sparse regimes. The text now reads as (ll. 114-146): “*To mitigate the observed effects of class imbalance, we explored two complementary strategies aimed at upsampling the minority classes and adjusting learning dynamics. ...*”

Comment #nM1.2:

In summary, the authors’ interpretation suggests a fundamental misunderstanding of the distinction between motif attribution and predictive accuracy. Without quantitative benchmarks or prospective evaluation in low-data regimes, the argument that “some motifs can still be found” does not resolve the original conceptual flaw.

Response

We appreciate the reviewer pressing this critical conceptual point. We completely agree: feature attribution does not equal predictive accuracy, and simply 'finding a motif' does not validate a model.

To resolve this conceptual flaw, we have adjusted our interpretation to provide the exact quantitative benchmarks and low-data evaluations requested. We established a strict 'motif-only' null hypothesis. We quantitatively demonstrate that relying solely on motif attribution yields a higher average FDR in contrast to our model's predictive power—relying on learned sequence syntax. In addition, we explicitly benchmarked performance in data-sparse regimes. As shown in our new Figure 2 (panels c, d, and e), our offset-weighted training strategy successfully decoupled predictive performance from data abundance (Pearson's r dropping from 0.439 (p-value 0.002) to 0.254 (p-value 0.08)). We now report strict predictive metrics (MCC, Sensitivity, F1) for all rare families, proving true predictive recovery (e.g., TCP, MADS) rather than mere motif attribution.

We have addressed the raised issue and provide results in the section “**Model interpretation for de-novo identification of TF family-specific binding motifs**”. The text now reads as (ll. 165): “*Consistent with feature attribution highlighting TFBS-like IPMs, the models recovered on average 50.4–94.5% of known binding sites for AP2/EREBP, bHLH, bZIP, G2like, HSF, MYB, and WRKY families from the JASPAR database across the genome (Suppl. Table 6), ...*”

In addition, we reference the individual recovery rates in comparison to the null-hypthesis throughout the section “**Sequence context and motif syntax contribute to predicting binding specificity of different TF families** “ (ll. 181 -281).

Comment #nM3:

While the authors implemented weighted loss and report mixed outcomes; however, this does not eliminate the underlying issue that model performance is strongly dependent on TF families with abundant high-quality data. This reinforces the broader conceptual concern raised in M1 — that the model’s predictive utility is limited in biologically relevant scenarios where data are sparse, thereby constraining generalization and interpretation.

Response

We thank the reviewer for raising this critical point. The reviewer correctly notes that applying a weighted loss function *alone* yielded mixed outcomes. However, we wish to clarify that the isolated weighted loss was only an intermediate evaluation step designed specifically to illustrate the challenges of the long-tail label problem.

We completely agree that a model limited only to data-abundant TFs lacks biological utility. As detailed in our response to comment M1, to address this issue, we implemented a combined 250 bp sliding-window with 75 bp offset + weighted loss strategy. While this unified architecture directly resolved the data-sparsity problem for several rare TF classes (e.g., TCP, MADS, RAV1), we acknowledge that this strategy could not resolve the issue of low data abundance completely. Even when we trained a sub-model *exclusively* on the least abundant classes (to remove the effects of majority classes overshadowing minority classes during training), transcription factor families like ARF_tnt and EIL_tnt still could not be recovered (**revised Fig. 2**). This demonstrates that for these specific families, the limitation lies in incomplete or highly variable experimental training data, rather than the model's architecture.

Finally, the evaluation forced us to make an architectural trade-off regarding the RWPRK family. RWPRK was the only TF family that could exclusively be recovered by the sub-model trained on least abundant classes (**revised Fig. 2e**). However, adopting a multi-model approach just to save this single family would isolate it from the rest of the network, preventing the model from learning potential TFBS combinations and co-occurrences between RWPRK and high-abundance TFs. Because capturing these complex combinatorial modules is the primary strength of our unified architecture, we opted to drop the RWPRK class from the final model rather than sacrifice the network's combinatorial predictive power.

We are addressing the long-tail problem in multi-label classification tasks within the manuscript in section “**Analysing the effects of class distribution on model performance**”. The text now reads as (ll. 114): “*To mitigate the observed effects of class imbalance, we explored two complementary strategies aimed at upsampling the minority classes and adjusting learning dynamics. ...*”

Explicitly, we reference the revised Fig. 2e described above as follows (ll. 141): “*In contrast, the bottom-k model showed substantially stronger improvements across the subset of minority families, most notably learning to predict the RWPRK_tnt family (MCC = 0.49) (Fig. 2e; Suppl. Table 4). However, some families (e.g., EIL_tnt, ARF_tnt) remained difficult to predict, indicating that factors beyond class frequency also contribute to performance disparities.*”

Comment #M2:

The authors have substantially improved the manuscript by including both re-analysis of public Plant STARR-seq datasets and new experiments on 340 variant enhancer constructs. These additions provide convincing functional support for the model's predictions and motif-based interpretations, particularly regarding combinatorial TFBS (IPMs).

Comment #M4|M6:

The authors have commendably incorporated MCC and other robust performance metrics throughout the revised manuscript. However, their interpretation of the model's biological relevance remains heavily reliant on the concept of interaction predictive motifs (IPMs) — composite features derived from SHAP explanations and TF-MoDISco clustering. The use of IPM and IPMciv raises several concerns:

- a) **Conceptual Ambiguity:** *The manuscript does not provide a clear, quantitative definition of IPM specificity or how these motifs are distinguished from background sequence patterns or spurious correlations.*

Response: We thank the reviewer for detailing these concerns, which highlighted an area where our original manuscript was unfortunately ambiguous. We agree that our previous descriptions of Interaction Predictive Motifs (IPMs) lacked the necessary mathematical formalization and orthogonal validation (or were hidden in the supplementary material). We also appreciate the observation regarding the BBRBPC anomaly. To address these concerns, we have expanded our results and revised the presentation of the here established metrics (inc. IPM context importance) to detail the exact quantitative frameworks used.

We have clarified that IPMs (extracted via SHAP and TF-MoDisco) represent the model's internal feature attribution and possess no standalone specificity without the model's contextual gating. To quantitatively distinguish functional IPMs from background sequence matches, we established the IPM Context Importance.

As detailed in the revised Methods, we mapped extracted IPMs against *A. thaliana* chromosome 1 using BLAMM. The IPM_{civ} is strictly defined as the ratio of windows predicted as bound (and containing the IPM) divided by the total genomic occurrences of that IPM. Furthermore, we assessed locational certainty via window offset independence (using a 250 bp sliding window with a 10 bp step size) to prove the model's predictions rely on stable, combinatorial sequence signals rather than spurious background alignments. We now explicitly state that IPMs are merely representations of the model's 'knowledge' and have no standalone specificity. However, by extracting them, we can prove that the model successfully anchors its predictions on *true* TFBS (which inherently possess higher sequence information), rather than learning aberrant background noise. Furthermore, because publicly available TFBS datasets frequently suffer from fragmented or incomplete annotations, utilizing IPMs provides a necessary, high-resolution insight into the model's actual capabilities. We have updated the manuscript to ensure IPMs are framed strictly as an interpretative baseline, not a predictive biological claim.

We have included the above points, addressing the reviewers remarks in the section “**Sequence context and motif syntax contribute to predicting binding specificity of different TF families**” (ll. 181-282). The text now reads: “Occurrences of IPMs can not explain the model's predictions alone, indicating that models likely learned extended sequence context ...”

b) **Biological Validation:** *There is limited orthogonal validation to support the functional relevance of IPMs.*

Response: To validate the functional relevance of these motifs beyond the model's internal logic, we implemented two independent validation layers. We benchmarked predictions against experimentally derived TFBSs from JASPAR2026 (incorporating independent ChIP-seq and PBM data). We show that baseline sequence-homology searches yield an average False Discovery Rate (FDR) of 81.9%, whereas combining these motifs with our model's learned syntax reduces the FDR to 60.5% (Suppl. Table 04_rev). We explicitly tested the biological relevance of the model's predictions using plant STARR-seq assays. We synthesized a library of 351 sequence pairs based on AraGWAS SNPs and tested them in *N. benthamiana*. To ensure the predicted disruption of an IPM was not a background artifact, we generated 30 *in silico* random background mutations per SNP. The STARR-seq experiments confirmed that variants causing a predicted loss of TF binding (e.g., bHLH, bZIP, WRKY) yielded statistically significant changes in *in vivo* regulatory activity (Wilcoxon rank-sum test, $P < .05$).

We have included the above points, addressing the reviewers remarks in the section “**Genetic variation affects TF-binding potential in loci associated with phenotypic traits**” (ll. 357 – 407). The text now reads: “*We demonstrated how single nucleotide polymorphisms (SNPs) alter predicted TF-binding profiles that influence gene expression.*”

In addition to this, we formulated a null-hypothesis of IPM occurrence, comparing it to the model predictive performance and comparison to database TFBSs occurrences and their recovery rates in the following two sections

(1) “**Model interpretation for de-novo identification of TF family-specific binding motifs**” (ll. 147 – 180). The text now reads as (ll. 165): “*Consistent with feature attribution highlighting TFBS-like IPMs, the models recovered on average 50.4–94.5% of known binding sites for AP2/EREBP, bHLH, bZIP, G2like, HSF, MYB, and WRKY families from the JASPAR database across the genome (Suppl. Table 6), ...*”

(2) “**Sequence context and motif syntax contribute to predicting binding specificity of different TF families**” (ll. 181-281). The text now reads as (ll. 229): “*Notably, however, the application of the model still increased precision compared to baseline based on the occurrence of IPMs solely by 1.605 and 3.65 fold for AP2/EREBP and BBRBPC, respectively. The non-optimized model recovers 94.5% of characterized TFBS of the the AP2/EREBP family from previous DAP-seq and PBM experiments, intriguingly, including for example ERF12, ERF14 and ERF17 that were not present in the training data, with 90%, 97.4% and 92%, respectively.*”

- c) **Potential for Misinterpretation:** *In families such as BBRBPC, the model exhibits high sensitivity but also extremely high FDR (up to 98%). Yet IPMs from these families are interpreted as predictive, which may misrepresent model reliability—especially in the context of label imbalance and family-specific data scarcity.*

Response: The reviewer is entirely correct that highlighting BBRBPC as "predictive" misrepresented its reliability. BBRBPC exhibits high sensitivity but massive FDR precisely because it represents an outlier where the model relies almost entirely on the core motif itself (an exceptionally low IPM_{civ}). In the case of BBRBPC, the multi-label classifier overestimates the core motif's importance and indeed fails to utilize this combinatorial syntax. By categorizing predictive events into discrete combinatorial bins (0 to 11 co-occurring TFs) using 100 bootstrap replicates, we found that highly reliable families rely on Weighted Average Co-occurrence and the sequence context measured in IPM_{civ}. Here, we detected BBRBPC as an outlier. We have rewritten the text and updated Suppl. Table 6 to explicitly contrast context-dependent families (like HB or BZR) with context-independent outliers (like BBRBPC), properly framing it as a cautionary example. Hereby, we provide a critical in depth analysis of how a deep learning model, usually presented as a "black box", can be dissected and weaknesses can be detected.

To address the raised points, we have rewritten the following section "**Sequence context and motif syntax contribute to predicting binding specificity of different TF families**" (ll. 181-281). The text now reads as (ll. 221): "*For instance, IPMs of AP2/EREBP (5'-GCGGC-3') and BBR/BPC (5'-GAGAGA-3') show high predictive performance...*"

Major concern: figure quality remains unresolved

Despite earlier feedback, figure layout and clarity remain below publication standards. Moreover, the submitted version lacks figure legends, making it difficult to assess whether the visualizations are interpretable, self-contained, or aligned with the narrative. The omission of legends and ongoing presentation issues raise concerns about the rigor and overall quality of the manuscript.

Response

We made substantial improvements to figures of the manuscripts, adding legends where missing and moving some figures to the supplement. Also, to assure the availability of the figure legends, we provide a supplementary document with all figures together with their respective captions.

Reviewer #2 (Remarks to the Author):

The authors have put in quite some effort to address all points of critique voiced by the four reviewers, even including novel experimental procedures to support their in-silico predictions. However, despite their efforts, I am sorry to say that by reading and reviewing the study again, and based on the provided response, I have grown more concerned about the study, actually.

Comment #1.

Specifically, the second time around, I feel the authors are overreaching. Yes, “feel” is not a scientific review category, but I’d like to offer a plausibility argument first. There are ~1500 TFs in the Ath genome, with a few hundred of them with available DAPseq data. In the study, the authors opted to condense them to 46 families. That is reasonable. But given that within each family, binding site motifs, while similar, vary, how can one reasonably expect that single-base pair variants will correctly reflect gain or loss of TF (family)-binding? In fact, 33% of the random allelic changes produced “de novo perturbation”. That many? To me, this looks more like frequent random fluctuations around the decision boundary, from score value 0.49 to 0.51, for example. Yes, as requested, you have provided experimental support, but it remains anecdotal (despite best and commendable efforts!). Furthermore, non-synonymous GWAS candidate SNPs (as the one mentioned in l440) may be more likely associated with phenotype via protein-related effects, not by expression-regulatory ones.

Response

We sincerely appreciate the reviewer’s critical assessment and understand the plausibility concerns raised. Predicting family-level binding dynamics from single-nucleotide variants (SNPs) is indeed a bold claim, which is exactly why we subjected it to rigorous *in silico* background checks and *in vivo* validation. In the revised version of the manuscript, we address these valid concerns mathematically and experimentally as follows:

We systematically quantified and plotted the distribution of the change in predicted probability (Δ prob) for all variants classified as inducing a gain or loss of TF binding (Suppl. Fig. 13). We observed that 41.73% of all predicted binding changes had a Δ prob \geq 0.2. This quantification confirms that a substantial portion of these SNPs are not merely causing random statistical fluctuations around the 0.5 boundary. Instead, they induce severe alterations in the model’s binding probabilities, indicating the disruption or creation of highly informative *cis*-regulatory features.

Manuscript Supplementary Figure 13: The distribution of predicted probability changes (Δ prob) for 250 bp windows where an SNP led to a predicted gain or loss in TF binding.

In addition, the 33% *de novo* perturbation rate from random *in silico* mutations is unlikely statistical noise, but rather a reflection of regulatory density. For this analysis, we exclusively extracted gene flanking regions (promoters and terminators) associated with known GWAS hits. As shown in our baseline analysis (**Fig. 4a**), these specific regulatory windows are dense with TFBSs (most true predicted windows are bound by multiple TF families simultaneously). The high density of regulatory syntax in these GWAS regions likely raises the biological probability that a random mutation will unintentionally create or destroy a secondary motif, resulting in the observed 33% *de novo* rate. Therefore, a baseline percentage for random mutations expected to alter predicted binding (p-value of 0.001) can be calculated (Revision Figure 1).

Revision Figure 1: Baseline sensitivity of distinct *Arabidopsis thaliana* genomic features to *in silico* sequence perturbation. The bar plot displays the percentage of predicted TF binding changes induced by a uniform 12% background mutation rate across different genomic regions. Notably, non-coding and regulatory environments (e.g., intergenic spaces, introns, and 5'UTRs) exhibit intrinsically higher baseline perturbation rates (~4.4–4.5%) compared to coding sequences (CDS; 3.98%).

We have modified the result subsection titled “**Genetic variation affects TF-binding potential in loci associated with phenotypic traits**” (ll. 357 – 406) and stated the observation about the delta probabilities. This reads (ll. 367): “Second, to ensure that the predicted gains and losses in TF binding were not artifacts of minor statistical fluctuations around the classification threshold (e.g., marginal shifts between 0.49 and 0.51), we systematically quantified the change in predicted probability for all variants. Indeed, 41.73% of the binding changes exhibited a shift in probability larger than 0.2 (Suppl. Fig. 13).”

Finally, we fully agree that non-synonymous GWAS SNPs affect phenotypes through protein alterations. However, by focusing our *in silico* perturbation analysis on gene flanking regions and evaluating feature sensitivity, our model quantifies regulatory syntax disruptions independent of coding potential. We do not claim expression-regulatory effects supersede protein-level effects for non-synonymous SNPs, but that our model successfully uncouples and quantifies this orthogonal regulatory layer.

Comment #2.

I appreciate that my suggestion to look at co-expression was followed up on. But I am surprised by the lack of technical precision/ clarity in the respective segments. Suppl Table 8 is supposed to show "Supplementary table 8 - Enrichment of regulatory binding modules under co-expressed gene regulatory networks", but instead, p-values are provided for the different clusters (raw and adjusted), i.e. where is the enrichment mentioned in the caption? Then: p-values alone do not tell the relevant story. It could also be significantly LESS correlated! So, what is the effect size (with sign)? What is the "random gene set"? It should be genes from the actual set (genes that belong to a cluster, not just any Ath gene), randomly paired up. Was it done that way?

*In the spreadsheet ("adjusted") - there are values greater than one. p-values are by definition between zero and one, even if adjusted. Are those values raw-p-value * N, i.e. kind of Bonferroni? This is a bit unorthodox, as one would apply the correction to the threshold, but ok... But one needs to be clear about it. What are the different test columns? Repeated random sets?*

Response

We sincerely thank the reviewer for this rigorous methodological check. We agree that our previous statistical presentation in Supplementary Table 8 lacked the necessary precision and clarity regarding multiple testing corrections. The reviewer was entirely correct in their assessment of the previous spreadsheet: the values greater than one were indeed the result of an unorthodox Bonferroni correction output by the previous script (raw p-value * N). We have replaced that approach with a permutation testing (n = 1000) followed by Benjamini-Hochberg FDR correction. We currently provide Supplementary Table 10 to report the observed mean co-expression scores, the effect size, and the corrected p-values. Furthermore, we have added a new subsection under Methods titled "**Linking regulatory clusters to gene co-expression.**" (ll. 665 – 672) The following steps outline the permutation framework:

1. We computed the mean coexpression logit score for gene pairs within the cluster.
2. To determine whether this value was significantly high, we generated a null distribution by randomly sampling the same number of genes from the full set (genes from the 14 clusters) and computing the mean of this random sample. This was repeated 1000 times.
3. The p-value was calculated from the proportion of random samples whose mean coexpression logit score was greater than or equal to the observed cluster mean.
4. P-values were adjusted for multiple testing using Benjamini-Hochberg FDR correction.

Manuscript Supplementary Figure 10: Permutation test (n=1000) revealed that genes belonging to the same regulatory binding cluster are coexpressed together. Genes belonging to eight of the fourteen regulatory clusters show significantly higher coexpression logit scores compared to genes selected randomly from the pool of fourteen clusters.

Comment #3.

Suppl. Table 2, detailed performance metric (and respective paragraphs/conclusions drawn from it in the main text): Following a reviewer request, you added F1 (micro/macro) scores. But why do you get different values for the different TF-families? As far as I understand the F1-multilabel reporting, there is one(!) F1 micro/macro. In fact, if F1, I prefer a per-class F1. But ok, MCC-class-wise is provided, which I consider informative enough. What I do miss are false-positives and true-negative counts. You provide in Suppl Fig 4C a dedicated view of FPs (which is nice). FP-counts are astonishingly low. But what are the true-negative counts? In a multilabel situation, these confusion matrices are somewhat challenging to begin with. Per-class analyses may be more helpful after all.

Response

We thank the reviewer for this observation. To clarify, yes, the values we reported were actually per-class F1 scores. We calculated them for each individual TF family because we calculated performance metrics independently for each class to provide a granular view of model performance across the different families. Second, we completely agree that providing the underlying counts is essential for full transparency, especially to contextualize the "astonishingly low" False Positive (FP) counts. In the context of genome-wide TF binding prediction, the dataset is inherently imbalanced; the vast majority of genomic windows do not contain a binding site for a given TF family. Consequently, the True Negative (TN) counts are exceptionally high, which naturally constrains the raw FP counts and underscores why we prioritize the Matthews Correlation Coefficient (MCC) for evaluating performance.

Consequently, we have updated Supplementary Table 2 to include the counts for TPs, FPs, TNs, and FNs for every TF family. These values provide the necessary context for the reported MCC and F1 scores, allowing for a clear assessment of the model's performance in the multilabel setting.

Comment #3.1.

In this context, what are you actually predicting? Sequence windows of length 250 bp for the whole hold-out chromosome? Or around TSS/TPS in that chromosome only? It was not clear to me.

Response

Our evaluation was performed genome-wide on the held-out chromosome. Specifically, we utilized a chromosome-level cross-validation strategy: the model was trained on non-overlapping 250 bp windows from four chromosomes and then evaluated on all 250 bp windows of the fifth, held-out chromosome. As such, for each cross-validation round, a chromosome becomes the validation set. The results from these individual held-out chromosome evaluations were then aggregated to compute the final genome-wide performance metrics and to generate the multi-label confusion matrix.

To ensure this is clear to the reader, we have explicitly stated in the revised Methods section under the section "**Model architecture and training strategy**" (ll. 583 – 596) and reads: "We performed 5-fold cross-validation such that, in each cross-validation step, windows from four chromosomes were used for training the model, while windows from the left-out chromosome were used for validation."

Comment #4.

On the "offset" discussion. I am not quite in agreement with the arguments provided in defense of the non-overlapping window approach - redundancy and consistent input size. Both would not be a problem if performed 250bp window step size 125bp, for example. Anyway, for the training, it may not matter (or did you center "0" on every gene TSS/TTS)?, but during inference, it could make a big difference if the window is off (boundary effects). The authors addressed it with incremental 10bp offset tests. But here, I do not understand the conclusion: l239, "This indicates that TF families with high performance are more often predicted independent of their placement within an input window and a distinct composition of the sequence context". But: that is exactly the strength of CNN to pick up motifs irrespective of their relative position. I don't quite agree that sequence context/composition explains good performance. To me, it instead shows the prudent choice of CNNs. The only concern would be cut-off/boundary effects, in my opinion.

Response

We thank the reviewer for this insightful comment. We agree that a non-overlapping window strategy can introduce boundary effects where critical motifs are split between windows, potentially leading to false negatives during inference. In response, we have implemented a training strategy using offsets. Specifically, we trained two additional sets of models using datasets generated with 75 bp and 125 bp offsets. As the reviewer hypothesized, we observed a significant increase in model performance as the offset size decreased (Fig. 2). This confirms that smaller offsets mitigate the edge cases where motifs were previously bifurcated at window boundaries. Furthermore, as discussed in our responses to prior comments, this overlapping architecture also proved crucial in improving predictive performance for TF families suffering from data scarcity.

Manuscript Figure 2a: Explanation of performance for the transcription factor binding sites (TFBS) predictive standard and optimized models of *Arabidopsis thaliana*. a) The predictive performance measured in weighted auPR of five optimized models, each trained on (n = 4) *A. thaliana* chromosomes, was evaluated for an offset of 75, 125 and 250 bp. This evaluation included a comparative analysis against single- and di-nucleotide shuffle controls (s-baseline, d-baseline).

We also appreciate the reviewer's perspective on the inherent strengths of CNNs. We agree that a primary advantage of CNNs is their ability to identify motifs irrespective of relative position. Our conclusion was intended to highlight that for high-performing families, the model's 'certainty' remains stable despite shifts in the surrounding sequence context and placement of a prediction window. This property, which we term 'offset independence,' is particularly relevant when prediction windows are artificially centered over an IPM, TFBS, or SNP. Our goal with this test was strictly to control for positional overfitting. We have rephrased this section to clarify that high performance is a result of both the model's robustness to positioning and its ability to distinguish core motifs from flanking sequence noise, thereby addressing the reviewer's concern about boundary effects.

We have addressed the comments above in the section of "**Analysing the effects of class distribution on model performance**" (ll. 106 – 146). The text now reads as (ll. 114): "*To mitigate the observed effects of class imbalance, we explored two complementary strategies aimed at upsampling the minority classes and adjusting learning dynamics. First, we modified the training sets by generating overlapping 250 bp sequence windows using step sizes of 75 bp and 125 bp, thereby increasing the effective representation of labelled regions. Second,...*"

In addition, we attempt to clarify the meaning of the here introduced "offset independence" as described above in the section of "**Sequence context and motif syntax contribute to predicting binding specificity of different TF families**" (ll. 181 – 281). The text now reads as (ll. 207): "*The average offset independence per TF family values exhibited a strong positive correlation explaining most of the variance between average MCC per TF family, too (Pearson's $R = 0.8$, p -value $4.3e-10$, Fig. 4c). This indicates that TF families with high performance required less sequence information and were independent of their placement..*"

Comment #5.

Finally, I wonder (and I should have wondered already the first time around. So at this point, this is to be taken as an afterthought) - How does all of the deep learning compare to results, if one had taken the PWMs reported for the DAPseq set(s) and gone with FIMO to annotate the sequence regions? It would have been nice to know.

Response

We appreciate the reviewer's insightful 'afterthought,' as it directly addresses the core question of whether deep learning provides added value over traditional motif-scanning methods and linked it to other open issues of our work. To address this, we established a 'null hypothesis' where motif occurrence alone indicates binding. We performed a comparable analysis on experimentally verified TFBS and described the model's sensitivity.

Using a custom PWM scanning pipeline (using FIMO; see updated Methods), we scanned the *A. thaliana* genome for these PWMs using an 85% match threshold. Our results show that our model mostly identifies “true” TFBS across the genome. We show that motif mapping e.g. for the IPMs results in an exceptionally high average FDR of **81.9%**. In contrast, our model, which incorporates broader sequence syntax and context, reduces the FDR to **60.5%**—a ~21% improvement over the PWM-only baseline. This confirms that the model is not merely acting as a motif scanner but is successfully filtering out non-functional motif matches by learning the surrounding genomic 'grammar'.

We have included these findings in the main manuscript section “**Sequence context and motif syntax contribute to predicting binding specificity of different TF families**” (ll. 181 – 281), e.g. highlighting detection of distinct TFBS like bHLH10. The text now reads as (ll. 264):
“The model's MCC for bZIP and bHLH was 0.63 and 0.644, respectively, compared to 0.503 for BES1 ...”

The **methods** section was extended and now reads as (ll. 655):
“To evaluate the added value of the deep learning model over traditional motif-scanning approaches, we utilized annotated motifs from JASPAR2026, including TFBS from different experimental sources (E.g. DAP-seq, ChIP-seq, SELEX et c.). The *A. thaliana* genome was scanned on both forward and reverse-complement strands and genomic "hits" were recorded whenever a sequence achieved a score larger than 85% of the motif's maximum possible theoretical score using FIMO⁶⁵...”

Minor points

- It is the "Matthews" correlation coefficient, not "Mathews" or "Mathew's". Since the MCC is such a central metric in all of machine learning, it should not be that grossly misspelled by machine learning specialists.

Response

Done (thank you)

- l242 actual value of the Pearson correlation coefficient is missing, only p-value is given (2.7e-5).

Response

Now included

- "Wilcoxon" - it's an own-name, always capitalized (l411/412)

Response

Done

- l376 "We filtered..." - briefly state what for.

Response

Done

-424 for the phenotype days to flowering...- put "days in flowering" in quotes or days_to_flowering

Response

Done

-l440 "hydroxy serine" - should read "serine" - there is no hydroxy serine. Serine contains a hydroxy group to begin with.

Response

We thank the reviewer for this comment. We now refer to a "proline-to-serine" amino acid exchange.

-l695 I consider the statement: "The model proves valuable in identifying and functionally annotating intricate gene-regulatory networks." overreaching. To me, regulatory networks are cascades of TF-target_gene (TG) relationships. The authors have looked at co-occurrences of TF-binding events and predicted effects of variants. While this also reflects on relationships, it does not capture layered TF-TG networks.

We agree that the presented approach here is not capable of capturing the overall topology of gene-regulatory networks, and thus the sentence might sound overarching. We removed it from the revised version of the discussion section. Nevertheless we want to stress that our approach is still relevant for the reconstruction of the gene regulatory networks, providing partial, but fundamental information on TF-binding and biologically relevant TF-profiles, indeed to be found co-regulated.

-FYI, the noted font-changes and the missing page-numbers were still a “problem” in this revision.

Response

We sincerely apologize for the issues raised by the reviewers and the inconvenience caused, particularly regarding the supplementary material (see other reviewers responses). We believe we have now fully addressed all concerns and ensured the completeness and accessibility of the material.

Reviewer #2 (Remarks on code availability):

I looked at the GitHub repository during the first round and considered it comprehensive and informative.

Reviewer #3 (Remarks to the Author):

This is a strong study. The authors have addressed my comments.

Reviewer #4 (Remarks to the Author):

The authors have made detailed responses to the reviewers' comments and made revisions accordingly. The manuscript is now significantly improved. In particular, I note that the authors devoted considerable effort to addressing my previous concern regarding experimental validation. However, it remains unclear to me whether the hundreds of significant GWAS SNPs used by the authors were derived from trait-associated GWAS or from eQTL mapping of gene expression levels. Additionally, the line numbers referenced in the response letter do not

correspond to those in the revised manuscript, which further complicates the understanding of this section.

Response

We thank the reviewer for this comment. The GWAS SNPs we use are publicly available, manually curated and standardized collection of marker-trait associations for available phenotypes from AraPheno (<https://arapheno.1001genomes.org/>) provided by the AraGWAS catalogue (<https://aragwas.1001genomes.org/>).

In the revised manuscript we made sure it's stated clearly for the reader in the methods section and now reads (ll. 694): *“Third, to evaluate the model’s ability to predict the functional impact of natural genetic variation, we used single nucleotide polymorphisms (SNPs) identified in trait-associated genome-wide association studies (GWAS) in A. thaliana, obtained from the AraGWAS catalogue (Togninalli et al., 2018)”*.

Finally, we have also solved the issues of the mismatches in line numbering by referencing the subsections where necessary.